# Sequential Density Ratio Estimation for Simultaneous Optimization of Speed and Accuracy

**Akinori F. Ebihara[1]**    **Taiki Miyagawa[1,2]**    **Kazuyuki Sakurai[1]**    **Hitoshi Imaoka[1]**
[1]NEC Corporation
[2]RIKEN Center for Advanced Intelligence Project (AIP)
`aebihara@nec.com`

## Abstract

Classifying sequential data as early and as accurately as possible is a challenging yet critical problem, especially when a sampling cost is high. One algorithm that achieves this goal is the sequential probability ratio test (SPRT), which is known as Bayes-optimal: it can keep the expected number of data samples as small as possible, given the desired error upper-bound. However, the original SPRT makes two critical assumptions that limit its application in real-world scenarios: (i) samples are independently and identically distributed, and (ii) the likelihood of the data being derived from each class can be calculated precisely. Here, we propose the SPRT-TANDEM, a deep neural network-based SPRT algorithm that overcomes the above two obstacles. The SPRT-TANDEM sequentially estimates the log-likelihood ratio of two alternative hypotheses by leveraging a novel Loss function for Log-Likelihood Ratio estimation (LLLR) while allowing correlations up to $N(\in \mathbb{N})$ preceding samples. In tests on one original and two public video databases, Nosaic MNIST, UCF101, and SiW, the SPRT-TANDEM achieves statistically significantly better classification accuracy than other baseline classifiers, with a smaller number of data samples. The code and Nosaic MNIST are publicly available at `https://github.com/TaikiMiyagawa/SPRT-TANDEM`.

## 1    Introduction

The sequential probability ratio test, or SPRT, was originally invented by Abraham Wald, and an equivalent approach was also independently developed and used by Alan Turing in the 1940s (Good, 1979; Simpson, 2010; Wald, 1945). SPRT calculates the log-likelihood ratio (LLR) of two competing hypotheses and updates the LLR every time a new sample is acquired until the LLR reaches one of the two thresholds for alternative hypotheses (Figure 1). Wald and his colleagues proved that when sequential data are sampled from independently and identically distributed (i.i.d.) data, SPRT can minimize the required number of samples to achieve the desired upper-bounds of false positive and false negative rates comparably to the Neyman-Pearson test, known as the most powerful likelihood test (Wald & Wolfowitz, 1948) (see also Theorem (A.5) in Appendix A). Note that Wald used the i.i.d. assumption only for ensuring a finite decision time (i.e., LLR reaches a threshold within finite steps) and for facilitating LLR calculation: the non-i.i.d. property does not affect other aspects of the SPRT including the error upper bounds (Wald, 1947). More recently, Tartakovsky et al. verified that the non-i.i.d. SPRT is optimal or at least asymptotically optimal as the sample size increases (Tartakovsky et al., 2014), opening the possibility of potential applications of the SPRT to non-i.i.d. data series.

About 70 years after Wald's invention, neuroscientists found that neurons in the part of the primate brain called the lateral intraparietal cortex (LIP) showed neural activities reminiscent of the SPRT (Kira et al., 2015); when a monkey sequentially collects random pieces of evidence to make a binary choice, LIP neurons show activities proportional to the LLR. Importantly, the time of the decision can be predicted from when the neural activity reaches a fixed threshold, the same as the SPRT's decision rule. Thus, the SPRT, the optimal sequential decision strategy, was re-discovered to be an

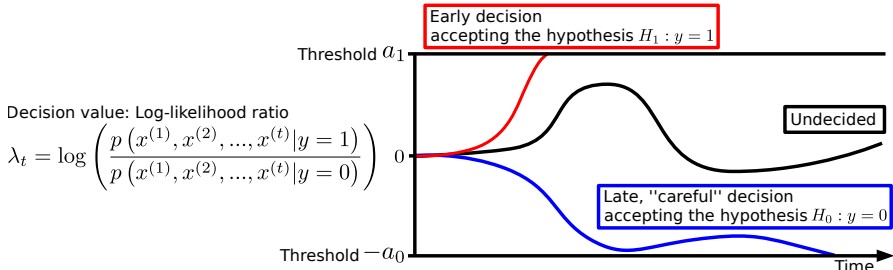

Figure 1: Conceptual figure explaining the SPRT. The SPRT calculates the log-likelihood ratio (LLR) of two competing hypotheses and updates the LLR every time a new sample ($x^{(t)}$ at time $t$) is acquired, until the LLR reaches one of the two thresholds. For data that is easy to be classified, the SPRT outputs an answer after taking a few samples, whereas for difficult data, the SPRT takes in numerous samples in order to make a "careful" decision. For formal definitions and the optimality in early classification of time series, see Appendix A.

algorithm explaining primate brains' computing strategy. It remains an open question, however, what algorithm will be used in the brain when the sequential evidence is correlated, non-i.i.d. series.

The SPRT is now used for several engineering applications (Cabri et al., 2018; Chen et al., 2017; Kulldorff et al., 2011). However, its i.i.d. assumption is too crude for it to be applied to other real-world scenarios, including time-series classification, where data are highly correlated, and key dynamic features for classification often extend across more than one data point, violating the i.i.d. assumption. Moreover, the LLR of alternative hypotheses needs to be calculated as precisely as possible, which is infeasible in many practical applications.

In this paper, we overcome the above difficulties by using an SPRT-based algorithm that Treats data series As an N-th orDEr Markov process (SPRT-TANDEM), aided by a sequential probability density ratio estimation based on deep neural networks. A novel Loss function for Log-Likelihood Ratio estimation (LLLR) efficiently estimates the density ratio that let the SPRT-TANDEM approach close to asymptotic Bayes-optimality (i.e., Appendix A.4). In other words, LLLR optimizes classification speed and accuracy at the same time. The SPRT-TANDEM can classify non-i.i.d. data series with user-defined model complexity by changing $N (\in \mathbb{N})$, the order of approximation, to define the number of past samples on which the given sample depends. By dynamically changing the number of samples used for classification, the SPRT-TANDEM can maintain high classification accuracy while minimizing the sample size as much as possible. Moreover, the SPRT-TANDEM enables a user to flexibly control the speed-accuracy tradeoff without additional training, making it applicable to various practical applications.

We test the SPRT-TANDEM on our new database, Nosaic MNIST (NMNIST), in addition to the publicly available UCF101 action recognition database (Soomro et al., 2012) and Spoofing in the Wild (SiW) database (Liu et al., 2018). Two-way analysis of variance (ANOVA, (Fisher, 1925)) followed by a Tukey-Kramer multi-comparison test (Tukey, 1949; Kramer, 1956) shows that our proposed SPRT-TANDEM provides statistically significantly higher accuracy than other fixed-length and variable-length classifiers at a smaller number of data samples, making Wald's SPRT applicable even to non-i.i.d. data series. Our contribution is fivefold:

1. We invented a deep neural network-based algorithm, SPRT-TANDEM, which enables Wald's SPRT on arbitrary sequential data without knowing the true LLR.

2. The SPRT-TANDEM extends the SPRT to non-i.i.d. data series without knowing the true LLR.

3. With a novel loss, LLLR, the SPRT-TANDEM sequentially estimates LLR to optimize speed and accuracy simultaneously.

4. The SPRT-TANDEM can control the speed-accuracy tradeoff without additional training.

5. We introduce Nosaic MNIST, a novel early-classification database.

## 2 RELATED WORK

The SPRT-TANDEM has multiple interdisciplinary intersections with other fields of research: Wald's classical SPRT, probability density estimation, neurophysiological decision making, and time-series

classification. The comprehensive review is left to Appendix B, while in the following, we introduce the SPRT, probability density estimation algorithms, and early classification of the time series.

**Sequential Probability Ratio Test (SPRT).** The SPRT, denoted by $\delta^*$, is defined as the tuple of a decision rule and a stopping rule (Tartakovsky et al., 2014; Wald, 1947):

**Definition 2.1. Sequential Probability Ratio Test (SPRT).** *Let $\lambda_t$ as the LLR at time $t$, and $X^{(1,T)}$ as a sequential data $X^{(1,T)} := \{x^{(t)}\}_{t=1}^T$. Given the absolute values of lower and upper decision threshold, $a_0 \geq 0$ and $a_1 \geq 0$, SPRT, $\delta^*$, is defined as*

$$\delta^* = (d^*, \tau^*), \tag{1}$$

*where the decision rule $d^*$ and stopping time $\tau^*$ are*

$$d^*(X^{(1,T)}) = \begin{cases} 1 & \text{if } \lambda_{\tau^*} \geq a_1 \\ 0 & \text{if } \lambda_{\tau^*} \leq -a_0 \,, \end{cases} \tag{2}$$

$$\tau^* = \inf\{T \geq 0 | \lambda_T \notin (-a_0, a_1)\}\,. \tag{3}$$

We review the proof of optimality in Appendix A.4, while Figure 1 shows an intuitive explanation.

**Probability density ratio estimation.** Instead of estimating numerator and denominator of a density ratio separately, the probability density ratio estimation algorithms estimate the ratio as a whole, reducing the degree of freedom for more precise estimation (Sugiyama et al., 2010; 2012). Two of the probability density ratio estimation algorithms that closely related to our work are the probabilistic classification (Bickel et al., 2007; Cheng & Chu, 2004; Qin, 1998) and density fitting approach (Sugiyama et al., 2008; Tsuboi et al., 2009) algorithms. As we show in Section 4 and Appendix E, the SPRT-TANDEM sequentially estimates the LLR by combining the two algorithms.

**Early classification of time series.** To make decision time as short as possible, algorithms for early classification of time series can handle variable length of data (Mori et al., 2018; Mori et al., 2016; Xing et al., 2009; 2012) to minimize high sampling costs (e.g., medical diagnostics (Evans et al., 2015; Griffin & Moorman, 2001), or stock crisis identification (Ghalwash et al., 2014)). Leveraging deep neural networks is no exception in the early classification of time series (Dennis et al., 2018; Suzuki et al., 2018). Long short-term memory (LSTM) variants LSTM-s/LSTM-m impose monotonicity on classification score and inter-class margin, respectively, to speed up action detection (Ma et al., 2016). Early and Adaptive Recurrent Label ESTimator (EARLIEST) combines reinforcement learning and a recurrent neural network to decide when to classify and assign a class label (Hartvigsen et al., 2019).

## 3 PROPOSED ALGORITHM: SPRT-TANDEM

In this section, we propose the TANDEM formula, which provides the $N$-th order approximation of the LLR with respect to posterior probabilities. The i.i.d. assumption of Wald's SPRT greatly simplifies the LLR calculation at the expense of the precise temporal relationship between data samples. On the other hand, incorporating a long correlation among multiple data may improve the LLR estimation; however, calculating too long a correlation may potentially be detrimental in the following cases. First, if a class signature is significantly shorter than the correlation length in consideration, uninformative data samples are included in calculating LLR, resulting in a late or wrong decision (Campos et al., 2018). Second, long correlations require calculating a long-range of backpropagation, prone to vanishing gradient problem (Hochreiter et al., 2001). Thus, we relax the i.i.d. assumption by keeping only up to the $N$-th order correlation to calculate the LLR.

**The TANDEM formula.** Here, we introduce the TANDEM formula, which computes the approximated LLR, the decision value of the SPRT-TANDEM algorithm. The data series is approximated as an $N$-th order Markov process. For the complete derivation of the 0th (i.i.d.), 1st, and $N$-th order TANDEM formula, see Appendix C. Given a maximum timestamp $T \in \mathbb{N}$, let $X^{(1,T)}$ and $y$ be a sequential data $X^{(1,T)} := \{x^{(t)}\}_{t=1}^T$ and a class label $y \in \{1, 0\}$, respectively, where $x^{(t)} \in \mathbb{R}^{d_x}$ and $d_x \in \mathbb{N}$. By using Bayes' rule with the $N$-th order Markov assumption, the joint LLR of data at a

timestamp $t$ is written as follows:

$$
\begin{aligned}
\log & \left( \frac{p(x^{(1)}, x^{(2)}, ..., x^{(t)}|y=1)}{p(x^{(1)}, x^{(2)}, ..., x^{(t)}|y=0)} \right) \\
= & \sum_{s=N+1}^{t} \log \left( \frac{p(y=1|x^{(s-N)}, ..., x^{(s)})}{p(y=0|x^{(s-N)}, ..., x^{(s)})} \right) - \sum_{s=N+2}^{t} \log \left( \frac{p(y=1|x^{(s-N)}, ..., x^{(s-1)})}{p(y=0|x^{(s-N)}, ..., x^{(s-1)})} \right) \\
& - \log \left( \frac{p(y=1)}{p(y=0)} \right)
\end{aligned}
\tag{4}
$$

(see Equation (84) and (85) in Appendix C for the full formula). Hereafter we use terms *k-let* or *multiplet* to indicate the posterior probabilities, $p(y|x^{(1)}, ..., x^{(k)}) = p(y|X^{(1,k)})$ that consider correlation across $k$ data points. The first two terms of the TANDEM formula (Equation (4)), $N+1$-let and $N$-let, have the opposite signs working in "tandem" adjusting each other to compute the LLR. The third term is a prior (bias) term. In the experiment, we assume a flat prior or zero bias term, but a user may impose a non-flat prior to handling the biased distribution of a dataset. The TANDEM formula can be interpreted as a realization of the probability matching approach of the probability density estimation, under an $N$-th order Markov assumption of data series.

**Neural network that calculates the SPRT-TANDEM formula.** The SPRT-TANDEM is designed to explicitly calculate the $N$-th order TANDEM formula to realize sequential density ratio estimation, which is the critical difference between our SPRT-TANDEM network and other architecture based on convolutional neural networks (CNNs) and recurrent neural networks (RNN). Figure 2 illustrates a conceptual diagram explaining a generalized neural network structure, in accordance with the 1st-order TANDEM formula for simplicity. The network consists of a feature extractor and a temporal integrator (highlighted by red and blue boxes, respectively). They are arbitrary networks that a user can choose depending on classification problems or available computational resources. The feature extractor and temporal integrator are separately trained because we find that this achieves better performance than the end-to-end approach (also see Appendix D). The feature extractor outputs single-frame features (e.g., outputs from a global average pooling layer), which are the input vectors of the temporal integrator. The output vectors from the temporal integrator are transformed with a fully-connected layer into two-dimensional logits, which are then input to the softmax layer to obtain posterior probabilities. They are used to compute the LLR to run the SPRT (Equation (2)). Note that during the training phase of the feature extractor, the global average pooling layer is followed by a fully-connected layer for binary classification.

**How to choose the hyperparameter $N$?** By tuning the hyperparameter $N$, a user can efficiently boost the model performance depending on databases; in Section 5, we change $N$ to visualize the model performance as a function of $N$. Here, we provide two ways to choose $N$. One is to choose $N$ based on the *specific time scale,* a concept introduced in Appendix D, where we describe in detail how to guess on the best $N$ depending on databases. The other is to use a hyperparameter tuning algorithm, such as Optuna, (Akiba et al., 2019) to choose $N$ objectively. Optuna has multiple hyperparameter searching algorithms, the default of which is the Tree-structured Parzen Estimator (Bergstra et al., 2011). Note that tuning $N$ is not computationally expensive, because $N$ is only related to the temporal integrator, not the feature extractor. In fact, the temporal integrator's training speed is much faster than that of the feature extractor: 9 mins/epoch vs. 10 hrs/epoch ($N = 49$, NVIDIA RTX2080Ti, SiW database).

## 4 LLLR AND MULTIPLET CROSS-ENTROPY LOSS

Given a maximum timestamp $T \in \mathbb{N}$ and dataset size $M \in \mathbb{N}$, let $S := \{(X_i^{(1,T)}, y_i)\}_{i=1}^{M}$ be a sequential dataset. Training our network to calculate the TANDEM formula involves the following loss functions in combination: (i) the Loss for Log Likelihood Ratio estimation (LLLR), $L_{\text{LLR}}$, and (ii) multiplet cross-entropy loss, $L_{\text{multiplet}}$. The total loss, $L_{\text{total}}$ is defined as

$$
L_{\text{total}} = L_{\text{LLR}} + L_{\text{multiplet}}.
\tag{5}
$$

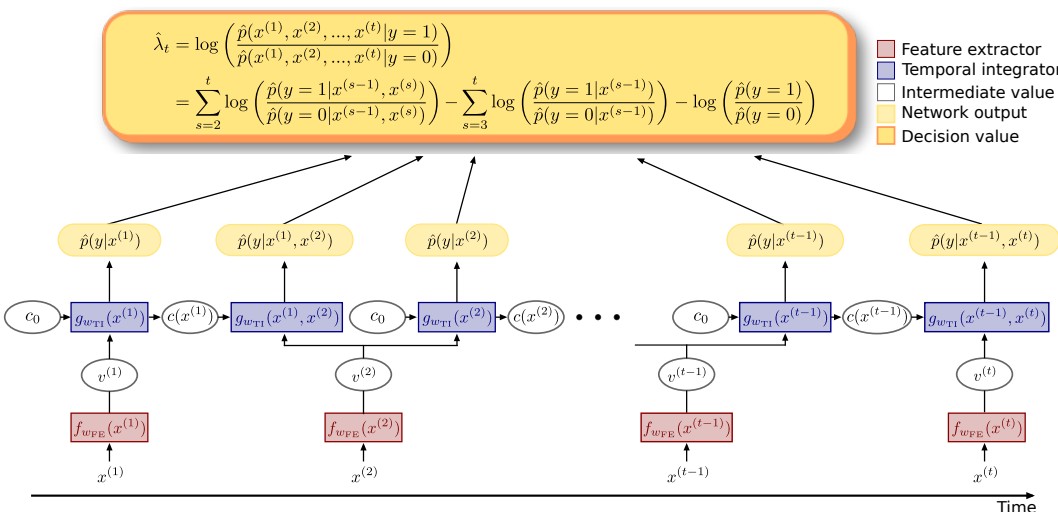

Figure 2: Conceptual diagram of neural network for the SPRT-TANDEM where the order of approximation $N = 1$. The feature extractor (red) extracts the feature vector for classification and outputs it to the temporal integrator (blue). Note that the temporal integrator memorizes up to $N$ preceding states in order to calculate the TANDEM formula (Equation (4)). LLR is calculated using the estimated probability densities that are output from the temporal integrator. We use $\hat{\cdot}$ to highlight a quantity estimated by a neural network. Trainable weight parameters are shared across the boxes with the same color in the figure.

## 4.1 Loss for Log-Likelihood Ratio estimation (LLLR).

The SPRT is Bayes-optimal as long as the true LLR is available; however, the true LLR is often inaccessible under real-world scenarios. To empirically estimate the LLR with the TANDEM formula, we propose the LLLR

$$
L_{\mathrm{LLR}} = \frac{1}{MT} \sum_{i=1}^{M} \sum_{t=1}^{T} \left| y_i - \sigma\left( \log\left( \frac{\hat{p}(x_i^{(1)}, x_i^{(2)}, ..., x_i^{(t)}|y=1)}{\hat{p}(x_i^{(1)}, x_i^{(2)}, ..., x_i^{(t)}|y=0)} \right) \right) \right|,
\tag{6}
$$

where $\sigma$ is the sigmoid function. We use $\hat{p}$ to highlight a probability density estimated by a neural network. The LLLR minimizes the Kullback-Leibler divergence (Kullback & Leibler, 1951) between the estimated and the true densities, as we briefly discuss below. The full discussion is given in Appendix E due to page limit.

**Density fitting.** First, we introduce *KLIEP* (Kullback-Leibler Importance Estimation Procedure, Sugiyama et al. (2008)), a density fitting approach of the density ratio estimation Sugiyama et al. (2010). KLIEP is an optimization problem of the Kullback-Leibler divergence between $p(X|y=1)$ and $\hat{r}(X)p(X|y=0)$ with constraint conditions, where $X$ and $y$ are random variables corresponding to $X_i^{(1,t)}$ and $y_i$, and $\hat{r}(X) := \hat{p}(X|y=1)/\hat{p}(X|y=0)$ is the estimated density ratio. Formally,

$$
\underset{\hat{r}}{\operatorname{argmin}} \left[ \mathrm{KL}(p(X|y=1)||\hat{r}(X)p(X|y=0)) \right] = \underset{\hat{r}}{\operatorname{argmin}} \left[ -\int dX p(X|y=1) \log(\hat{r}(X)) \right]
\tag{7}
$$

with the constraints $0 \le \hat{r}(X)$ and $\int dX \hat{r}(X)p(X|y=0) = 1$. The first constraint ensures the positivity of the estimated density $\hat{r}(X)p(X|y=0)$, while the second one is the normalization condition. Applying the empirical approximation, we obtain the final optimization problem:

$$
\underset{\hat{r}}{\operatorname{argmin}} \left[ \frac{1}{M_1} \sum_{i \in I_1} -\log \hat{r}(X_i^{(1,t)}) \right], \quad \text{with} \quad \hat{r}(X_i^{(1,t)}) \ge 0 \quad \text{and} \quad \frac{1}{M_0} \sum_{i \in I_0} \hat{r}(X_i^{(1,t)}) = 1,
\tag{8}
$$

where $I_1 := \{i \in [M]|y_i = 1\}$, $I_0 := \{i \in [M]|y_i = 0\}$, $M_1 := |I_1|$, and $M_0 := |I_0|$.

**Stabilization.** The original KLIEP (8), however, is asymmetric with respect to $p(X|y=1)$ and $p(X|y=0)$. To recover the symmetry, we add $\frac{1}{M_0} \sum_{i \in I_0} -\log(\hat{r}(X_i^{(1,t)})^{-1})$ to the objective and impose an additional constraint $\frac{1}{M_1} \sum_{i \in I_1} \hat{r}(X_i^{(1,t)})^{-1} = 1$. Besides, the symmetrized objective

still has unbounded gradients, which cause instability in the training. Therefore, we normalize the LLRs with the sigmoid function, obtaining the LLLR (6). We can also show that the constraints are effectively satisfied due to the sigmoid funciton. See Appendix E for the details.

In summary, we have shown that *the LLLR minimizes the Kullback-Leibler divergence of the true and the estimated density and further stabilizes the training by restricting the value of LLR.* Here we emphasize the contributions of the LLLR again. The LLLR enables us to conduct the stable LLR estimation and thus to perform the SPRT, the algorithm optimizing two objectives: stopping time and accuracy. In previous works (Mori et al., 2018; Hartvigsen et al., 2020), on the other hand, these two objectives are achieved with separate loss sub-functions.

Compared to KLIEP, the proposed LLLR statistically significantly boosts the performance of the SPRT-TANDEM (Appendix E.4). Besides, experiment on multivariate Gaussian with a simple toy-model also shows that the LLLR minimize errors between the estimated and the true density ratio (Appendix F).

## 4.2 MULTIPLET CROSS-ENTROPY LOSS.

To further facilitate training the neural network, we add binary cross-entropy losses, though the LLLR suffices to estimate LLR. We call them multiplet cross-entropy loss here, and defined as:

$$L_{\text{multiplet}} := \sum_{k=1}^{N+1} L_{k\text{-let}} \, , \tag{9}$$

where

$$L_{k\text{-let}} := \frac{1}{M(T-N)} \sum_{i=1}^{M} \sum_{t=k}^{T-(N+1-k)} \left( -\log \hat{p}(y_i | x_i^{(t-k+1)}, ..., x_i^{(t)}) \right) \, . \tag{10}$$

Minimizing the multiplet cross-entropy loss is equivalent to minimizing the Kullback-Leibler divergence of the estimated posterior $k$-let $\hat{p}(y_i | x_i^{(t-k+1)}, ..., x_i^{(t)})$ and the true posterior $p(y_i | x_i^{(t-k+1)}, ..., x_i^{(t)})$ (shown in Appendix G), which is a consistent objective with the LLLR and thus the multiplet loss accelerates the training. Note also that the multiplet loss optimizes all the logits output from the temporal integrator, unlike the LLLR.

## 5 EXPERIMENTS AND RESULTS

In the following experiments, we use two quantities as evaluation criteria: (i) balanced accuracy, the arithmetic mean of the true positive and true negative rates, and (ii) mean hitting time, the average number of data samples used for classification. Note that the balanced accuracy is robust to class imbalance (Luque et al., 2019), and is equal to accuracy on balanced datasets.

Evaluated public databases are NMNIST, UCF, and SiW. Training, validation, and test datasets are split and fixed throughout the experiment. We selected three early-classification models (LSTM-s (Ma et al., 2016), LSTM-m (Ma et al., 2016), and EARLIEST (Hartvigsen et al., 2019)) and one fixed-length classifier (3DResNet (Hara et al., 2017)), as baseline models. All the early-classification models share the same feature extractor as that of the SPRT-TANDEM for a fair comparison.

Hyperparameters of all the models are optimized with Optuna unless otherwise noted so that no models are disadvantaged by choice of hyperparameters. See Appendix H for the search spaces and fixed final parameters. After fixing hyperparameters, experiments are repeated with different random seeds to obtain statistics. In each of the training runs, we evaluate the validation set after each training epoch and then save the weight parameters if the balanced accuracy on the validation set updates the largest value. The last saved weights are used as the model of that run. The model evaluation is performed on the test dataset.

During the test stage of the SPRT-TANDEM, we used various values of the SPRT thresholds to obtain a range of balanced accuracy-mean hitting time combinations to plot a speed-accuracy tradeoff (SAT) curve. If all the samples in a video are used up, the thresholds are collapsed to $a_1 = a_0 = 0$ to force a decision.

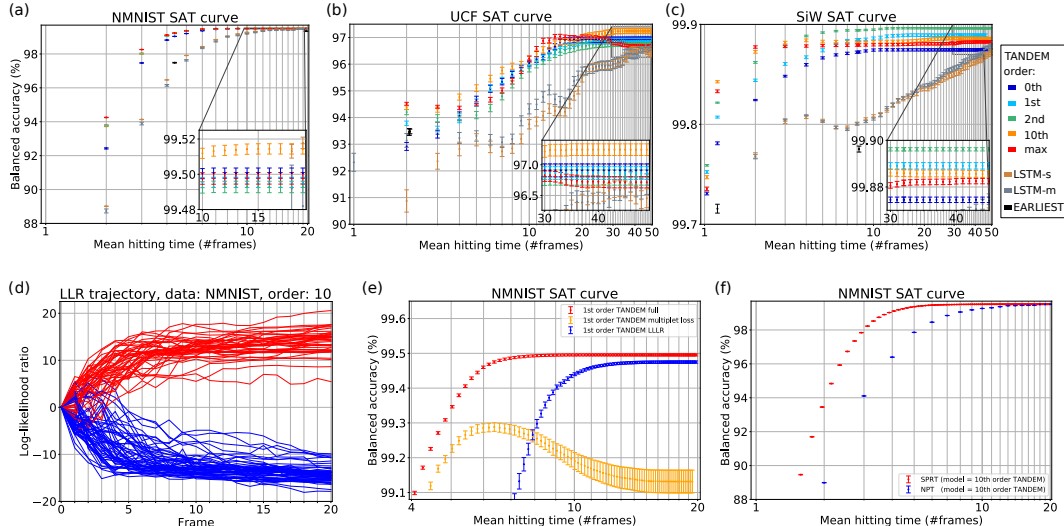

Figure 3: Experimental results. (a-c) Speed-accuracy tradeoff (SAT) curves for three databases: NMNIST, UCF, and SiW. Note that only representative results are shown. Error bars show the standard error of the mean (SEM). (d) Example LLR trajectories calculated on the NMNIST database with the 10th-order SPRT-TANDEM. Red and blue trajectories represent odd and even digits, respectively. (e) SAT curves of the ablation test comparing the effect of the $L_{\mathrm{multiplet}}$ and the $L_{\mathrm{LLR}}$. (f) SAT curves comparing the SPRT and Neyman-Pearson test (NPT) using the same 1st-order SPRT-TANDEM network trained on the NMNIST database.

To objectively compare all the models with various trial numbers, we conducted the two-way ANOVA followed by the Tukey-Kramer multi-comparison test to compute statistical significance. For the details of the statistical test, see Appendix I.

We show our experimental results below. Due to space limitations, we can only show representative results. For the full details, see Appendix J. For our computing infrastructure, see Appendix K.

**Nosaic MNIST (Noise + mosaic MNIST) database.** We introduce a novel dataset, NMNIST, whose video is buried with noise at the first frame, and gradually denoised toward the last, 20th frame (see Appendix L for example data). The motivation to create NMNIST instead of using a preexisting time-series database is as follows: for simple video databases such as Moving MNIST (MMNIST, (Srivastava et al., 2015)), each data sample contains too much information so that well-trained classifiers can correctly classify a video only with one or two frames (see Appendix M for the results of the SPRT-TANDEM and LSTM-m on MMNIST).

We design a parity classification task, classifying $0 - 9$ digits into an odd or even class. The training, validation, and test datasets contain 50,000, 10,000, and 10,000 videos with frames of size $28 \times 28 \times 1$ (gray scale). Each pixel value is divided by $127.5$, before subtracted by $1$. The feature extractor of the SPRT-TANDEM is ResNet-110 (He et al., 2016a), with the final output reduced to 128 channels. The temporal integrator is a peephole-LSTM (Gers & Schmidhuber, 2000; Hochreiter & Schmidhuber, 1997), with hidden layers of 128 units. The total numbers of trainable parameters on the feature extractor and temporal integrator are 6.9M and 0.1M, respectively. We train 0th, 1st, 2nd, 3rd, 4th, 5th, 10th, and 19th order SPRT-TANDEM networks. LSTM-s / LSTM-m and EARLIEST use peephole-LSTM and LSTM, respectively, both with hidden layers of 128 units. 3DResNet has 101 layers with 128 final output channels so that the total number of trainable parameters is in the same order (7.7M) as that of the SPRT-TANDEM.

Figure 3a and Table 1 shows representative results of the experiment. Figure 3d shows example LLR trajectories calculated with the 10th order SPRT-TANDEM. The SPRT-TANDEM outperforms other baseline algorithms by large margins at all mean hitting times. The best performing model is the 10th order TANDEM, which achieves statistically significantly higher balanced accuracy than the other algorithms ($p$-value $< 0.001$). Is the proposed algorithm's superiority because the SPRT-TANDEM successfully estimates the true LLR to approach asymptotic Bayes optimality? We discuss potential interpretations of the experimental results in the Appendix D.

Table 1: Representative mean balanced accuracy (%) calculated on NMNIST. For the complete list including standard errors, see Appendix J.

| Model | | Mean hitting time | | | | | | | | | | #trials |
|---|---|---|---|---|---|---|---|---|---|---|---|---|
| | | 2 | 3 | 4 | 4.37 | 5 | 6 | 10 | 15 | 19 | 19.66 | |
| SPRT-TANDEM (proposed) | 0th | 92.43 | 97.47 | 98.82 | 99.03 | 99.20 | 99.37 | 99.50 | 99.50 | 99.50 | 99.50 | 100 |
| | 1st | 93.81 | 98.04 | 99.07 | 99.21 | 99.34 | 99.46 | 99.50 | 99.50 | 99.50 | 99.50 | 100 |
| | 2nd | 93.73 | 98.01 | 99.07 | 99.22 | 99.36 | 99.45 | 99.49 | 99.49 | 99.49 | 99.50 | 120 |
| | 10th | 93.77 | 98.02 | 99.09 | **99.23** | **99.37** | **99.47** | **99.51** | **99.51** | **99.51** | **99.51** | 139 |
| | 19th (max) | **94.25** | **98.26** | **99.12** | **99.23** | **99.37** | 99.46 | 99.50 | 99.50 | 99.50 | 99.50 | 100 |
| LSTM-m | | 88.74 | 93.89 | 96.15 | | 97.62 | 98.35 | 99.19 | 99.42 | 99.48 | | 138 |
| LSTM-s | | 89.01 | 94.13 | 96.47 | | 97.91 | 98.43 | 99.28 | 99.45 | 99.52 | | 120 |
| EARLIEST | | | | | 97.48 | | | | | | 99.34 | 130 |
| 3DResNet | | | | | | 93.81 | | 96.98 | | | | 100 |

**UCF101 action recognition database.** To create a more challenging task, we selected two classes, handstand-pushups and handstand-walking, from the 101 classes in the UCF database. At a glimpse of one frame, the two classes are hard to distinguish. Thus, to correctly classify these classes, temporal information must be properly used. We resize each video's duration as multiples of 50 frames and sample every 50 frames with 25 frames of stride as one data. Training, validation, and test datasets contain 1026, 106, and 105 videos with frames of size $224 \times 224 \times 3$, randomly cropped to $200 \times 200 \times 3$ at training. The mean and variance of a frame are normalized to zero and one, respectively. The feature extractor of the SPRT-TANDEM is ResNet-50 (He et al., 2016b), with the final output reduced to 64 channels. The temporal integrator is a peephole-LSTM, with hidden layers of 64 units. The total numbers of trainable parameters in the feature extractor and temporal integrator are 26K and 33K, respectively. We train 0th, 1st, 2nd, 3rd, 5th, 10th, 19th, 24th, and 49th-order SPRT-TANDEM. LSTM-s / LSTM-m and EARLIEST use peephole-LSTM and LSTM, respectively, both with hidden layers of 64 units. 3DResNet has 50 layers with 64 final output channels so that the total number of trainable parameters (52K) is on the same order as that of the SPRT-TANDEM.

Figure 3b and Table 2 shows representative results of the experiment. The best performing model is the 10th order TANDEM, which achieves statistically significantly higher balanced accuracy than other models ($p$-value $< 0.001$). The superiority of the higher-order TANDEM indicates that a classifier needs to integrate longer temporal information in order to distinguish the two classes (also see Appendix D).

Table 2: Representative mean balanced accuracy (%) calculated on UCF. For the complete list including standard errors, see Appendix J.

| Model | | Mean hitting time | | | | | | | | | | #trials |
|---|---|---|---|---|---|---|---|---|---|---|---|---|
| | | 2 | 2.01 | 2.09 | 3 | 4 | 5 | 10 | 15 | 25 | 49 | |
| SPRT-TANDEM (proposed) | 0th | 92.92 | 92.94 | 93.00 | 93.38 | 94.06 | 94.66 | 96.04 | 96.83 | 96.91 | 96.91 | 200 |
| | 1st | 93.79 | 93.78 | 93.73 | 93.57 | 93.93 | 94.56 | 95.96 | 96.55 | 96.87 | 96.87 | 200 |
| | 2nd | 94.20 | 94.20 | 94.18 | 93.97 | 94.01 | 94.09 | 95.84 | 96.46 | 96.76 | 96.79 | 200 |
| | 10th | 94.37 | 94.37 | 94.31 | 94.29 | **94.77** | **95.10** | 96.18 | 96.85 | **97.12** | **97.25** | 256 |
| | 49th (max) | **94.52** | **94.51** | **94.52** | **94.40** | 94.36 | 94.51 | **96.20** | **97.03** | 96.96 | 96.72 | 200 |
| LSTM-m | | 93.14 | | | 93.59 | 93.23 | 93.31 | 94.32 | 94.59 | 95.93 | 96.68 | 100 |
| LSTM-s | | 90.87 | | | 92.36 | 92.82 | 93.17 | 93.75 | 94.23 | 95.93 | 96.45 | 101 |
| EARLIEST | | | 93.38 | 93.48 | | | | | | | | 50 |
| 3DResNet | | | | | | | | | 64.42 | 90.08 | | 100 |

**Spoofing in the Wild (SiW) database.** To test the SPRT-TANDEM in a more practical situation, we conducted experiments on the SiW database. We use a sliding window of 50 frames-length and 25 frames-stride to sample data, which yields training, validation, and test datasets of 46,729, 4,968, and 43,878 videos of live or spoofing face. Each frame is resized to $256 \times 256 \times 3$ pixels and randomly cropped to $244 \times 244 \times 3$ at training. The mean and variance of a frame are normalized to zero and one, respectively. The feature extractor of the SPRT-TANDEM is ResNet-152, with the final output reduced to 512 channels. The temporal integrator is a peephole-LSTM, with hidden layers of 512 units. The total number of trainable parameters in the feature extractor and temporal integrator is 3.7M and 2.1M, respectively. We train 0th, 1st, 2nd, 3rd, 5th, 10th, 19th, 24th, and 49th-order SPRT-TANDEM networks. LSTM-s / LSTM-m and EARLIEST use peephole-LSTM and LSTM, respectively, both with hidden layers of 512 units. 3DResNet has 101 layers with 512 final output channels so that the total number of trainable parameters (5.3M) is in the same order as that of the SPRT-TANDEM. Optuna is not applied due to the large database and network size.

Figure 3c and Table 3 shows representative results of the experiment. The best performing model is the 10th order TANDEM, which achieves statistically significantly higher balanced accuracy than other models ($p$-value $< 0.001$). The superiority of the lower-order TANDEM indicates that each video frame contains a high amount of information necessary for the classification, imposing less need to collect a large number of frames (also see Appendix D).

Table 3: Representative mean balanced accuracy (%) calculated on SiW. For the complete list including standard errors, see Appendix J.

| Model | | Mean hitting time | | | | | | | | | | #trials |
| | | 1.19 | 2 | 3 | 5 | 8.21 | 10 | 15 | 25 | 32.06 | 49 | |
|---|---|---|---|---|---|---|---|---|---|---|---|---|
| SPRT-TANDEM (proposed) | 0th | 99.78 | 99.82 | 99.85 | 99.87 | 99.87 | 99.87 | 99.87 | 99.87 | 99.87 | 99.87 | 100 |
| | 1st | 99.81 | 99.84 | 99.86 | 99.87 | 99.88 | **99.89** | 99.89 | 99.89 | 99.89 | 99.89 | 112 |
| | 2nd | 99.82 | 99.86 | **99.88** | **99.89** | **99.89** | **99.89** | **99.90** | **99.90** | **99.90** | **99.90** | 110 |
| | 10th | **99.84** | 99.87 | **99.88** | 99.88 | 99.88 | 99.88 | 99.88 | 99.89 | 99.88 | 99.88 | 107 |
| | 49th (max) | 99.83 | **99.88** | **99.88** | 99.88 | 99.88 | 99.88 | 99.88 | 99.88 | 99.88 | 96.72 | 73 |
| LSTM-m | | | 99.77 | 99.80 | 99.80 | | 99.81 | 99.83 | 99.85 | | 99.88 | 63 |
| LSTM-s | | | 99.77 | 99.80 | 99.80 | | 99.81 | 99.83 | 99.84 | | 99.87 | 58 |
| EARLIEST | | 99.72 | | | | 99.77 | | | | 99.76 | | 30 |
| 3DResNet | | | | | 98.82 | | | 98.97 | 98.56 | | | 5 |

**Ablation study.** To understand contributions of the $L_{\mathrm{LLR}}$ and $L_{\mathrm{multiplet}}$ to the SAT curve, we conduct an ablation study. The 1st-order SPRT-TANDEM is trained with $L_{\mathrm{LLR}}$ only, $L_{\mathrm{multiplet}}$ only, and both $L_{\mathrm{LLR}}$ and $L_{\mathrm{multiplet}}$. The hyperparameters of the three models are independently optimized using Optuna (see Appendix H). The evaluated database and model are NMNIST and the 1st-order SPRT-TANDEM, respectively. Figure 3e shows the three SAT curves. The result shows that $L_{\mathrm{LLR}}$ leads to higher classification accuracy, whereas $L_{\mathrm{multiplet}}$ enables faster classification. The best performance is obtained by using both $L_{\mathrm{LLR}}$ and $L_{\mathrm{multiplet}}$. We also confirmed this tendency with the 19th order SPRT-TANDEM, as shown in Appendix N.

**SPRT vs. Neyman-Pearson test.** As we discuss in Appendix A, the Neyman-Person test is the optimal likelihood ratio test with a *fixed* number of samples. On the other hand, the SPRT takes a *flexible* number of samples for an earlier decisions. To experimentally test this prediction, we compare the SPRT-TANDEM and the corresponding Neyman-Pearson test. The Neyman-Pearson test classifies the entire data into two classes at each number of frames, using the estimated LLRs with threshold $\lambda = 0$. Results support the theoretical prediction, as shown in Figure 3f: the Neyman-Pearson test needs a larger number of samples than the SPRT-TANDEM.

## 6   CONCLUSION

We presented the SPRT-TANDEM, a novel algorithm making Wald's SPRT applicable to arbitrary data series without knowing the true LLR. Leveraging deep neural networks and the novel loss function, LLLR, the SPRT-TANDEM minimizes the distance of the true LLR and the LLR sequentially estimated with the TANDEM formula, enabling simultaneous optimization of speed and accuracy. Tested on the three publicly available databases, the SPRT-TANDEM achieves statistically significantly higher accuracy over other existing algorithms with a smaller number of data points. The SPRT-TANDEM enables a user to control the speed-accuracy tradeoff without additional training, opening up various potential applications where either high-accuracy or high-speed is required.

## ACKNOWLEDGEMENTS

The authors thank anonymous reviewers for their careful reading to improve the manuscript. We would also like to thank Hirofumi Nakayama and Yuka Fujii for insightful discussions. Special thanks to Yuka for naming the proposed algorithm.

## AUTHOR CONTRIBUTIONS

A.F.E. conceived the study. A.F.E. and T.M. constructed the theory, conducted the experiments, and wrote the paper. T. M. organized python codes to be ready for the release. K.S. and H.I. supervised the study.

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

CONTENTS

APPENDIX

## A  THEORETICAL ASPECTS OF THE SEQUENTIAL PROBABILITY RATIO TEST

In this section, we review the mathematical background of the SPRT following the discussion in Tartakovsky et al. (2014). First, we define the SPRT based on the measure theory and introduce Stein's lemma, which assures the termination of the SPRT. To define the optimality of the SPRT, we introduce two performance metrics that measure the false alarm rate and the expected stopping time, and discuss their tradeoff — the SPRT solves it. Through this analysis, we utilize two important approximations, the asymptotic approximation, and the no-overshoot approximation, which play essential roles to simplify our analysis. The asymptotic approximation assumes the upper and lower thresholds are infinitely far away from the origin, being equivalent to making the most careful decision to reduce the error rate, at the expense of the stopping time. On the other hand, the no-overshoot approximation assumes that we can neglect the threshold overshoots of the likelihood ratio.

Next, we show the superiority of the SPRT to the Neyman-Pearson test, using a simple Gaussian model. The Neyman-Pearson test is known to be optimal in the two-hypothesis testing problem and is often compared with the SPRT. Finally, we introduce several types of optimal conditions of the SPRT.

### A.1  PRELIMINARIES

**Notations.**  Let $(\Omega, \mathcal{F}, P)$ be a probability space; $\Omega$ is a sample space, $\mathcal{F} \subset \mathcal{P}^\Omega$ is a sigma-algebra of $\Omega$, where $\mathcal{P}^A$ denotes the power set of a set $A$, and $P$ is a probability measure. Intuitively, $\Omega$ represents the set of all the elementary events under consideration, e.g., $\Omega = \{$all the possible elementary events such that "a human is walking through a gate."$\}$. $\mathcal{F}$ is defined as a set of the subsets of $\Omega$, and stands for all the possible combinations of the elementary events; e.g., $\mathcal{F} \ni \{$ "Akinori is walking through the gate at the speed of 80 m/min," "Taiki is walking through the gate at the speed of 77 m/min," or "Nothing happened."$\}$ $P : \mathcal{F} \to [0,1]$ is a probability measure, a function that is normalized and countably additive; i.e., $P$ measures the probability that the event $A \in \mathcal{F}$ occurs. A random variable $X$ is defined as the measurable function from $\Omega$ to a measurable space, practically $\mathbb{R}^d$ ($d \in \mathbb{N}$); e.g., if $\omega (\in \Omega)$ is "Taiki is walking through the gate with a big smile," then $X(\omega)$ may be 100 frames of the color images with 128×128 pixels ($d = 128 \times 128 \times 3 \times 100$), i.e., a video recorded with a camera attached at the top of the gate. The probability that a random variable $X$ takes a set of values $S \in \mathbb{R}^d$ is defined as $P(X \in S) := P(X^{-1}(S))$, where $X^{-1}$ is the preimage of $X$. By definition of the measurable function, $X^{-1}(S) \in \mathcal{F}$ for all $S \in \mathbb{R}^d$. Let $\{\mathcal{F}_t\}_{t \geq 0}$ be a filtration. By definition, $\{\mathcal{F}_t\}_{t \geq 0}$ is a non-decreasing sequence of sub-sigma-algebras of $\mathcal{F}$; i.e., $\mathcal{F}_s \subset \mathcal{F}_t \subset \mathcal{F}$ for all $s$ and $t$ such that $0 < s < t$. Each element of filtration can be interpreted as the available information at a given point $t$. $(\Omega, \mathcal{F}, \{\mathcal{F}_t\}_{t \geq 0}, P)$ is called a filtered probability space.

As in the main manuscript, let $X^{(1,T)} := \{x^{(t)}\}_{t=1}^T$ be a sequential data point sampled from the density $p$, where $T \in \mathbb{N} \cup \{\infty\}$. For each $t \in [T]$, $x^{(t)} \in \mathbb{R}^{d_x}$, where $d_x \in \mathbb{N}$ is the dimensionality of the input data. In the i.i.d. case, $p(X^{(1,T)}) = \prod_{t=1}^T f(x^{(t)})$, where $f$ is the density of $x^{(1)}$. For each time-series data $X^{(1,T)}$, the associated label $y$ takes the value 1 or 0; we focus on the binary classification, or equivalently the two-hypothesis testing throughout this paper. When $y$ is a class label, $p(X^{(1,T)}|\theta)$ is the likelihood density function. Note that $X^{(1,T)}$ with label $y$ is sampled according to density $p(X^{(1,T)}|y)$.

Our goal is, given a sequence $X^{(1,T)}$, to identify which one of the two densities $p_1$ or $p_0$ the sequence $X^{(1,T)}$ is sampled from; formally, to test two hypotheses $H_1 : y = 1$ and $H_0 : y = 0$ given $X^{(1,T)}$. The decision function or test of a stochastic process $X^{(1,T)}$ is denoted by $d(X^{(1,T)}) : \Omega \to \{1,0\}$. We can identify this definition with, for each realization of $X^{(1,T)}$, $d : \mathbb{R}^{d_x \times T} \to \{1,0\}$, i.e., $X^{(1,T)} \mapsto y$, where $y \in \{1,0\}$. Thus we write $d$ instead of $d(X^{(1,T)})$, for simplicity. The stopping time of $X^{(1,T)}$ with respect to a filtration $\{\mathcal{F}_t\}_{t \geq 0}$ is defined as $\tau := \tau(X^{(1,T)}) : \Omega \to \mathbb{R}_{\geq 0}$ such that $\{\omega \in \Omega | \tau(\omega) \leq t\} \in \mathcal{F}_t$. Accordingly, for fixed $T \in \mathbb{N} \cup \{\infty\}$ and $y \in \{1,0\}$, $\{d = y\}$ means the set of time-series data such that the decision function accepts the hypothesis $H_i$ with a finite stopping time; more specifically, $\{d = y\} = \{\omega \in \Omega | d(X^{(1,T)})(\omega) = y, \tau(X^{(1,T)})(\omega) < \infty\}$. The decision rule $\delta$ is defined as the doublet $(d, \tau)$. Let $\Lambda_T := \Lambda(X^{(1,T)}) := \frac{p(X^{(1,T)}|y=1)}{p(X^{(1,T)}|y=0)}$ and

$\lambda_T := \log \Lambda_T$ be the likelihood ratio and the log-likelihood ratio of $X^{(1,T)}$. In the i.i.d. case, $\Lambda_T = \prod_{t=1}^{T} \frac{p(x^{(t)}|y=1)}{p(x^{(t)}|y=0)} = \prod_{t=1}^{T} Z^{(t)}$, where $p(X^{(1,T)}|y) = \prod_{t=1}^{T} p(x^{(t)}|y)$ $(y \in \{1,0\})$ and $Z^{(t)} := \frac{p(x^{(t)}|y=1)}{p(x^{(t)}|y=0)}$.

## A.2 DEFINITION AND THE TRADEOFF OF FALSE ALARMS AND STOPPING TIME

Let us overview the theoretical structure of the SPRT. In the following, we assume that the time-series data points are i.i.d. until otherwise stated.

**Definition of the SPRT.** The sequential probability ratio test (SPRT), denoted by $\delta^*$ is defined as the doublet of the decision function and the stopping time.

**Definition A.1. Sequential probability ratio test (SPRT)**
*Let $a_0 = -\log A_0 \geq 0$ and $a_1 = \log A_1 \geq 0$ be (the absolute values of) a lower and an upper threshold respectively.*

$$\delta^* = (d^*, \tau^*), \tag{11}$$

$$d^*(X^{(1,T)}) = \begin{cases} 1 & \text{if } \lambda_{\tau^*} \geq a_1 \\ 0 & \text{if } \lambda_{\tau^*} \leq -a_0, \end{cases} \tag{12}$$

$$\tau^* = \inf\{T \geq 0 | \lambda_T \notin (-a_0, a_1)\}. \tag{13}$$

Note that $d^*$ and $\tau^*$ implicitly depend on a stochastic process $X^{(1,T)}$. In general, a doublet $\delta$ of a terminal decision function and a stopping time is called a decision rule or a hypothesis test.

**Termination.** The i.i.d.-SPRT terminates with probability one and all the moments of the stopping time are finite, provided that the two hypotheses are distinguishable:

**Lemma A.1. Stein's lemma**
*Let $(\Omega, \mathcal{F}, P)$ be a probability space and $\{Y^{(t)}\}_{t \geq 1}$ be a sequence of i.i.d. random variables under $P$. Define $\tau := \inf\{T \geq 1 | \sum_{t=1}^{T} Y^{(t)} \notin (-a_0, a_1)\}$. If $P(Y^{(1)}) \neq 1$, the stopping time $\tau$ is exponentially bounded; i.e., there exist constants $C > 0$ and $0 < \rho < 1$ such that $P(\tau > T) \leq C\rho^T$ for all $T \geq 1$. Therefore, $P(\tau < \infty) = 1$ and $\mathbb{E}[\tau^k] < \infty$ for all $k > 0$.*

**Two performance metrics.** Considering the two-hypothesis testing, we employ two kinds of performance metrics to evaluate the efficiency of decision rules from complementary points of view: the false alarm rate and the stopping time. The first kind of metrics is the operation characteristic, denoted by $\beta(\delta, y)$, and its related metrics. The operation characteristic is the probability of the decision being 0 when the true label is $y = y$; formally,

**Definition A.2. Operation characteristic**
*The operation characteristic is the probability of accepting the hypothesis $H_0$ as a function of $y$:*

$$\beta(\delta, y) := P(d = 0|y). \tag{14}$$

Using the operation characteristic, we can define four statistical measures based on the confusion matrix; namely, False Positive Rate (FPR), False Negative Rate (FNR), True Negative Rate (TNR), and True Positive Rate (TPR).

$$\text{FPR: } \alpha_0(\delta) := 1 - \beta(\delta, 0) = P(d = 1|y = 0) \tag{15}$$
$$\text{FNR: } \alpha_1(\delta) := \beta(\delta, 1) = P(d = 0|y = 1) \tag{16}$$
$$\text{TNR: } \beta(\delta, 0) = 1 - \alpha_0(\delta) = 1 - P(d = 1|y = 0) \tag{17}$$
$$\text{TPR: } 1 - \beta(\delta, 1) = 1 - \alpha_1(\delta) = 1 - P(d = 0|y = 1) \tag{18}$$

Note that balanced accuracy is denoted by $(1 + \beta(\delta, 0) - \beta(\delta, 1))/2$ according to this notation. The second kind of metrics is the mean hitting time, and is defined as the expected stopping time of the decision rule:

**Definition A.3. Mean hitting time** *The mean hitting time is the expected number of time-series data points that are necessary for testing a hypothesis when the true parameter value is $y$: $\mathbb{E}_y \tau = \int_\Omega \tau dP(\cdot|y)$. The mean hitting time is also referred to as the expected sample size of the average sample number.*

There is a tradeoff between the false alarm rate and the mean hitting time. For example, the quickness may be sacrificed, if we use a decision rule $\delta$ that makes careful decisions, i.e., with the false alarm rate $1 - \beta(\delta, 0)$ less than some small constant. On the other hand, if we use $\delta$ that makes quick decisions, then $\delta$ may make careless decisions, i.e., raise lots of false alarms because the amount of evidences is insufficient. At the end of this section, we show that the SPRT is optimal in the sense of this tradeoff.

**The tradeoff of false alarms and stopping times for both i.i.d. and non-i.i.d.** We formulate the tradeoff of the false alarm rate and the stopping time. We can derive the fundamental relation of the threshold to the operation characteristic in both i.i.d. and non-i.i.d. cases (Tartakovsky et al. (2014)):

$$\begin{cases} \alpha_1^* \leq e^{-a_0}(1 - \alpha_0^*) \\ \alpha_0^* \leq e^{-a_1}(1 - \alpha_1^*), \end{cases} \tag{19}$$

where we defined $\alpha_y^* := \alpha_y(\delta^*)$ ($y \in \{1, 0\}$). These inequalities essentially represent the tradeoff of the false alarm rate and the stopping time. For example, as the thresholds $a_y$ ($y \in \{1, 0\}$) increase, the false alarm rate and the false rejection rate decrease, as (19) suggests, but the stopping time is likely to be larger, because more observations are needed to accumulate log-likelihood ratios to hit the larger thresholds.

**The asymptotic approximation and the no-overshoot approximation.** Equation 19 is an example of the tradeoff of the false alarm rate and the stopping time; further, we can derive another example in terms of the mean hitting time. Before that, we introduce two types of approximations that simplify our analysis.

The first one is the no-overshoot approximation. It assumes to ignore the threshold overshoots of the log-likelihood ratio at the decision time. This approximation is valid when the log-likelihood ratio of a single frame is sufficiently small compared to the gap of the thresholds, at least around the decision time. On the other hand, the second one is the asymptotic approximation, which assumes $a_0, a_1 \to \infty$, being equivalent to sufficiently low false alarm rates and false rejection rates at the expense of the stopping time. These approximations drastically facilitate the theoretical analysis; in fact, the no-overshoot approximation alters (19) as follows (see Tartakovsky et al. (2014)):

$$\alpha_1^* \approx e^{-a_0}(1 - \alpha_0^*), \qquad \alpha_0^* \approx e^{-a_1}(1 - \alpha_1^*), \tag{20}$$

which is equivalent to

$$\alpha_0^* \approx \frac{e^{a_0} - 1}{e^{a_0 + a_1} - 1}, \qquad \alpha_1^* \approx \frac{e^{a_1} - 1}{e^{a_0 + a_1} - 1} \tag{21}$$

$$\iff -a_0 \approx \log\left(\frac{\alpha_1^*}{1 - \alpha_0^*}\right), \quad a_1 \approx \log\left(\frac{1 - \alpha_1^*}{\alpha_0^*}\right) \tag{22}$$

$$\iff \beta^*(0) \approx \frac{e^{a_1} - 1}{e^{a_1} - e^{-a_0}}, \quad \beta^*(1) \approx \frac{e^{-a_1} - 1}{e^{-a_1} - e^{a_0}}, \tag{23}$$

where $\beta^*(y) := \beta(\delta^*, y)$ ($y \in \{1, 0\}$). Further assuming the asymptotic approximation, we obtain

$$\alpha_0^* \approx e^{-a_1}, \quad \alpha_1^* \approx e^{-a_0}. \tag{24}$$

Therefore, as the threshold gap increases, the false alarm rate and the false rejection rate decrease exponentially, while the decision making becomes slow, as is shown in the following.

**Mean hitting time without overshoots.** Let $I_y := \mathbb{E}_y[Z^{(1)}]$ ($y \in \{1, 0\}$) be the Kullback-Leibler divergence of $f_1$ and $f_0$. $I_y$ is larger if the two densities are more distinguishable. Note that $I_y \gtrless 0$ since $P_y(Z^{(1)} = 0) \lneq 1$, and thus the mean hitting times of the SPRT without overshoots are

expressed as

$$\mathbb{E}_1[\tau^*] = \frac{1}{I_1} \left[ (1 - \alpha_1^*) \log(\frac{1 - \alpha_1^*}{\alpha_0^*}) - \alpha_1^* \log(\frac{1 - \alpha_0^*}{\alpha_1^*}) \right], \tag{25}$$

$$\mathbb{E}_0[\tau^*] = \frac{1}{I_0} \left[ (1 - \alpha_0^*) \log(\frac{1 - \alpha_0^*}{\alpha_1^*}) - \alpha_0^* \log(\frac{1 - \alpha_1^*}{\alpha_0^*}) \right] \tag{26}$$

In Tartakovsky et al. (2014). Introducing the function

$$\gamma(x, y) := (1 - x) \log(\frac{1 - x}{y}) - x \log(\frac{1 - y}{x}), \tag{27}$$

we can simplify (25-26):

$$\mathbb{E}_1[\tau^*] = \frac{1}{I_1} \gamma(\alpha_1^*, \alpha_0^*) \tag{28}$$

$$\mathbb{E}_0[\tau^*] = \frac{1}{I_1} \gamma(\alpha_0^*, \alpha_1^*). \tag{29}$$

(25-26) shows the tradeoff as we mentioned above: the mean hitting time of positive (negative) data diverges if we are to set the false alarm (rejection) rate to be zero.

**The tradeoff with overshoots.** Introducing the overshoots explicitly, we can obtain the equality, instead of the inequality such as (19), that connects the the error rates and the thresholds. We first define the overshoots of the thresholds $a_0$ and $a_1$ at the stopping time as

$$\kappa_1(a_0, a_1) := \lambda_{\tau^*} - a_1 \qquad \text{on}\{\lambda_{\tau*} \geq a_1\} \tag{30}$$
$$\kappa_0(a_0, a_1) := -(\lambda_{\tau^*} + a_0) \qquad \text{on}\{\lambda_{\tau^*} \leq -a_0\}. \tag{31}$$

We further define the expectation of the exponentiated overshoots as

$$e_1(a_0, a_1) := \mathbb{E}_1[e^{-\kappa_1(a_0, a_1)} | \lambda_{\tau^*} \geq a_1] \tag{32}$$

$$e_0(a_0, a_1) := \mathbb{E}_0[e^{-\kappa_0(a_0, a_1)} | \lambda_{\tau^*} \leq -a_0]. \tag{33}$$

Then we can relate the thresholds to the error rates (without the no-overshoots approximation, Tartakovsky (1991)):

$$\alpha_0^* = \frac{e_1(a_0, a_1) e^{a_0} - e_1(a_0, a_1) e_0(a_0, a_1)}{e^{a_1 + a_0} - e_1(a_0, a_1) e_0(a_0, a_1)}, \quad \alpha_1^* = \frac{e_0(a_0, a_1) e^{a_1} - e_1(a_0, a_1) e_0(a_0, a_1)}{e^{a_1 + a_0} - e_1(a_0, a_1) e_0(a_0, a_1)}. \tag{34}$$

To obtain more specific dependence on the thresholds $a_y$ ($y \in \{1, 0\}$), we adopt the asymptotic approximation. Let $T_0(a_0)$ and $T_1(a_1)$ be the one-sided stopping times, i.e., $T_0(a_0) := \inf\{T \geq 1 | \lambda_T \leq -a_0\}$ and $T_1(a_1) := \inf\{T \geq 1 | \lambda_T \geq a_1\}$. We then define the associated overshoots as

$$\tilde{\kappa}_1(a_1) := \lambda_{T_1} - a_1 \quad \text{on}\{T_1 < \infty\}, \tag{35}$$
$$\tilde{\kappa}_0(a_0) := -(\lambda_{T_0} + a_0) \quad \text{on}\{T_0 < \infty\}. \tag{36}$$

According to Lotov (1988), we can show that

$$\alpha_0^* \approx \frac{\zeta_1 e^{a_0} - \zeta_1 \zeta_0}{e^{a_0 + a_1} - \zeta_1 \zeta_0}, \quad \alpha_1^* \approx \frac{\zeta_0 e^{a_1} - \zeta_1 \zeta_0}{e^{a_0 + a_1} - \zeta_1 \zeta_0} \tag{37}$$

under the asymptotic approximation. Note that

$$\zeta_y := \lim_{a_y \to \infty} \mathbb{E}_y[e^{-\tilde{\kappa}_y}] \quad (y \in \{1, 0\}) \tag{38}$$

have no dependence on the thresholds $a_y$ ($y \in \{1, 0\}$). Therefore we have obtained more precise dependence of the error rates on the thresholds than (24):

**Theorem A.1. The Asymptotic tradeoff with overshoots** *Assume that $0 < I_y < \infty$ ($y \in \{1, 0\}$). Let $\zeta_y$ be given in (38). Then*

$$\alpha_0^* = \zeta_1 e^{-a_1}(1 + o(1)), \quad \alpha_1^* = \zeta_0 e^{-a_0}(1 + o(1)) \quad (a_0, a_1 \longrightarrow \infty). \tag{39}$$

**Mean hitting time with overshoots.** A more general form of the mean hitting time is provided in Tartakovsky (1991). We can show that

$$\mathbb{E}_1 \tau^* = \frac{1}{I_1} \left[ \left(1 - \alpha_1^*\right)\left(a_1 + \mathbb{E}_1[\kappa_1 | \tau^* = T]\right) - \alpha_1^*\left(a_0 + \mathbb{E}_1[\kappa_0 | \tau^* = T_0]\right) \right] \qquad (40)$$

$$\mathbb{E}_0 \tau^* = \frac{1}{I_0} \left[ \left(1 - \alpha_0^*\right)\left(a_0 + \mathbb{E}_0[\kappa_0 | \tau^* = T]\right) - \alpha_0^*\left(a_1 + \mathbb{E}_0[\kappa_1 | \tau^* = T_1]\right) \right]. \qquad (41)$$

The mean hitting times (40-41) explicitly depend on the overshoots, compared with (25-26). Let

$$\chi_y := \lim_{a_t h \to \infty} \mathbb{E}_y[\tilde{\kappa}_y] \quad (y \in \{1, 0\}) \qquad (42)$$

be the limiting average overshoots in the one-sided tests. Note that $\chi_y$ have no dependence on $a_y$ ($y \in \{1, 0\}$). The asymptotic mean hitting times with overshoots are

$$\mathbb{E}_1 \tau^* = \frac{1}{I_1}(a_1 + \chi_1) + o(1), \quad \mathbb{E}_0 \tau^* = \frac{1}{I_0}(a_0 + \chi_0) + o(1) \quad (a_0 e^{-a_1} \to 0, \quad a_1 e^{-a_0} \to 0) \qquad (43)$$

As expressed in Tartakovsky et al. (2014). Therefore, they have an asymptotically linear dependence on the thresholds.

## A.3 THE NEYMAN-PEARSON TEST AND THE SPRT

So far, we have discussed the tradeoff of the false alarm rate and the mean hitting time and several properties of the operation characteristic and the mean hitting time. Next, we compare the SPRT with the Neyman-Pearson test, which is well-known to be optimal in the *classification* of time-series with *fixed* sample lengths; in contrast, the SPRT is optimal in the *early classification* of time-series with *indefinite* sample lengths, as we show in the next section.

We show that the Neyman-Pearson test is optimal in the two-hypothesis testing problem or the binary classification of time-series. Nevertheless, we show that in the i.i.d. Gaussian model, the SPRT terminates earlier than the Neyman-Pearson test despite the same error rates.

**Preliminaries.** Before defining the Neyman-Pearson test, we specify what the "best" test should be. There are three criteria, namely the most powerful test, Bayes test, and minimax test. To explain them in detail, we have to define the size and the power of the test. The significance level, or simply the size of test $d$ is defined as[1]

$$\alpha := P(d = 1 | y = 0). \qquad (44)$$

It is also known as the false positive rate, the false alarm rate, or the false acceptance rate of the test. On the other hand, the power of the test $d$ is given by

$$\gamma := 1 - \beta := P(d = 1 | y = 1). \qquad (45)$$

$\gamma$ is also called the true positive rate, the true acceptance rate. the recall, or the sensitivity. $\beta$ is known as the false negative rate or the false rejection rate.

Now, we can define the three criteria mentioned above.

**Definition A.4. Most powerful test**
*The most powerful test $d$ of significance level $\alpha(> 0)$ is defined as the test that for every other test $d'$ of significance level $\alpha$, the power of $d$ is greater than or equal to that of $d'$:*

$$P(d = 1 | y = 1) \geq P(d' = 1 | y = 1). \qquad (46)$$

**Definition A.5. Bayes test**
*Let $\pi_0 := P(y = 0)$ and $\pi_1 := P(y = 1) = 1 - \pi_0$ be the prior probabilities of hypotheses $H_0$ and $H_1$, and $\bar{\alpha}(d)$ be the average probability of error:*

$$\bar{\alpha}(d) := \sum_{i=1,0} \pi_i \alpha_i(d), \qquad (47)$$

---

[1] $P(d = 1 | y = 0)$ is short for $P(\{\omega \in \Omega | d(X^{(1,T)})(\omega) = 1\} | y = 0)$ and is equivalent to $\mathbb{P}_{X^{(1,T)} \sim p(X^{(1,T)} | y=1)}[d(X^{(1,T)}) = 1]$ (i.e., the probability of the decision being 1, where $X^{(1,T)}$ is sampled from the density $p(X^{(1,T)} | y = 1)$ ).

where $\alpha_i(d) := P(d \neq i | y = i)$ *is the false negative rate of the class* $i \in \{1, 0\}$. *A Bayes test,* *denoted by* $d^{\mathrm{B}}$, *for the priors is defined as the test that minimizes the average probability of error:*

$$d^{\mathrm{B}} := \underset{d}{\mathrm{arginf}}\{\bar{\alpha}(d)\}\,, \tag{48}$$

*where the infimum is taken over all fixed-sample-size decision rules.*

**Definition A.6. Minimax test**
*Let* $\alpha_{\max}(d)$ *be the maximum error probability:*

$$\alpha_{\max}(d) := \max_{i \in \{1,0\}}\{\alpha_i(d)\}\,. \tag{49}$$

*A minimax test, denoted by* $d^{\mathrm{M}}$, *is defined as the test that minimizes the maximum error probability:*

$$\alpha_{\max}(d^{\mathrm{M}}) = \inf_{d}\{\alpha_{\max}(d)\}\,, \tag{50}$$

*where the infimum is taken over all fixed-sample-size tests.*

Note that a *fixed-sample-size decision rule* or *non-sequential rule* is the decision rule with a fixed stopping time $T = N$ with probability one.

**Definition and the optimality of the Neyman-Pearson test.** Based on the above notions, we state the definition and the optimality of the Neyman-Pearson test. We see the most powerful test for the two-hypothesis testing problem is the Neyman-Pearson test; the theorem below is also the definition of the Neyman-Pearson test.

**Theorem A.2. Neyman-Pearson lemma** *Consider the two-hypothesis testing problem, i.e., the problem of testing two hypotheses* $H_o : P = P_0$ *and* $H_1 : P_1$, *where* $P_0$ *and* $P_1$ *are two probability distributions with densities* $p_0$ *and* $p_1$ *with respect to some probability measure. The most powerful test is given by*

$$d^{\mathrm{NP}}(X^{(1,T)}) := \begin{cases} 1 & if\ \Lambda(X^{(1,T)}) \geq h(\alpha) \\ 0 & otherwise\,, \end{cases} \tag{51}$$

*where* $\Lambda(X^{(1,T)}) = \frac{p_1(X^{(1,T)})}{p_0(X^{(1,T)})}$ *is the likelihood ratio and the threshold* $h(\alpha)$ *is defined as*

$$\alpha_0(d^{\mathrm{NP}})\left(\equiv P(d^{\mathrm{NP}}(X^{(1,T)}) = 1|H_0) = \mathbb{E}_0[d^{\mathrm{NP}}(X^{(1,T)})]\right) = \alpha \tag{52}$$

*to ensure for the false positive rate to be the user-defined value* $\alpha(> 0)$.

$d^{\mathrm{NP}}$ is referred to as the Neyman-Pearson test and is also optimal with respect to the Bayes and minimax criteria:

**Theorem A.3. Neyman-Pearson test is Bayes optimal** *Consider the two-hypothesis testing problem. Given a prior distribution* $\pi_i$ *(*$i \in \{1, 0\}$*) the Bayes test* $d^{\mathrm{B}}$, *which minimizes the average error probability* $\bar{\alpha}(d) = \pi_0\alpha_0(d) + \pi_1\alpha_1(d)$, *is given by*

$$d^{\mathrm{B}}(X^{(1,T)}) = \begin{cases} 1 & (if\ \Lambda(X^{(1,T)}) \geq \pi_0/\pi_1) \\ 0 & (otherwise)\,. \end{cases} \tag{53}$$

*That is, the Bayesian test is given by the Neyman-Pearson test with the threshold* $\pi_0/\pi_1$.

**Theorem A.4. Neyman-Pearson test is minimax optimal** *Consider the two-hypothesis testing problem. the minimax test* $d^{\mathrm{M}}$, *which minimizes the maximal error probability* $\alpha_{\max}(d) = \max_{i \in \{1,0\}}\{\alpha_i(d)\}$, *is the Neyman-Pearson test with the threshold such that* $\alpha_0(d^{\mathrm{M}}) = \alpha_1(d^{\mathrm{M}})$.

The proofs are given in Borovkov (1998) and Lehmann & Romano (2006).

**The SPRT is more efficient.** We have shown that the Neyman-Pearson test is optimal in the two-hypothesis testing problem, in the sense that the Neyman-Pearson test is the most powerful, Bayes, and minimax test; nevertheless, we can show that the SPRT terminates faster than the Neyman-Pearson test even when these two show the same error rate.

Consider the two-hypothesis testing problem for the i.i.d. Gaussian model:

$$\begin{cases} H_i : y = y_i & (i \in \{1, 0\}) \\ x^{(t)} = y + \xi^{(t)} & (t \geq 1, y \in \mathbb{R}^1) \\ \xi^{(t)} \sim \mathcal{N}(0, \sigma^2) & (\sigma \geq 0), \end{cases} \tag{54}$$

where $\mathcal{N}(0, \sigma^2)$ denotes the Gaussian distribution with mean 0 and variance $\sigma^2$. The Neyman-Pearson test has the form

$$d^{\mathrm{NP}}(X^{(1,n(\alpha_0,\alpha_1))}) = \begin{cases} 1 & (\text{if } \lambda_{n(\alpha_0,\alpha_1)} \geq h(\alpha_0, \alpha_1)) \\ 0 & (\text{otherwise}). \end{cases} \tag{55}$$

The sequence length $n = n(\alpha_0, \alpha_1)$ and the threshold $h = h(\alpha_0, \alpha_1)$ are defined so as for the false positie rate and the false negative rate to be equal to $\alpha_0$ and $\alpha_1$ respectively; i.e.,

$$P(\lambda_n \geq h | y = y_0) = \alpha_0, \tag{56}$$

$$P(\lambda_n < h | y = y_1) = \alpha_1. \tag{57}$$

We can solve them for the i.i.d. Gaussian model (Tartakovsky et al. (2014)). To see the efficiency of the SPRT to the Neyman-Pearson test, we define

$$\mathcal{E}_0(\alpha_0, \alpha_1) = \frac{\mathbb{E}[\tau^* | y = y_0]}{n(\alpha_0, \alpha_1)} \tag{58}$$

$$\mathcal{E}_1(\alpha_0, \alpha_1) = \frac{\mathbb{E}[\tau * | y = y_1]}{n(\alpha_0, \alpha_1)}. \tag{59}$$

Assuming the overshoots are negligible, we obtain the following asymptotic efficiency (Tartakovsky et al. (2014)):

$$\lim_{\max\{\alpha_0, \alpha_1\} \to 0} \mathcal{E}_y(\alpha_0, \alpha_1) = \frac{1}{4} \quad (y \in \{1, 0\}). \tag{60}$$

In other words, under the no-overshoot and the asymptotic assumptions, the SPRT terminates four times earlier than the Neyman-Pearson test in expectation, despite the same false positive and negative rates.

### A.4 THE OPTIMALITY OF THE SPRT

**Optimality in i.i.d. cases.** The theorem below shows that the SPRT minimizes the expected hitting times in the class of decision rules that have bounded false positive and negative rates. Consider the two-hypothesis testing problem. We define the class of decision rules as

$$C(\alpha_0, \alpha_1) = \{\delta \quad \text{s.t.} \quad P(d = 1 | H_0) \leq \alpha_0, P(d = 0 | H_1) \leq \alpha_1, \mathbb{E}[\tau | H_0] < \infty, \mathbb{E}[\tau | H_1] < \infty\}. \tag{61}$$

Then the optimality theorem states:

**Theorem A.5. I.I.D. Optimality (Tartakovsky et al. (2014))** *Let the time-series data points $x^{(t)}$, $t = 1, 2, ...$ be i.i.d. with density $f_0$ under $H_0$ and with density $f_1$ under $H_1$, where $f_0 \not\equiv f_1$. Let $\alpha_0 > 0$ and $\alpha_1 > 0$ be fixed constants such that $\alpha_0 + \alpha_1 < 1$. If the thresholds $-a_o$ and $a_1$ satisfies $\alpha_0^*(a_0, a_1) = \alpha_0$ and $\alpha_1^*(a_0, a_1) = \alpha_1$, then the SPRT $\delta^* = (d^*, \tau^*)$ satisfies*

$$\inf_{\delta = (d,\tau) \in C(\alpha_0,\alpha_1)} \left\{ \mathbb{E}[\tau | H_0] \right\} = \mathbb{E}[\tau^* | H_0] \quad \text{and} \quad \inf_{\delta = (d,\tau) \in C(\alpha_0,\alpha_1)} \left\{ \mathbb{E}[\tau | H_1] \right\} = \mathbb{E}[\tau^* | H_1] \tag{62}$$

A similar optimality holds for continuous-time processes (Irle & Schmitz (1984)). Therefore the SPRT terminates at the earliest stopping time in expectation of any other decision rules achieving the same or less error rates — the SPRT is optimal.

Theorem A.5 tells us that given user-defined thresholds, the SPRT attains the optimal mean hitting time. Also, remember that the thresholds determine the error rates (e.g., Equation (24)). Therefore, the SPRT can minimize the required number of samples and achieve the desired upper-bounds of false positive and false negative rates.

**Asymptotic optimality in general non-i.i.d. cases.** In most of the discussion above, we have assumed the time-series samples are i.i.d. For general non-i.i.d. distributions, we have the asymptotic optimality; i.e., the SPRT asymptotically minimizes the moments of the stopping time distribution (Tartakovsky et al. (2014)).

Before stating the theorem, we first define a type of convergence of random variables.

**Definition A.7. r-quick convergence** *Let $\{x^{(t)}\}_{t\geq 1}$ be a stochastic process. Let $\mathcal{T}_\epsilon(\{x^{(t)}\}_{t\geq 1})$ be the last entry time of the stochastic process $\{x^{(t)}\}_{t\geq 1}$ in the region $(\epsilon, \infty) \cup (-\infty, -\epsilon)$, i.e.,*

$$\mathcal{T}_\epsilon(\{x^{(t)}\}_{t\geq 1}) = \sup_{t\geq 1}\{t \ \ s.t. \ \ |x^{(t)}| > \epsilon\}, \quad \sup\{\emptyset\} := 0. \tag{63}$$

*Then, we say that the stochastic process $\{x^{(t)}\}_{t\geq 1}$ converges to zero r-quickly, or*

$$x^{(t)} \xrightarrow[t\to\infty]{r-\text{quickly}} 0, \tag{64}$$

*for some $r > 0$, if*

$$\mathbb{E}[(\mathcal{T}_\epsilon(\{x^{(t)}\}_{t\geq 1}))^r] < \infty \quad \text{for every } \epsilon > 0. \tag{65}$$

$r$-quick convergence ensures that the last entry time in the large-deviation region ($\mathcal{T}_\epsilon(\{x^{(t)}\}_{t\geq 1})$) is finite almost surely. The asymptotic optimality theorem is:

**Theorem A.6. Non-i.i.d. asymptotic optimality** *If there exist positive constants $I_0$ and $I_1$ and an increasing non-negative function $\psi(t)$ such that*

$$\frac{\lambda_t}{\psi(t)} \xrightarrow[t\to\infty]{P_1-r-\text{quickly}} I_1 \quad \text{and} \quad \frac{\lambda_t}{\psi(t)} \xrightarrow[t\to\infty]{P_0-r-\text{quickly}} -I_0, \tag{66}$$

*where $\lambda_t$ is defined in section A.1, then*

$$\mathbb{E}[(\tau^*)^r|y = i] < \infty \quad (i \in \{1,0\}) \text{ for any finite } a_0 \text{ and } a_1. \tag{67}$$

*Moreover, if the thresholds $a_0$ and $a_1$ are chosen to satisfy (19), $a_0 \to \log(1/\alpha_1^*)$, and $a_1 \to \log(1/\alpha_0^*)$ ($a_i \to \infty$), then for all $0 < m \leq r$,*

$$\inf_{\delta\in C(\alpha_0,\alpha_1)} \{\mathbb{E}[\tau^m|y=y_1]\} - \mathbb{E}[(\tau^*)^m|y=y_1] \longrightarrow 0 \tag{68}$$

$$\inf_{\delta\in C(\alpha_0,\alpha_1)} \{\mathbb{E}[\tau^m|y=y_0]\} - \mathbb{E}[(\tau^*)^m|y=y_0] \longrightarrow 0 \tag{69}$$

*as $\max\{\alpha_0, \alpha_1\} \longrightarrow 0$ with $|\log\alpha_0/\log\alpha_1| \longrightarrow c$, where $c \in (0,\infty)$.*

## B  SUPPLEMENTARY REVIEW OF THE RELATED WORK

**Primate's decision making and parietal cortical neurons.**  The process of decision making involves multiple steps, such as evidence accumulation, reward prediction, risk evaluation, and action selection. We give a brief overview regarding mainly to neural activities of primate parietal lobe and their relationship to the evidence accumulation, instead of providing a comprehensive review of the decision making literature. Interested readers may refer to review articles, such as Doya (2008); Gallivan et al. (2018); Gold & Shadlen (2007).

In order to study neural correlates of decision making, Roitman & Shadlen (2002) used a random dot motion (RDM) task on non-human primates. They found that the neurons in the cortical area lateral intraparietal cortex, or LIP, gradually accumulated sensory evidence represented as increasing firing rate, toward one of the two thresholds corresponding to the two-alternative choices. Moreover, while a steeper increase of firing rates leads to an early decision of the animal, the final firing rates at the decision time is almost constant regardless of reaction time. Thus, at least in population-level LIP neurons are representing information very similar to that of the LLR in the SPRT algorithm (It is under active discussion whether the ramping activity is seen only in averaged population firing rate or both in population and single neuron level. See Latimer et al. (2015); Shadlen et al. (2016)). But also see Okazawa et al. (2021) for a recent finding that evidence accumulation is represented in a high-dimensional manifold of neural population. In any case, LIP neurons seem to represent accumulated evidence as their activity patterns.

To test whether the ramping activity is explained by Wald's SPRT, Kira et al. (2015) used visual stimuli associated with reward likelihood: each stimulus indicates the answer of the binary choice task with a certain probability (e.g., if stimulus 'A' is presented, choice 1 is the correct answer with $30\%$ probability). LIP neurons' activities in response to these randomly presented stimuli are proportional to LLR calculated from the associated likelihood of the stimuli, letting authors concluded that the activity of LIP neurons are best explained by SPRT than other alternative models. It remains unclear, however, what algorithm is used in the brain when stimuli are not randomly presented but temporary dependent.

More complex decision making involving risk evaluation such as "delayed, large reward V.S. immediate, small reward" is thought to be guided by other regions including orbitofrontal cortex, dorsal striatum or dorsal prefrontal cortex (McClure et al. (2004); Rudebeck et al. (2006); Tanaka et al. (2004)).

**Application of SPRT.**  Ever since Wald's formulation, the sequential hypothesis testing was applied to study decision making and its reaction time (Stone (1960); Edwards (1965); Ashby (1983)). Several extensions to more general problem settings were also proposed. In order to test more than two hypotheses, multi-hypothesis SPRT (MSPRT) was introduced (Armitage (1950); Baum & Veeravalli (1994)), and shown to be asymptotically optimal (Dragalin et al. (1999; 2000); Veeravalli & Baum (1995)). The SPRT was also generalized for non-i.i.d. data (Lai (1981); Tartakovsky (1999)), and theoretically shown to be asymptotically optimal, given the known LLR (Dragalin et al. (1999; 2000)). Tartakovsky et al. (2014) provided a comprehensive review of these theoretical analyses, a part of whose reasoning we also follow to show optimality in Appendix A. The SPRT, and closely related, generalized LLR test, applied to solve several problems includes drug safety surveillance (Kulldorff et al. (2011)), exoplanet detection (Hu et al. (2019)), and the LLR test out of weak classifiers (WaldBoost, Sochman & Matas (2005)), to name a few. On an A/B test, Johari et al. (2017) tackled an important problem of inflating error rates at the sequential hypothesis testing. Ju et al. (2019) proposed an inputed Girshick test to determine a better variant.

**Time-series classification.**  Here, we use the term "Time-series" interchangeably to mention both continuous data or discrete data such as video frames.

One of the traditional approaches to univariate or multivariate time series classification is distance-based methods, such as dynamic time warping (Bagnall (2014); Jeong et al. (2011); Kate (2015)) or k-nearest neighbors (Dau et al. (2018); Wei & Keogh (2006); Yang & Shahabi (2007)). More recently, Collective Of Transformation-based Ensembles (COTE) and its variant, COTE with Hierarchical Vote system (HIVE-COTE) showed high classification performance at the expense of their high computational cost (Bagnall et al. (2015); Lines et al. (2016)). Word Extraction for time series

classification (WEASEL) and its variant, WEASEL+MUSE take a bag-of-pattern approach to utilize carefully designed feature vectors (Schäfer & Leser (2017)).

The advent of deep learning allows researchers to classify not only univariate/multivariate data, but also large-size, video data using convolutional neural networks (Hara et al. (2017); Carreira & Zisserman (2017); Karim et al. (2018); Wang et al. (2017)). Thanks to the increasing computation power and memory of modern processing units, each video data in a minibatch are designed to be sufficiently long in the time domain such that class signature can be contained. Video length of the training, validation, and test data are often assumed to be fixed; however, ensuring sufficient length for all data may compromise the classification speed (i.e., number of samples that used for classification). We extensively test this issue in Section 5.

## C   DERIVATION OF THE TANDEM FORMULA

The derivations of the important formulas in Section 3 are provided below.

**The 0th order (i.i.d.) TANDEM formula.**   We use the following probability ratio to identify if the input sequence $\{x^{(s)}\}_{s=1}^t$ is derived from either hypothesis $H_1 : y = 1$ or $H_0 : y = 0$.

$$\frac{p(x^{(1)}, ..., x^{(t)}|y = 1)}{p(x^{(1)}, ..., x^{(t)}|y = 0)} . \tag{70}$$

We can rewrite it with the posterior. First, by repeatedly using the Bayes rule, we obtain

$$
\begin{aligned}
&p(x^{(1)}, x^{(2)}, ..., x^{(t)}|y) \\
&= p(x^{(t)}|x^{(t-1)}, x^{(t-2)}, ..., x^{(1)}, y)p(x^{(t-1)}, x^{(t-2)}, ..., x^{(1)}|y) \\
&= p(x^{(t)}|x^{(t-1)}, x^{(t-2)}, ..., x^{(1)}, y) \\
&\times p(x^{(t-1)}|x^{(t-2)}, x^{(t-3)}, ..., x^{(1)}, y)p(x^{(t-2)}, x^{(t-3)}, ..., x^{(1)}, y) \\
&= ... \\
&\vdots \\
&= p(x^{(t)}|x^{(t-1)}, x^{(t-2)}, ..., x^{(1)}, y)p(x^{(t-1)}|x^{(t-2)}, x^{(t-3)}, ..., x^{(1)}, y) \ldots p(x^{(2)}|x^{(1)}, y) . 
\end{aligned}
\tag{71}
$$

We use this formula hereafter. Let us assume that the process $\{x^{(s)}\}_{s=1}^t$ is conditionally-independently and identically distributed (hereafter simply noted as i.i.d.), namely

$$p(x^{(1)}, x^{(2)}, ..., x^{(t)}|y) = \prod_{s=1}^t p(x^{(s)}|y) , \tag{72}$$

which yields the following LLR representation ("0-th order Markov process"):

$$p(x^{(t)}|x^{(t-1)}, x^{(t-2)}, ..., x^{(1)}, y) = p(x^{(t)}|y) . \tag{73}$$

Then

$$
\begin{aligned}
&p(x^{(1)}, x^{(2)}, ..., x^{(t)}|y) \\
&= p(x^{(t)}|y)p(x^{(t-1)}|y) \ldots p(x^{(2)}|y)p(x^{(1)}|y) \\
&= \prod_{s=1}^t \left[ p(x^{(s)}|y) \right] \\
&= \prod_{s=1}^t \left[ \frac{p(y|x^{(s)})p(x^{(s)})}{p(y)} \right] .
\end{aligned}
$$

Hence

$$\frac{p(x^{(1)}, x^{(2)}, ..., x^{(t)}|y = 1)}{p(x^{(1)}, x^{(2)}, ..., x^{(t)}|y = 0)} = \prod_{s=1}^t \left[ \frac{p(y = 1|x^{(s)})}{p(y = 0|x^{(s)})} \right] \left( \frac{p(y = 0)}{p(y = 1)} \right)^t , \tag{74}$$

or

$$\log\left( \frac{p(x^{(1)}, x^{(2)}, ..., x^{(t)}|y = 1)}{p(x^{(1)}, x^{(2)}, ..., x^{(t)}|y = 0)} \right) = \sum_{s=1}^t \log\left( \frac{p(y = 1|x^{(s)})}{p(y = 0|x^{(s)})} \right) - t\log\left( \frac{p(y = 1)}{p(y = 0)} \right) . \tag{75}$$

**The 1st-order TANDEM formula.**   So far, we have utilized the i.i.d. assumption (73) or (72). Now let us derive the probability ratio of the *first-order Markov process*, which assumes

$$p(x^{(t)}|x^{(t-1)}, x^{(t-2)}, ..., x^{(1)}, y) = p(x^{(t)}|x^{(t-1)}, y) . \tag{76}$$

Applying (76) to (71), we obtain

$$p(x^{(1)}, x^{(2)}, ..., x^{(t)}|y)$$

$$= p(x^{(t)}|x^{(t-1)}, y)p(x^{(t-1)}|x^{(t-2)}, y)\ldots p(x^{(2)}|x^{(1)}, y)p(x^{(1)}|y)$$

$$= \prod_{s=2}^{t} \left[ p(x^{(s)}|x^{(s-1)}, y) \right] p(x^{(1)}|y)$$

$$= \prod_{s=2}^{t} \left[ \frac{p(y|x^{(s)}, x^{(s-1)})p(x^{(s)}, x^{(s-1)})}{p(x^{(s-1)}, y)} \right] \frac{p(y|x^{(1)})p(x^{(1)})}{p(y)}$$

$$= \prod_{s=2}^{t} \left[ \frac{p(y|x^{(s)}, x^{(s-1)})p(x^{(s)}, x^{(s-1)})}{p(y|x^{(s-1)})p(x^{(s-1)})} \right] \frac{p(y|x^{(1)})p(x^{(1)})}{p(y)}, \tag{77}$$

for $t \geq 2$. Hence

$$\frac{p(x^{(1)}, x^{(2)}, ..., x^{(t)}|y = 1)}{p(x^{(1)}, x^{(2)}, ..., x^{(t)}|y = 0)} =$$

$$\prod_{s=2}^{t} \left[ \frac{p(y = 1|x^{(s)}, x^{(s-1)})}{p(y = 0|x^{(s)}, x^{(s-1)})} \right] \prod_{s=3}^{t} \left[ \frac{p(y = 0|x^{(s-1)})}{p(y = 1|x^{(s-1)})} \right] \frac{p(y = 0)}{p(y = 1)}, \tag{78}$$

or

$$\log \left( \frac{p(x^{(1)}, x^{(2)}, ..., x^{(t)}|y = 1)}{p(x^{(1)}, x^{(2)}, ..., x^{(t)}|y = 0)} \right) =$$

$$\sum_{s=2}^{t} \log \left( \frac{p(y = 1|x^{(s)}, x^{(s-1)})}{p(y = 0|x^{(s)}, x^{(s-1)})} \right) - \sum_{s=3}^{t} \log \left( \frac{p(y = 1|x^{(s-1)})}{p(y = 0|x^{(s-1)})} \right) - \log \left( \frac{p(y = 1)}{p(y = 0)} \right). \tag{79}$$

For $t = 1$ and $t = 2$, the natural extensions are

$$\log \left( \frac{p(x^{(1)}|y = 1)}{p(x^{(1)}|y = 0)} \right) = \log \left( \frac{p(y = 1|x^{(1)})}{p(y = 0|x^{(1)})} \right) - \log \left( \frac{p(y = 1)}{p(y = 0)} \right)$$

$$\log \left( \frac{p(x^{(1)}, x^{(2)}|y = 1)}{p(x^{(1)}, x^{(2)}|y = 0)} \right) = \log \left( \frac{p(y = 1|x^{(1)}, x^{(2)})}{p(y = 0|x^{(1)}, x^{(2)})} \right) - \log \left( \frac{p(y = 1)}{p(y = 0)} \right). \tag{80}$$

**The $N$-th order TANDEM formula.** Finally we extend the 1st order TANDEM formula so that it can calculate the general $N$-th order log-likelihood ratio. The $N$-th order Markov process is defined as

$$p(x^{(t)}|x^{(t-1)}, x^{(t-2)}, ..., x^{(1)}, y) = p(x^{(t)}|x^{(t-1)}, ..., x^{(t-N)}, y). \tag{81}$$

Therefore, for $t \geq N + 2$

$$p(x^{(1)}, x^{(2)}, ..., x^{(t)}|y)$$

$$= p(x^{(t)}|x^{(t-1)}, ..., x^{(t-N)}, y)p(x^{(t-1)}|x^{(t-2)}, ..., x^{(t-N-1)}, y)\ldots p(x^{(2)}|x^{(1)}, y)p(x^{(1)}|y)$$

$$= \prod_{s=N+1}^{t} \left[ p(x^{(s)}|x^{(s-1)}, ..., x^{(s-N)}, y) \right] p(x^{(N)}, x^{(N-1)}, ..., x^{(1)}|y)$$

$$= \prod_{s=N+1}^{t} \left[ \frac{p(y|x^{(s)}, ..., x^{(s-N)})p(x^{(s)}, ..., x^{(s-N)})}{p(x^{(s-1)}, ..., x^{(s-N)}, y)} \right] \frac{p(y|x^{(N)}, ..., x^{(1)})p(x^{(N)}, ..., x^{(1)})}{p(y)}$$

$$= \prod_{s=N+1}^{t} \left[ \frac{p(y|x^{(s)}, ..., x^{(s-N)})p(x^{(s)}, ..., x^{(s-N)})}{p(y|x^{(s-1)}, ..., x^{(s-N)})p(x^{(s-1)}, ..., x^{(s-N)})} \right] \frac{p(y|x^{(N)}, ..., x^{(1)})p(x^{(N)}, ..., x^{(1)})}{p(y)}. \tag{82}$$

Hence

$$\frac{p(x^{(1)}, x^{(2)}, ..., x^{(t)}|y = 1)}{p(x^{(1)}, x^{(2)}, ..., x^{(t)}|y = 0)} =$$

$$\prod_{s=N+1}^{t} \left[ \frac{p(y = 1|x^{(s)}, ..., x^{(s-N)})}{p(y = 0|x^{(s)}, ..., x^{(s-N)})} \right] \prod_{s=N+2}^{t} \left[ \frac{p(y = 0|x^{(s-1)}, ..., x^{(s-N)})}{p(y = 1|x^{(s-1)}, ..., x^{(s-N)})} \right] \frac{p(y = 0)}{p(y = 1)}, \tag{83}$$

or

$$\log \left( \frac{p(x^{(1)}, x^{(2)}, ..., x^{(t)}|y = 1)}{p(x^{(1)}, x^{(2)}, ..., x^{(t)}|y = 0)} \right)$$
$$= \sum_{s=N+1}^{t} \log \left( \frac{p(y = 1|x^{(s)}, ..., x^{(s-N)})}{p(y = 0|x^{(s)}, ..., x^{(s-N)})} \right) \quad - \sum_{s=N+2}^{t} \log \left( \frac{p(y = 1|x^{(s-1)}, ..., x^{(s-N)})}{p(y = 0|x^{(s-1)}, ..., x^{(s-N)})} \right)$$
$$- \log \left( \frac{p(y = 1)}{p(y = 0)} \right) . \tag{84}$$

For $t < N + 2$, we obtain

$$\log \left( \frac{p(x^{(1)}, x^{(2)}, ..., x^{(t)}|y = 1)}{p(x^{(1)}, x^{(2)}, ..., x^{(t)}|y = 0)} \right)$$
$$= \log \left( \frac{p(y = 1|x^{(1)}, x^{(2)}, ..., x^{(t)})}{p(y = 0|x^{(1)}, x^{(2)}, ..., x^{(t)})} \right) - \log \left( \frac{p(y = 1)}{p(y = 0)} \right) . \tag{85}$$

# D    SUPPLEMENTARY DISCUSSION

**Why is the SPRT-TANDEM superior to other baselines?**    The potential drawbacks common to the LSTM-s/m and EARLIEST is that they incorporate long temporal correlation: it may lead to (1) the class signature length problem and (2) vanishing gradient problem, as we described in Section 3. (1) If a class signature is significantly shorter than the correlation length in consideration, uninformative data samples are included in calculating the log-likelihood ratio, resulting in a late or wrong decision. (2) long correlations require calculating a long-range of backpropagation, prone to the vanishing gradient problem.

An LSTM-s/m-specific drawback is similar to that of Neyman-Pearson test, in the sense that it fixes the number of samples before performance evaluations. On the other hand, the SPRT, and the SPRT-TANDEM, classify various lengths of samples: thus, the SPRT-TANDEM can achieve a smaller sampling number with high accuracy on average. Another potential drawback of LSTM-s/m is that their loss function explicitly imposes monotonicity to the scores. While the monotonicity is advantageous for quick decisions, it may sacrifice flexibility: the LSTM-s/m can hardly change its mind during a classification.

EARLIEST, the reinforcement-learning based classifier, decides on the various length of samples. A potential EARLIEST-specific drawback is that deep reinforcement learning is known to be unstable (Nikishin et al. (2018); Kumar et al.).

**How optimal is the SPRT-TANDEM?**    In practice, it is difficult to strictly satisfy the necessary conditions for the SPRT's optimality (Theorem A.5 and A.6) because of experimental limitations.

One of our primary interests is to apply the SPRT, the provably optimal algorithm, to real-world datasets. A major concern about extending Wald's SPRT is that we need to know the true likelihood ratio a priori to implement the SPRT. Thus we propose the SPRT-TANDEM with the help of machine learning and density ratio estimation to remove the concern, if not completely. However, some technical limitations still exist. Let us introduce two properties of the SPRT that can prevent the SPRT-TANDEM from approaching exact optimality.

Firstly, the SPRT is assumed to terminate for all the LLR trajectories under consideration with probability one. The corresponding equation stating this assumption is Equation (61) and (66) under the i.i.d. and non-i.i.d. condition, respectively. Given that this assumption (and the other minor technical conditions in Theorem A.5 and A.6) is satisfied, *the more precisely we estimate the LLRs, the more we approach the genuine SPRT implementation and thus its asymptotic Bayes optimality.*

Secondly, the non-i.i.d. SPRT is asymptotically optimal when the maximum number of samples allowed is not fixed (*infinite horizon*). On the other hand, our experiment truncates the SPRT (*finite horizon*) at the maximum timestamp, which depends on the datasets. Under the truncation, gradually collapsing thresholds are proven to give the optimal stopping (Tartakovsky et al. (2014)); however, the collapsing thresholds are obtained via backward induction (Bingham et al. (2006)), which is possible only after observing the *full* sequence. Thus, under the truncation, finding the optimal solutions in a strict sense critically limits practical applicability.

The truncation is just an experimental requirement and is not an essential assumption for the SPRT-TANDEM. Under the infinite horizon settings, the LLRs is assumed to increase or decrease toward the thresholds (Theorem A.6) in order to ensure the asymptotic optimality. However, we observed that the estimated LLRs tend to be asymptotically flat, especially when $N$ is large (Figure 10, 11, and 12); the estimated LLRs can violate the assumption of Theorem A.6.

One potential reason for the flat LLRs is the TANDEM formula: the first and second term of the formula has a different sign. Thus, the resulting log-likelihood ratio will be updated only when the difference between the two terms are non-zero. Because the first and second term depends on $N + 1$ and $N$ inputs, respectively, it is expected that the contribution of one input becomes relatively small as $N$ is enlarged. We are aware of this issue and already started working on it as future work.

Nevertheless, the flat LLRs at least do not spoil the practical efficiency of the SPRT-TANDEM, as our experiment shows. In fact, because we cannot know the true LLR of real-world datasets, it is not easy to discuss whether the assumption of the increasing LLRs is valid on the three databases (NMNIST,

UCF, and SiW) we tested. Numerical simulation may be possible, but it is out of our scope because our primary interest is to implement a practically usable SPRT under real-world scenarios.

**The best order $N$ of the SPRT-TANDEM.** The order $N$ is a hyperparamer, as we mentioned in Section 3 and thus needs to be tuned to attain the best performance. However, each dataset has its own temporal structure, and thus it is challenging to acquire the best order a priori. In the following, we provide a rough estimation of the best order, which may give dramatic benefit to the users and may lead to exciting future works.

Let us introduce a concept, *specific time scale*, which is used in physics to analyze qualitative behavior of a physical system. Here, we define the specific time scale of a physical system as a temporal interval in which the physical system develops dramatically. For example, suppose that a physical system under consideration is a small segment of spacetime in which an unstable particle, ortho-positronium (o-Ps), exists. In this case, a specific time scale can be defined as the lifetime of o-Ps, $0.14\mu s$ (Czarnecki (1999)), because the o-Ps is likely to vanish in $0.14 \times O(1)\mu s$ — the physical system has changed completely. Note that the definition of specific time scale is not unique for one physical system; it depends on the phenomena the researcher focuses on. Specific (time) scale are often found in fundamental equations that describes physical systems. In the example above, the decay equation $N(t) = A \exp(-t/\tau)$ has the lifetime $\tau \in \mathbb{R}$ in itself. Here $N(t) \in \mathbb{R}$ is the expected number of o-Ps' at time $t \in \mathbb{R}$, and $A \in \mathbb{R}$ is a constant.

Let us borrow the concept of the specific time scale to estimate the best order of the SPRT-TANDEM before training neural networks, though there is a gap in scale. In this case, we define the specific time scale of a dataset as the number of frames after which a typical video in the dataset shows completely different scene. As is discussed below, we claim that *the specific time scale of a dataset is a good estimation of the best order of the SPRT-TANDEM*, because the correlations shorter than the specific time scale are insufficient to distinguish each class, while the longer correlations may be contaminated with noise and keep redundant information.

First, we consider Nosaic MNSIT (NMNIST). The specific time scale of NMNIST can be defined as the half-life[2] of the noise, i.e., the necessary temporal interval for half of the noise to disappear. It is 10 frames by definition of NMNIST, and approximately matches the best order of the SPRT-TANDEM: in Figure 3, our experiment shows that the 10th order SPRT-TANDEM (with 11-frames correlation) outperforms the other orders in the latter timestamps, though we did not perform experiments with all the possible orders. A potential underlying mechanism is: Too long correlations keep noisy information in earlier timestamps, causing degradation, while too short correlations do not fully utilize the past information.

Next, we discuss the two classes in UCF101 action recognition database, handstand pushups and handstand walking, which are used in our experiment. A specific time scale is $\sim 10$ frames because of the following reasons. The first class, handstand pushups, has a specific time scale of one cycle of raising and lowering one's body $\sim 50$ frames (according to the shortest video in the class). The second class, handstand walking, has a specific time scale of one cycle of walking, i.e., two steps, $\sim 10$ frames (according to the longest video in the class). Therefore the specific time scale of UCF is $\sim 10$, the smaller one, since we can see whether there is a class signature in a video within at most $\sim 10$ frames. The specific time scale matches the best order of the SPRT-TANDEM according to Figure 3.

Finally, a specific time scale of SiW is $\sim 1$ frame, because a single image suffices to distinguish a real person and a spoofing image, because of the reflection of the display, texture of the photo, or the movement specific to a live person[3]. The best order in Figure 3 is $\sim 1$, matching the specific time scale.

We make comments on two potential future works related to estimation of the best order of the SPRT-TANDEM. First, as our experiments include only short videos, it is an interesting future work to estimate the best order of the SPRT-TANDEM in super-long video classification, where gradient vanishing becomes a problem and likelihood estimation becomes more challenging. Second, it is an

---

[2]This choice of words is, strictly speaking, not correct, because the noise decay in NMNIST is *linear*; the definition of half-life in physics assumes the decay to be *exponential*.

[3]In fact, the feature extractor, which classified a single frame to two classes, showed fairly high accuracy in our experiment, without temporal information.

exciting future work to analyse the relation of the specific time scale to the best order when there are multiple time scales. For example, recall the discussion of locality above in this Appendix D: Applying the SPRT-TANDEM to a dataset with distributed class signatures is challenging. Distributed class signatures may have two specific time scales: e.g., one is the mean length of the signatures, and the other is the mean interval between the signatures.

**The best threshold $\lambda_{\tau^*}$ of the SPRT-TANDEM.**    In practice, a user can change the thresholds after deploying the SPRT-TANDEM algorithm once and control the speed-accuracy tradeoff. Computing the speed-accuracy-tradeoff curve is not expensive, and importantly computable without re-training. According to the speed-accuracy-tradeoff curve, a user can choose the desired accuracy and speed. Note that this flexible property is missing in most other deep neural networks: controlling speed usually means changing the network structures and training it all over again.

**End-to-end v.s. separate training.**    The design of SPRT-TANDEM does not hamper an end-to-end training of neural networks; the feature extractor and temporal integrator can be readily connected for thorough backpropagation calculation. However, in Section 5, we trained the feature integrator and temporal integrator separately: after training the feature integrator, its trainable parameters are fixed to start training the temporal integrator. We decided to train the two networks separately because we found that it achieves better balanced accuracy and mean hitting time. Originally we trained the network using NMNIST database with an end-to-end manner, but the accuracy was far lower than the result reported in Section 5. We observed the same phenomenon when we trained the SPRT-TANDEM on our private video database containing 1-channel infrared videos. These observations might indicate that while the separate training may lose necessary information for classification compared to the end-to-end approach, it helps the training of the temporal integrator by fixing information at each data point. It will be interesting to study if this is a common problem in early-classification algorithms and find the right balance between the end-to-end and separate training to benefit both approaches.

**Feedback to the field of neuroscience.**    Kira et al. (2015) experimentally showed that the SPRT could explain neural activities in the area LIP at the macaque parietal lobe. They randomly presented a sequence of visual objects with associated reward probability. A natural question arises from here: what if the presented sequence is not random, but a time-dependent visual sequence? Will the neural activity be explained by our SPRT-TANDEM, or will the neurons utilize a completely different algorithm? Our research provides one driving hypothesis to lead the neuroscience community to a deeper understanding of the brain's decision-making system.

**Usage of statistical tests.**    As of writing this manuscript, not all of the computer science papers use statistical tests to evaluate their experiments. However, in order to provide an objective comparison across proposed and existing models, running multiple validation trials with random seeds followed by a statistical test is helpful. Thus, the authors hope that our paper stimulates the field of computer science to utilize statistical tests more actively.

**Ethical concern.**    The proposed method, SPRT-TANDEM, is a general algorithm applicable to a broad range of serial data, such as auditory signals or video frames. Thus, any ethical concerns entirely depend on the application and training database, not on our algorithm *per se*. For example, if SPRT-TANDEM is applied to a face spoofing detection, using faces of people of one particular racial or ethnic group as training data may lead to a bias toward or against people of other groups. However, this bias is a concern in machine learning in general, not specific to the SPRT-TANDEM.

**Is the SPRT-TANDEM "too local"?**    In our experiments in Section 5, the SPRT-TANDEM with maximum correlation allowed (i.e., 19th, 49th, and 49th on NMNIST, UCF, and SiW databases, respectively) does not necessarily reach the highest accuracy with a larger number of frames. Instead, depending on the database, the lower order of approximation, such as 10th order TANDEM, outperforms the other orders. In the SiW database, this observation is especially prominent: the model records the highest balanced accuracy is the 2nd order SPRT-TANDEM. While this may indicate our TANDEM formula with the "dropping correlation" strategy works well as we expected, a remaining concern is the SPRT may integrate too local information. What if class signatures are far separated in time?

In such a case, the SPRT-TANDEM may fail to integrate the distributed class signatures for correct classification. On the other hand, the SPRT-TANDEM may be able to add the useful information of the class signatures to the LLR only when encountering the signatures (in other words, do not add non-zero values to the LLR without seeing class signatures). The SPRT-TANDEM may be able to skip unnecessary data points without modification, or with a modification similar to SkipRNN (Campos et al. (2018)), which actively achieve this goal: by learning unnecessary data point, the SkipRNN skips updating the internal state of RNN to attend just to informative data. Similarly, we can modify the SPRT-TANDEM so that it learns to skip updating LLR upon encountering uninformative data. It will be exciting future work, and the authors are looking forward to testing the SPRT-TANDEM on a challenging database with distributed class signatures.

**A more challenging dataset: Nosaic MNIST-Hard (NMNIST-H).** In the main text, we see the accuracy of the SPRT-TANDEM saturates within a few timestamps. Therefore, it is worth testing the models on a dataset that require more samples for reaching good performance. We create a more challenging dataset, *Nosaic MNIST-Hard*: The MNIST handwritten digits are buried with heavier noise than the orifinal NMNIST (only 10 pixels/frame are revealed, while it is 40 pixels/frame for the original NMNIST). The resulting speed-accuracy tradeoff curves below show that the SPRT-TANDEM outperforms LSTM-s/m more than the error-bar range, even on the more challenging dataset requiring more timestamps to attain the accuracy saturation.

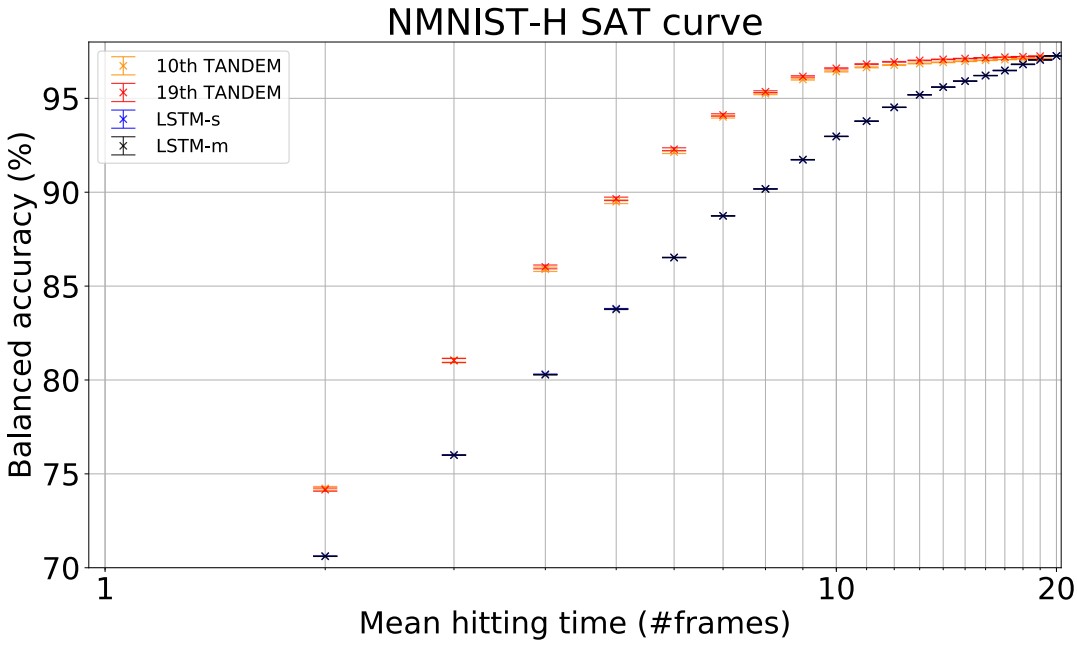

Figure 4: The speed-accuracy tradeoff curve. Compare this with Figure 3 in the main text. "10th TANDEM" means the 10-th order SPRT-TANDEM. "19th TANDEM" means the 19-th order SPRT-TANDEM. The numbers of trials for hyperparameter tuning if 200 for all the models. The error bars are standard error of mean (SEM). The numbers of trials for statistics are 440, 240, 200, and 200 for 10th TANDEM, 19th TANDEM, LSTM-s, and LSTM-m, respectively.

# E  LOSS FOR LOG-LIKELIHOOD RATIO ESTIMATION (LLLR)

In this section, we discuss a deep connection of the novel loss function, LLLR

$$L_{\mathrm{LLR}} = \frac{1}{M} \sum_{i \in I_1} |1 - \sigma(\log \hat{r}(X_i))| + \frac{1}{M} \sum_{i \in I_0} \sigma(\log \hat{r}(X_i)), \tag{86}$$

to density ratio estimation (Sugiyama et al. (2012; 2010)). Here, $X_i := \{x_i^{(t)} \in \mathbb{R}^{d_x}\}_{t=1}^T$ and $y_i \in \{1, 0\}$ ($i \in I := I_1 \cup I_0$, $T \in \mathbb{N}$, $d_x \in \mathbb{N}$) are a sequence of samples and a label, respectively, where $I$, $I_1$, and $I_0$ are the index sets of the whole dataset, class 1, and class 0, respectively. $\hat{r}(X_i)$ ($i \in I$) is the likelihood ratio of $X_i$. The hatted notation ($\hat{\cdot}$) means that the quantity is an estimation with, e.g., a neural network on the training dataset $\{(X_i, y_i)\}_{i \in I}$. Note that we do not necessarily have to compute $\hat{p}(X_i|y = 1)$ and $\hat{p}(X_i|y = 0)$ separately to obtain the likelihood ratio $\hat{r}(X_i) = \frac{\hat{p}(X|y=1)}{\hat{p}(X|y=0)}$; we can estimate $\hat{r}$ directly, as is explained in the following subsections.

In the following, we first introduce *KLIEP* (Kullback-Leibler Importance Estimation Procedure, Sugiyama et al. (2008)), which underlies the theoretical aspects of the LLLR. KLIEP was originally invented to estimate density ratio without directly estimating the densities. The idea is to minimize the Kullback-Leibler divergence of the true density $p(X|y = 1)$ and the estimated density $\hat{r}(X)p(X|y = 0)$, where $(X, y)$ is a sequential data-label pair defined on the same space as $(X_i, y_i)$'s. Next, we introduce the symmetrized KLIEP, which cares about not only $p(X|y = 1)$ and $\hat{r}(X)p(X|y = 0)$, but $p(X|y = 0)$ and $\hat{r}^{-1}(X)p(X|y = 1)$ to remove the asymmetry inherent in the Kullback-Leibler divergence. Finally, we show the equivalence of the symmetrized KLIEP to the LLLR; specifically, we show that *the LLLR minimizes the Kullback-Leibler divergence of the true and the estimated density, and further stabilizes the training by restricting the value of likelihood ratio.*

## E.1  DENSITY RATIO ESTIMATION AND KLIEP

In this section, we briefly review density ratio estimation and introduce KLIEP.

Density estimation is the construction of underlying probability densities based on observed datasets. Taking their ratio, we can naively estimate the density ratio; however, division by an estimated quantity is likely to enhance the estimation error (Sugiyama et al. (2012; 2010)). Density ratio estimation has been developed to circumvent this problem. We can categorize the methods to the following four: probabilistic classification, moment matching, density ratio fitting, and density fitting.

**Probabilistic classification.**    The idea of the probabilistic classification is that the posterior density $p(Y|X)$ is easier to estimate than the likelihood $p(X|Y)$. Notice that

$$\hat{r}(X) = \frac{\hat{p}(X|y=1)}{\hat{p}(X|y=0)} = \frac{\hat{p}(y=1|X)}{\hat{p}(y=0|X)} \frac{\hat{p}(y=0)}{\hat{p}(y=1)} = \frac{\hat{p}(y=1|X)}{\hat{p}(y=0|X)} \frac{M_0}{M_1}, \tag{87}$$

where $M_1$ and $M_0$ denote the number of the training data points with label 1 and 0 respectively. Thus we can estimate the likelihood ratio from the estimated posterior ratio. The multiplet cross-entropy loss conducts the density ratio estimation in this way.

**Moment matching.**    The moment matching approach aims to match the moments of $p(X|y = 1)$ and $\hat{r}(X)p(X|y = 0)$, according to the fact that two distributions are identical if and only if all moments agree with each other.

**Density ratio fitting.**    Without knowing the true densities, we can directly minimize the difference between the true and estimated ratio as follows:

$$\operatorname*{argmin}_{\hat{r}} \left[ \int dX p(X|y = 0)(\hat{r}(X) - r(X))^2 \right] \tag{88}$$

$$= \operatorname*{argmin}_{\hat{r}} \left[ \int dX p(X|y = 0)\hat{r}(X)^2 - 2 \int dX p(X|y = 1)\hat{r}(X) \right] \tag{89}$$

$$\doteq \operatorname*{argmin}_{\hat{r}} \left[ \frac{1}{M_0} \sum_{i \in I_0} \hat{r}(X_i)^2 - \frac{2}{M_1} \sum_{i \in I_1} \hat{r}(X_i) \right]. \tag{90}$$

Here, we applied the empirical approximation. In addition, we restrict the value of $\hat{r}(X)$: $\hat{r}(X) \geq 0$. Since (90) is not bounded below, we must add other terms or put more constraints, as is done in the original paper (Kanamori et al. (2009)). This formulation of density ratio estimation is referred to as least-squares importance fitting (LSIF, Kanamori et al. (2009)).

**Density fitting.** Instead of the squared expectation, KLIEP minimizes the Kullback-Leibler divergence:

$$\operatorname*{argmin}_{\hat{r}} \left[ \mathrm{KL}(p(X|y=1)||\hat{r}p(X|y=0)) \right] \tag{91}$$

$$= \operatorname*{argmin}_{\hat{r}} \left[ \int dX p(X|y=1) \log(\frac{p(X|y=1)}{\hat{r}(X)p(X|y=0)}) \right] \tag{92}$$

$$= \operatorname*{argmin}_{\hat{r}} \left[ - \int dX p(X|y=1) \log(\hat{r}(X)) \right] . \tag{93}$$

We need to restrict $\hat{r}$:

$$\begin{cases} 0 \leq \hat{r}(X) & \text{(94)} \\ \int dX \hat{r}(X) p(X|y=0) = 1 \quad , & \text{(95)} \end{cases}$$

The first inequality ensures the positivity of the probability ratio, while the second equation is the normalization condition. Applying the empirical approximation, we obtain the final objective and the constraints:

$$\operatorname*{argmin}_{\hat{r}} \left[ \frac{1}{M_1} \sum_{i \in I_1} - \log \hat{r}(X_i) \right] \tag{96}$$

$$\begin{cases} \hat{r}(X) \geq 0 & \text{(97)} \\ \dfrac{1}{M_0} \sum_{i \in I_0} \hat{r}(X_i) = 1 & \text{(98)} \end{cases}$$

Several papers implement the algorithms mentioned above using deep neural networks. In Nam & Sugiyama (2015), LSIF is applied to outlier detection with the deep neural network implementation, whereas in Khan et al. (2019), KLIEP and its variant are applied to changepoint detection.

## E.2 THE SYMMETRIZED KLIEP LOSS

As shown above, KLIEP minimizes the Kullback-Leibler divergence; however, its asymmetry can cause instability of the training, and thus we introduce the *symmetrized KLIEP loss*. A similar idea was proposed in Khan et al. (2019) independently of our analysis.

First, notice that

$$\mathrm{KL}(p(X|y=1)||\hat{r}p(X|y=0)) = \int dX p(X|y=1) \log(\frac{p(X|y=1)}{\hat{r}(X)p(X|y=0)}) \tag{99}$$

$$= - \int dX p(X|y=1) \log(\hat{r}(X)) + \mathrm{const.} \tag{100}$$

The constant term is independent of the weight parameters of the network and thus negligible in the following discussion. Similarly,

$$\mathrm{KL}(p(X|y=0)||\hat{r}^{-1}p(X|y=1)) = - \int dX p(X|y=1) \log(\hat{r}(X)^{-1}) + \mathrm{const.} \tag{101}$$

We need to restrict the value of $\hat{r}$ in order for $p(X|y=1)$ and $p(X|y=0)$ to be probability densities:

$$\begin{cases} 0 \leq \hat{r}(X)p(X|y=0) & \text{(102)} \\ \int dX \hat{r}(X) p(X|y=0) = 1 \quad , & \text{(103)} \end{cases}$$

and

$$
\begin{cases}
0 \leq \hat{r}(X)^{-1} p(X|y=1) & (104) \\
\int dX \hat{r}(X)^{-1} p(X|y=1) = 1 \qquad , & (105)
\end{cases}
$$

Therefore, we define the symmetrized KLIEP loss as

$$
L_{\mathrm{KLIEP}} := \int dX (-p(X|y=1) \log \hat{r}(X)) - \int dX (-p(X|y=0) \log \hat{r}(X)) \qquad (106)
$$

with the constraints (102)-(105). The estimated ratio function $\mathrm{argmin}_{\hat{r}(X)} L_{\mathrm{KLIEP}}$ with the constraints minimizes $\mathrm{KL}(p(X|y=1)||\hat{r}(X)p(X|y=0)) + \mathrm{KL}(p(X|y=0)||\hat{r}^{-1}p(X|y=1)))$. According to the empirical approximation, they reduce to

$$
L_{\mathrm{KLIEP}}(\{X_i\}_{i=1}^M) \doteqdot \frac{1}{M_1} \sum_{i \in I_1} -\log(\hat{r}(X_i)) + \frac{1}{M_0} \sum_{i \in I_0} -\log(\hat{r}(X_i)^{-1}), \qquad (107)
$$

$$
\begin{cases}
\hat{r}(X) \geq 0 & (108) \\
\dfrac{1}{M_0} \sum_{i \in I_0} \hat{r}(X_i) = 1 & (109) \\
\dfrac{1}{M_1} \sum_{i \in I_1} \hat{r}(X_i)^{-1} = 1 \qquad . & (110)
\end{cases}
$$

### E.3 THE LLLR AND DENSITY RATIO ESTIMATION

Let us investigate the LLLR in connection with the symmetrized KLIEP loss.

**Divergence terms.** First, we focus on the divergence terms in (107):

$$
\frac{1}{M_1} \sum_{i \in I_1} -\log(\hat{r}(X_i)) \qquad (111)
$$

$$
\frac{1}{M_0} \sum_{i \in I_0} -\log(\hat{r}(X_i)^{-1}) . \qquad (112)
$$

As shown above, decreasing (111) and (112) leads to minimizing the Kullback-Leibler divergence of $p(X|y=1)$ and $\hat{r}p(X|y=0)$ and that of $p(X|y=0)$ and $\hat{r}^{-1}p(X|y=1)$ respectively. The counterparts in the LLLR are

$$
L_{\mathrm{LLR}} = \frac{1}{M} \sum_{i \in I_1} |1 - \sigma(\log \hat{r}(X_i))| \qquad\qquad \leftrightarrow \frac{1}{M_1} \sum_{i \in I_1} -\log(\hat{r}(X_i)) \qquad (113)
$$

$$
+ \frac{1}{M} \sum_{i \in I_0} \sigma(\log \hat{r}(X_i)) \qquad\qquad \leftrightarrow \frac{1}{M_0} \sum_{i \in I_0} -\log(\hat{r}(X_i)^{-1}), \qquad (114)
$$

because, on one hand, both terms in (113) ensures the likelihood ratio $\hat{r}$ to be large for class 1, and, on the other hand, both terms in (114) ensures $\hat{r}$ to be small for class 0. Therefore, minimizing $L_{\mathrm{LLR}}$ is equivalent to decreasing both (111) and (112) and therefore to minimizing (108), i.e., the Kullback-Leibler divergences of the true and estimated densities.

Again, we emphasize that the LLLR is more stable, since $L_{\mathrm{LLR}}$ is lower-bounded unlike the KLIEP loss.

**Constraints.** Next, we show that the LLLR implicitly keeps $\hat{r}$ not too large nor too small; specifically, with increasing $\hat{R}(X_i) := |\log \hat{r}(X_i)|$, the gradient converges to zero before $\hat{R}(X_i)$ enters the region, e.g., $\hat{R}(X_i) \gtrsim 1$. Therefore the gradient descent converges before $\hat{r}(X_i)$ becomes too large or small. To show this, we first write the gradients explicitly:

$$
\nabla_W \sigma(\log(\hat{r}(X_i))) = \sigma'(\log \hat{r}(X_i)) \cdot \nabla_W \log \hat{r}(X_i) \qquad (115)
$$

where $W$ is the weight and $\sigma$ is the sigmoid function. We see that with increasing $\hat{R}(X_i) = |\log \hat{r}(X_i)|$, the factor

$$\sigma'(\log \hat{r}(X_i)) \tag{116}$$

converges to zero, because (116) $\sim 0$ for too large or small $\hat{r}(X_i)$, e.g., for around $\hat{R}(X_i) \gtrsim 1$. Thus the gradient (115) vanishes before $\hat{r}(X_i)$ becomes too large or small, i.e., keeping $\hat{r}(X_i)$ moderate.

In conclusion, the LLLR minimizes the difference between the true ($p(X|y=1)$ and $p(X|y=0)$) and the estimated ($\hat{r}^{-1}(X)p(X|y=1)$ and $\hat{r}(X)p(X|y=0)$) densities in the sense of the Kullback-Leibler divergence, including the effective constraints.

THE TRIVIAL SOLUTION IN EQUATION (115).    We show that vanishing

$$\nabla_W \log \hat{r}(X_i) \tag{117}$$

in (115) corresponds to a trivial solution ; i.e., we show that (117)$= 0$ forces the bottleneck feature vectors to be zero. Let us follow the notations in Table 4 and Figure 5. We particularly focus on the last components in the gradient $\nabla_W \log \hat{r}(X_i)$, i.e., $\nabla_{W_{ab}^{(L)}} \log \hat{r}(X_i)$ ($a \in [d_{L-1}]$ and $b \in [d_L] = \{0,1\}$). Specifically,

$$\nabla_{W_{ab}^{(L)}} \log \hat{r} = \nabla_{W_{ab}^{(L)}} \log p(X|\hat{y}=1) - \nabla_{W_{ab}^{(L)}} \log p(X|\hat{y}=0)$$
$$= \sum_{y=1,0} \left[ \frac{\partial g_y^{(L)}}{\partial W_{ab}^{(L)}} \frac{\partial \log p(X|\hat{y}=1)}{\partial g_y^{(L)}} - \frac{\partial g_y^{(L)}}{\partial W_{ab}^{(L)}} \frac{\partial \log p(X|\hat{y}=0)}{\partial g_y^{(L)}} \right] . \tag{118}$$

Since $\partial \log \hat{p}_{y'}/\partial g_y^{(L)} = \delta_{yy'} - \hat{p}_y$, where $y, y' \in \{1,0\}$ and $\delta_{y,y'}$ is the Kronecker delta, we see

$$\frac{\partial \log \hat{r}}{\partial g_y^{(L)}} = (\delta_{y1} - \hat{p}_y) - (\delta_{y0} - \hat{p}_y) = \begin{cases} 1 & (\text{if } y = 1) \\ -1 & (\text{if } y = 0) \end{cases} \tag{119}$$

$$\therefore (118) = \frac{\partial g_1^{(L)}}{W_{ab}^{(L)}} \cdot 1 + \frac{\partial g_0^{(L)}}{\partial W_{ab}^{(L)}} \cdot (-1) = \delta_{1b} f_a^{(L-1)} - \delta_{0b} f_a^{(L-1)}. \tag{120}$$

Thus (117) $= 0 \Longrightarrow$ (118)$= 0 \Longleftrightarrow f_a^{(L-1)} = 0$ ($\forall a \in [d_{L-1}]$), which is a trivial solution, because the bottleneck feature vector collapses to zero at convergence. Our experiments, however, show that our model does not tend to such a trivial solution; otherwise, the SPRT-TANDEM cannot attain such a high performance.

$$\begin{aligned}
\boldsymbol{f}^{(0)}(\boldsymbol{x}_i) &= (f_1^{(0)}, ..., f_{d_x}^{(0)})^{\mathrm{T}} = \boldsymbol{x_i} & &\in \mathbb{R}^{d_0 = d_x} \\
\boldsymbol{f}^{(l)}(\boldsymbol{x}_i) &= (f_1^{(l)}, ..., f_{d_l}^{(l)})^{\mathrm{T}} = \sigma(\boldsymbol{g}^{(l)}(\boldsymbol{x}_i)) & &\in \mathbb{R}^{d_l} (l = 0, 1, 2..., L-1) \\
\boldsymbol{f}^{(L)}(\boldsymbol{x}_i) &= (f_1^{(L)}, ..., f_{d_L}^{(L)})^{\mathrm{T}} = \mathrm{softmax}(\boldsymbol{g}^{(L)}(\boldsymbol{x}_i)) & &\in \mathbb{R}^{d_L = 2} \\
\boldsymbol{g}^{(l)}(\boldsymbol{x}_i) &= (g_1^{(l)}, ..., g_{d_l}^{(l)})^{\mathrm{T}} = W^{(l)\mathrm{T}} \boldsymbol{f}^{(l-1)}(\boldsymbol{x}_i) & &\in \mathbb{R}^{d_l} (l = 0, 1, 2..., L-1) \\
W^{(l)} & & &\in \mathbb{R}^{d_{l-1} \times d_l} (l = 0, 1, 2..., L-1)
\end{aligned}$$

$S = \{(\boldsymbol{x}_i, \boldsymbol{t}_i)\}_{i=1}^M$: training dataset       $d_x$: input dimension
$\boldsymbol{x}_i \in \mathbb{R}^{d_x}$: input vector       $L$: number of layers
$\boldsymbol{t}_i \in \mathbb{R}^2$: one-hot label vector       $\sigma$: activation function

Table 4: Notation. $\boldsymbol{f}^{(0)}(\boldsymbol{x}_i) = \boldsymbol{x}_i$ is the input vector with dimension $d_x \in \mathbb{N}$. $\boldsymbol{f}^{(l)}(\boldsymbol{x}_i)$ is a feature vector after the activation function $\sigma$ with dimension $d_l \in \mathbb{N}$. $\boldsymbol{f}^{(L)}(\boldsymbol{x}_i)$ is the output of the softmax function, and has $d_L = 2$, since we focus on the binary classification problem in this paper. $\boldsymbol{g}^{(l)}(\boldsymbol{x}_i)$ is a feature vector before the activation function with dimension $d_l \in \mathbb{N}$. $W^{(l)}$ is a weight matrix in the neural network. Figure 5 visualizes the network structure.

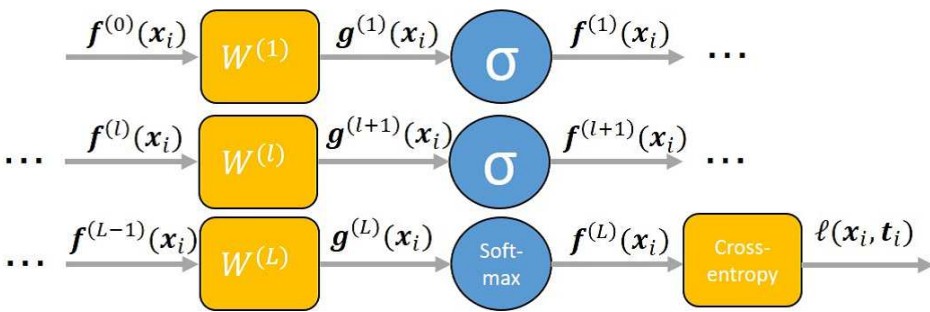

Figure 5: Visualization of the notation given in Table 4. We assume that the network has $L$ fully-connected layers $W^l$ and the activation function $\sigma$, with the final softmax with cross-entropy loss.

### E.4 PREPARATORY EXPERIMENT TESTING THE EFFECTIVENESS OF THE LLLR

To test if the proposed $L_{\text{LLR}}$ could effectively train a neural network, we ran two preliminary experiments before the main manuscript. First, we compared training the proposed network architecture $L_{\text{multiplet}}$, with and without $L_{\text{LLR}}$. Next, we compared training the network using $L_{\text{multiplet}}$ with $L_{\text{LLR}}$, and training with $L_{\text{multiplet}}$ with the KLIEP loss, $L_{\text{KLIEP}}$, whose numerator and denominator were carefully bounded so that the $L_{\text{KLIEP}}$ did not diverge.

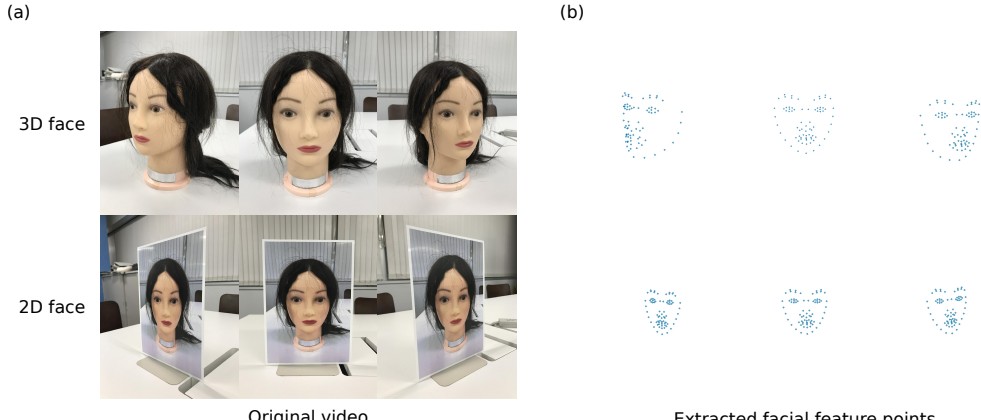

Original video                                        Extracted facial feature points

Figure 6: Depth-from-motion dataset for the face 3D-ness detection task. Only three frames out of ten frames are shown. Top and bottom faces are 3D and 2D faces, respectively. (a) Video of faces taken from various angles. (b) Facial feature points that are extracted with the feature extractor, $f_{w_{\text{FE}}}(x^{(t)})$

We tested the effectiveness of $L_{\text{LLR}}$ on a 3D-ness detection task on a depth-from-motion (DfM) dataset [4]. The DfM dataset was a small dataset containing 2320 and 2609 3D- and 2D- facial videos, respectively, each of which consisted of 10 frames. In each of the video, a face was filmed from various angles (Figure 6a), so that the dynamics of the facial features could be used to determine whether the face appearing in a video is flat, 2D face, or had a 3D structure. The recording device was an iPhone7. Here, the feature extractor $f_{w_{\text{FE}}}(x^{(t)})$ was the LBP-AdaBoost algorithm (Viola & Jones (2001)) combined with the supervised descent method (Xiong & De la Torre (2013)), which took the $t$-th frame of the given video as input $x_{(t)}$ and output facial feature points (Figure 6b). The feature points were output as a vector of 152 lengths, consisted of vertical and horizontal pixel positions of 76 facial feature points. The temporal integrator $g_{w_{\text{TI}}}(x^{(t)})$ is an LSTM, whose number of hidden units was the same as that of feature points. The 1st-order SPRT-TANDEM was evaluated on two hypotheses, $y = 1$: 2D face, and $y = 0$: 3D face. We assumed a flat prior, $p(y = 1) = p(y = 0)$. The validation and test data were 10% of the entire data randomly selected at the beginning of training,

---

[4]For the protection of personal information, this database cannot be made public.

respectively. We conducted a 10-fold cross-validation test to evaluate the effect of $L_{\text{LLR}}$.

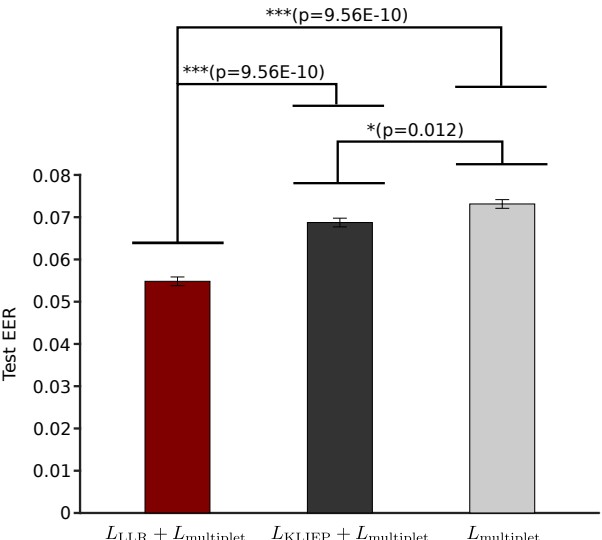

Figure 7: Statistical test of equal error rates (EERs) in 10-fold cross-validation test. Two-way ANOVA are conducted with a loss factor ($L_{\text{LLR}} + L_{\text{multiplet}}$, $L_{\text{KLIEP}} + L_{\text{multiplet}}$, and $L_{\text{multiplet}}$) and a epoch factor ($21 - 100$-th epoch). P-values with asterisks are statistically significant: one, two and three asterisks show $p < 0.05$, $p < 0.01$, and $p < 0.001$, respectively. Error bars show the standard errors of the mean (SEM).

We compared the classification performance of the SPRT-TANDEM network using both $L_{\text{LLR}}$ and $L_{\text{multiplet}}$, and using $L_{\text{multiplet}}$ only. We also compare $L_{\text{LLR}} + L_{\text{multiplet}}$ and $L_{\text{KLIEP}} + L_{\text{multiplet}}$. To use $L_{\text{KLIEP}}$ without making a loss diverge, we set the upper and lower bound of the numerator and denominator of $\hat{r}$ as $10^5$ and $10^{-5}$, respectively. Out of 100 training epochs, the results of the last 80 epochs were used to calculate test equal error rates (EERs). Two-way ANOVA with factors "loss type" and "epoch" were conducted to see if the difference in loss function caused statistically significantly different EERs. We included the epoch as a factor in order to see if the value of EER reached a plateau in the last 80 epochs (i.e., statistically NOT significant). As we expected, EER values across training epochs were not significantly different ($p = 0.17$). On the other hand, the loss type caused statistically significant differences between the loss groups (i.e., $L_{\text{LLR}} + L_{\text{multiplet}}$, $L_{\text{KLIEP}} + L_{\text{multiplet}}$, and $L_{\text{multiplet}}$. $p < 0.001$). Following Tukey-Kramer multi-comparison test showed that training with $L_{\text{LLR}}$ loss statistically significantly reduced the EER, compared to both $L_{\text{KLIEP}}$ ($p = 9.56 * 10^{-10}$) and the $L_{\text{LLR}}$-ablated loss ($p = 9.56 * 10^{-10}$). The result is plotted in Figure 7.

# F  PROBABILITY DENSITY RATIO ESTIMATION WITH THE LLLR

Below we test whether the proposed LLLR can help a neural network estimating the true probability density ratio. Providing the ground-truth probability density ratio was difficult in the three databases used in the main text, because it was prohibitive to find the true probability distribution out of the public databases containing real-world scenes. Thus, we create a toy-model estimating the probability density ratio of the two multivariate Gaussian distributions. Experimental results show that a multi-layer perceptron (MLP) trained with the proposed LLLR achieves smaller estimation error than an MLP with crossentropy (CE)-loss.

## F.1  EXPERIMENTAL SETTINGS

Following Sugiyama et al. 2008 Sugiyama et al. (2008), let $p_0(x)$ be the $d$-dimensional Gaussian density with mean $(2, 0, 0, ..., 0)$ and covariance identity, and $p_1(x)$ be the $d$-dimensional Gaussian density with mean $(0, 2, 0, ..., 0)$ and covariance identity.

The task for the neural network is to estimate the density ratio:

$$\hat{r}(x_i) = \frac{\hat{p}_1(x_i)}{\hat{p}_0(x_i)}. \tag{121}$$

Here, $x$ is sampled from one of the two Gaussian distributions, $p_0$ or $p_1$, and is associated with class label $y = 0$ or $y = 1$, respectively. We compared the two loss functions, CE-loss and LLLR:

$$\text{LLLR} := \frac{1}{N} \sum_{i=1}^{N} |y - \sigma(\log \hat{r}_i)| \tag{122}$$

where $\sigma$ is the sigmoid function.

A simple Neural network consists of 3-layer fully-connected network with nonlinear activation (ReLU) is used for estimating $\hat{r}(x)$.

Evaluation metric is normalized mean squared error (NMSE, Sugiyama et al. (2008)):

$$\text{NMSE} := \frac{1}{N} \sum_{i=1}^{N} \left( \frac{\hat{r_j}}{\sum_{j=1}^{N} \hat{r_j}} - \frac{r_i}{\sum_{j=1}^{N} r_j} \right)^2 \tag{123}$$

## F.2  DENSITY ESTIMATION RESULTS

To calculate statistics, the MLP was trained either with the LLLR or CE-loss, repeated 40 times with different random initial vairables. Figure 8 shows the mean NMSE with the shading shows standard error of the mean. Although the training with LLLR does not decrease NMSE well at the first few thousands of trials, the NMSE reaches as low as $10^{-5}$ around 14000 iterations. In contrast, the training with CE shows a steep decrease of NMSE in the first 2000 iterations, but saturates after that. Thus, the proposed LLLR not only facilitates the sequential binary hypothesis testing, but also facilitates the estimation of true density ratio.

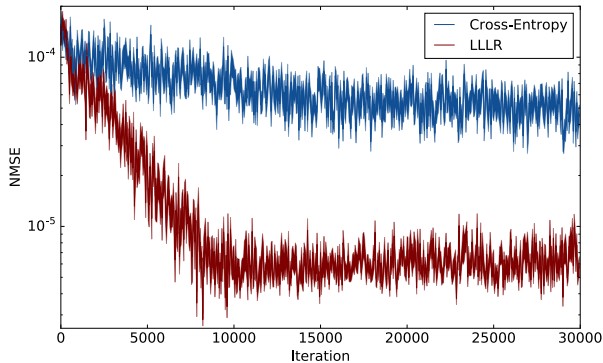

Figure 8: Normalized mean squared error (NMSE) - iteration curve. A multi-layer perceptron is trained either cross-entropy loss (blue) or the LLLR (red). Shades show the standard error of the mean (SEM).

## G  MULTIPLET CROSS-ENTROPY LOSS

In this section, we show that estimation of the true posterior is realized by minimizing the multiplet cross-entropy loss defined in Section 4 on the basis of the principle of maximum likelihood estimation.

First, let us consider the 1st order for simplicity. The multiplet cross-entropy loss ensures for the posterior $\hat{p}(y|x^{(t)})$ estimated by the network to be close to the true posterior $p(y|x^{(t)})$. Consider the Kullback-Leibler divergence of $\hat{p}(y|x^{(t)})$ and $p(y|x^{(t)})$ for some $x^{(t)} \in \mathbb{R}^{d_x}$ ($t \in \mathbb{N}$), where $y \in \{0,1\}$:

$$\underset{\hat{p}}{\operatorname{argmin}} \underset{x^{(t)} \sim p(x^{(t)})}{\mathbb{E}} [\mathrm{KL}(p(y|x^{(t)})||\hat{p}(y|x^{(t)}))] \tag{124}$$

$$= \underset{\hat{p}}{\operatorname{argmin}} \underset{(x^{(t)},y) \sim p(x^{(t)},y)}{\mathbb{E}} [-\log \hat{p}(y|x^{(t)})] \tag{125}$$

$$\fallingdotseq \underset{\hat{p}}{\operatorname{argmin}} \frac{1}{M} \sum_{i=1}^{M} [-\log \hat{p}(y_i|x_i^{(t)})] \tag{126}$$

Thus, the last line shows the smaller singlet loss leads to the smaller Kullback-Leibler divergence; in other words, we can estimate the true posterior density by minimizing the multiplet loss, which is necessary to run the SPRT algorithm.

Similarly, we adopt the doublet cross-entropy to estimate the true posterior $p(y|x^{(t)}, x^{(t+1)})$:

$$\underset{\hat{p}}{\operatorname{argmin}} \underset{(x^{(t)},x^{(t+1)}) \sim p(x^{(t)},x^{(t+1)})}{\mathbb{E}} [\mathrm{KL}(p(y|x^{(t)}, x^{(t+1)})||\hat{p}(y|x^{(t)}, x^{(t+1)}))] \tag{127}$$

$$\fallingdotseq \underset{\hat{p}}{\operatorname{argmin}} \frac{1}{M} \sum_{i=1}^{M} [-\log \hat{p}(y_i|x_i^{(t)}, x_i^{(t+1)})] . \tag{128}$$

The crucial difference from the singlet loss is that the doublet loss involves the temporal correlation between $x^{(t)}$ and $x^{(t+1)}$, being necessary to implement the SPRT-TANDEM. Similar statements hold for other orders.

## H  HYPERPARAMETER OPTIMIZATION

We used Optuna, the optimization software, to determine hyperparameters. Hyperparameter search trials are followed by performance evaluation trials of the fixed hyperparameter configuration. The evaluation criteria used by Optuna to find the best parameter combination is balanced accuracy. For models that produce multiple balanced accuracies / mean hitting time combinations, we use the average of balanced accuracy at every natural number of the mean hitting time (e.g., one frame, two frames).

### H.1  NOSAIC MNIST (NMNIST)

**SPRT-TANDEM: feature extractor.**   ResNet version 1 with 110 layers and 128 final output channels (total trainable parameters: 6.9M) is used. Hyperparameters are searched within the following space:

$$\text{learning rate} \in \{10^{-2}, 10^{-3}\}$$
$$\text{optimizer} \in \{\text{Adam}, \text{Momentum}, \text{RMSprop}\}$$
$$\text{weight decay} \in \{10^{-3}, 10^{-4}, 10^{-5}\}.$$

Where Adam, Momentum, and RMSprop are based on (Kingma & Ba (2014), Rumelhart et al. (1986), and Graves (2013)), respectively. Numbers of batch size and training epoch are fixed to 64 and 50, respectively. The best hyperparameter combination is summarized in Table 5.

Table 5: Hyperparameter tuning result of the SPRT-TANDEM feature extractor on NMNIST database.

| Trial | Learning rate | Batch size | Optimizer | Weight decay | Dropout |
|-------|---------------|------------|-----------|--------------|---------|
| 80 | $10^{-2}$ | 1024 | Adam | $10^{-5}$ | 0 |

One search trial takes approximately 5 hours on our computing infrastructure (see Appendix K).

**SPRT-TANDEM: temporal integrator.**   Peephole-LSTM with a hidden layer of size 128 (total trainable parameters: 0.1M) is used. Hyperparameters are searched within the following space:

$$\text{learning rate} \in \{10^{-2}, 10^{-3}, 10^{-4}, 10^{-5}\}$$
$$\text{batch size} \in \{256, 512, 1024\}$$
$$\text{optimizer} \in \{\text{Adam}, \text{Momentum}, \text{Adagrad}, \text{RMSprop}\}$$
$$\text{weight decay} \in \{10^{-3}, 10^{-4}, 10^{-5}\}.$$
$$\text{dropout} \in \{0, 0.1, 0.2, 0.3, 0.4, 0.5\}$$

Where Adagrad is based on (Duchi et al. (2011)). Number of training epochs is fixed to 50. The number of search trials and resulting best hyperparameter combination are summarized in Table 6.

Table 6: Hyperparameter tuning result of the SPRT-TANDEM temporal integrator. on NMNIST database.

|  | Trial | Learning rate | Batch size | Optimizer | Weight decay | Dropout |
|------|-------|---------------|------------|-----------|--------------|---------|
| 0th | 186 | $10^{-2}$ | 1024 | Adam | $10^{-5}$ | 0 |
| 1st | 151 | $10^{-3}$ | 1024 | RMSprop | $10^{-5}$ | 0 |
| 2nd | 182 | $10^{-2}$ | 1024 | RMSprop | $10^{-5}$ | 0 |
| 3rd | 140 | $10^{-2}$ | 1024 | Adam | $10^{-5}$ | 0 |
| 4th | 139 | $10^{-2}$ | 1024 | Adam | $10^{-5}$ | 0 |
| 5th | 140 | $10^{-2}$ | 1024 | Adam | $10^{-5}$ | 0 |
| 10th | 142 | $10^{-2}$ | 1024 | Adam | $10^{-5}$ | 0 |
| 19th | 193 | $10^{-3}$ | 1024 | RMSprop | $10^{-4}$ | 0 |

One search trial takes approximately 3 hours on our computing infrastructure (see Appendix K).

**LSTM-m / LSTM-s.** Peephole-LSTM with a hidden layer of size 128 (total trainable parameters: 0.1M) is used. Hyperparameters are searched within the following space:

$$\text{learning rate} \in \{10^{-2}, 10^{-3}, 10^{-4}, 10^{-5}\}$$
$$\text{optimizer} \in \{\text{Adam, Momentum, Adagrad, RMSprop}\}$$
$$\text{weight decay} \in \{10^{-3}, 10^{-4}, 10^{-5}\}.$$
$$\text{dropout} \in \{0, 0.1, 0.2, 0.3, 0.4, 0.5\}$$
$$\text{lambda} \in \{0.01, 0.1, 1, 6, 10, 100\}$$

where the lambda is a specific parameter of LSTM-m / LSTM-s. Batch size and number of training epochs are fixed to 1024 and 100, respectively. The number of search trials and resulting best hyperparameter combination are summarized in Table 7.

Table 7: Hyperparameter tuning result of the LSTM-m / LSTM-s on NMNIST database.

|  | Trial | Learning rate | Optimizer | Weight decay | Dropout | Lambda |
|---|---|---|---|---|---|---|
| LSTM-m | 160 | 1e-002 | Adam | 1e-004 | 0 | 1e-001 |
| LSTM-s | 159 | 1e-003 | RMSprop | 1e-004 | 0 | 1e-002 |

One search trial takes approximately 3 hours on our computing infrastructure (see Appendix K).

**EARLIEST.** LSTM with a hidden layer of size 128 (total trainable parameters: 0.1M) is used. Hyperparameters are searched within the following space:

$$\text{learning rate} \in \{10^{-1}, 10^{-2}, 10^{-3}, 10^{-4}, 10^{-5}\}$$
$$\text{optimizer} \in \{\text{Adam, Momentum, Adagrad, RMSprop}\}$$
$$\text{weight decay} \in \{10^{-3}, 10^{-4}, 10^{-5}\}.$$
$$\text{dropout} \in \{0, 0.1, 0.2, 0.3, 0.4, 0.5\}$$

Batch size and number of training epochs are fixed to $1^5$ and 2, respectively. The number of search trials and resulting best hyperparameter combination are summarized in Table 8.

Table 8: Hyperparameter tuning result of the EARLIEST on NMNIST database,

|  | Trial | Learning rate | Optimizer | Weight decay | Dropout s |
|---|---|---|---|---|---|
| EARLIEST (lambda:1e-002) | 161 | 1e-003 | Adam | 1e-004 | 0 |
| EARLIEST (lambda:1e-003) | 110 | 1e-003 | Adam | 1e-004 | 0 |

One search trial takes approximately 48 hours on our computing infrastructure (see Appendix K)

**3DResNet.** 3DResNet with 101 layers and 128 final output channels (total trainable parameters: 7.7M) is used. Hyperparameters are searched within the following space:

$$\text{learning rate} \in \{10^{-3}, 10^{-4}, 10^{-5}\}$$
$$\text{batch size} \in \{100, 200, 500\}$$
$$\text{weight decay} \in \{10^{-3}, 10^{-4}, 10^{-5}\}. \tag{129}$$

Optimizer and number of training epochs are fixed to Adam and 50, respectively. The number of search trials and resulting best hyperparameter combination are summarized in Table 9.

Table 9: Hyperparameter tuning result of the 3DResNet on NMNIST database,

| Input frames | Trial | Learning rate | Batch size | Weight decay |
|---|---|---|---|---|
| 5 | 50 | 1e-003 | 100 | 1e-004 |
| 10 | 50 | 1e-003 | 200 | 1e-004 |

---

[5]As of writing this manuscript, the original code of EARLIEST does not allow batch size larger than 1.

**Ablation experiment.** Peephole-LSTM with a hidden layer of size 128 (total trainable parameters: 0.1M) is used. Hyperparameters are searched within the following space:

$$\text{learning rate} \in \{10^{-1}, 10^{-2}, 10^{-3}, 10^{-4}, 10^{-5}\}$$
$$\text{optimizer} \in \{\text{Adam, Momentum, Adagrad, RMSprop}\}$$
$$\text{weight decay} \in \{10^{-3}, 10^{-4}, 10^{-5}\}.$$
$$\text{dropout} \in \{0, 0.1, 0.2, 0.3, 0.4, 0.5\}$$

Batch size and number of training epochs are fixed to 1024 and 100, respectively. The number of search trials and resulting best hyperparameter combination are summarized in Table 10.

Table 10: Hyperparameter tuning result on the ablation experiment.

|  |  | Trial | Learning rate | Optimizer | Weight decay | Dropout |
|---|---|---|---|---|---|---|
| Multiplet only | 1st | 143 | 1e-003 | Adam | 1e-004 | 0 |
|  | 19th | 156 | 1e-003 | RMSprop | 1e-005 | 0 |
| LLLR only | 1st | 141 | 1e-003 | RMSprop | 1e-005 | 0 |
|  | 19th | 156 | 1e-003 | Adam | 1e-005 | 0 |

## H.2 UCF101

**SPRT-TANDEM: feature extractor.** ResNet version 2 with 50 layers and 64 final output channels (total trainable parameters: 26K) is used. Hyperparameters are searched within the following space:

$$\text{learning rate} \in \{10^{-3}, 10^{-4}, 10^{-5}, 10^{-6}\}$$
$$\text{weight decay} \in \{10^{-3}, 10^{-4}, 10^{-5}\}.$$

$$(130)$$

Numbers of batch size, optimizer, and training epochs are fixed to 512, Adam, and 100, respectively. The best hyperparameter combination is summarized in Table 11.

Table 11: Hyperparameter tunnning result of the SPRT-TANDEM feature extractor on UCF database.

| Trial | Learning rate | Weight decay |
|---|---|---|
| 146 | $10^{-3}$ | $10^{-5}$ |

**SPRT-TANDEM: temporal integrator.** Peephole-LSTM with a hidden layer of size 64 (total trainable parameters: 33K) is used. Hyperparameters are searched within the following space:

$$\text{learning rate} \in \{10^{-4}, 10^{-5}, 10^{-6}, 10^{-7}\}$$
$$\text{batch size} \in \{57, 114, 171, 342\}$$
$$\text{optimizer} \in \{\text{Adam, RMSprop}\}$$
$$\text{dropout} \in \{0.1, 0.2, 0.3, 0.4\}$$

Numbers of weight decay and training epochs are fixed to $10^{-4}$ and 100, respectively. The number of search trials and the best hyperparameter combination are summarized in Table 12.

Table 12: Hyperparameter tuning result of the SPRT-TANDEM temporal integrator on UCF database.

|  | Trial | Learning rate | Batch size | Optimizer | Dropout |
|---|---|---|---|---|---|
| 0th | 100 | 1e-004 | 114 | RMSprop | 0.2 |
| 1st | 100 | 1e-004 | 57 | RMSprop | 0.4 |
| 2nd | 100 | 1e-004 | 342 | RMSprop | 0.3 |
| 3rd | 100 | 1e-004 | 57 | RMSprop | 0.3 |
| 5th | 100 | 1e-004 | 114 | RMSprop | 0.1 |
| 10th | 100 | 1e-004 | 171 | RMSprop | 0.1 |
| 14th | 100 | 1e-004 | 114 | RMSprop | 0.1 |
| 24th | 100 | 1e-004 | 171 | RMSprop | 0.1 |
| 49th | 100 | 1e-004 | 342 | RMSprop | 0.1 |

**LSTM-m / LSTM-s.** Peephole-LSTM with a hidden layer of size 64 (total trainable parameters: 33K) is used. Hyperparameters are searched within the following space:

$$\text{learning rate} \in \{10^{-4}, 10^{-5}, 10^{-6}, 10^{-7}\}$$
$$\text{batch size} \in \{57, 114, 171, 342\}$$
$$\text{optimizer} \in \{\text{Adam}, \text{Momentum}, \text{Adagrad}, \text{RMSprop}\}$$
$$\text{weight decay} \in \{10^{-3}, 10^{-4}, 10^{-5}\}.$$
$$\text{dropout} \in \{0.1, 0.2, 0.3, 0.4\}$$
$$\text{lambda} \in \{0.01, 0.1, 1, 6, 10, 100\}$$

The number of training epochs is fixed to 100. The number of search trials and resulting best hyperparameter combination are summarized in Table 13.

Table 13: Hyperparameter tuning result of the LSTM-m / LSTM-s on UCF database.

|  | Trial | Batch size | Learning rate | Optimizer | Weight decay | Dropout | Lambda |
|---|---|---|---|---|---|---|---|
| LSTM-m | 100 | 171 | 1e-003 | Adam | 1e-003 | 0 | 100 |
| LSTM-s | 100 | 171 | 1e-003 | Adam | 1e-003 | 0 | 100 |

**EARLIEST.** LSTM with a hidden layer of size 64 (total trainable parameters: 33K) is used. Hyperparameters are searched within the following space:

$$\text{learning rate} \in \{10^{-1}, 10^{-2}, 10^{-3}, 10^{-4}, 10^{-5}\}$$
$$\text{optimizer} \in \{\text{Adam}, \text{Momentum}, \text{Adagrad}, \text{RMSprop}\}$$
$$\text{weight decay} \in \{10^{-3}, 10^{-4}, 10^{-5}\}.$$
$$\text{dropout} \in \{0, 0.1, 0.2, 0.3, 0.4, 0.5\}$$

Batch size and number of training epochs are fixed to 1 and 30, respectively. The number of search trials and resulting best hyperparameter combination are summarized in Table 14.

Table 14: Hyperparameter combination of the EARLIEST on UCF database.

|  | Trial | Learning rate | Optimizer | Weight decay | Dropout |
|---|---|---|---|---|---|
| EARLIEST (lambda:1e-001) | 50 | 1e-005 | Adam | 1e-005 | 0.3 |
| EARLIEST (lambda:1e-005) | 50 | 1e-004 | Adam | 1e-004 | 0.3 |

**3DResNet.** 3DResNet with 50 layers and 64 final output channels (total trainable parameters: 52K) is used. Hyperparameters are searched within the following space:

$$\text{learning rate} \in \{10^{-3}, 10^{-4}, 10^{-5}\}$$
$$\text{weight decay} \in \{10^{-3}, 10^{-4}, 10^{-5}\}. \tag{131}$$

Batch size, optimizer and number of training epochs are fixed to 19, Adam, and 50, respectively. The number of search trials and resulting best hyperparameter combination are summarized in Table 15.

Table 15: Hyperparameter tuning result of the 3DResNet on NMNIST database,

| Input frames | Trial | Learning rate | Batch size | Weight decay |
|---|---|---|---|---|
| 15 | 50 | 1e-004 | 100 | 1e-005 |
| 25 | 50 | 1e-004 | 200 | 1e-005 |

## H.3   SIW

The large database and network size prevent us to run multiple parameter search trials on SiW database. Thus, we manually selected hyperparameters as follows.

**SPRT-TANDEM: feature extractor.**  ResNet version 2 with 152 layers and 512 final output channels (total trainable parameters: 3.7M) is used. Table 16 shows the fixed parameter combination.

Table 16: Hyperparameter tuning result of the SPRT-TANDEM feature extractor on SiW database.

| Epoch | Batch size | Learning rate | Optimizer | Weight decay |
|-------|-----------|---------------|-----------|--------------|
| 30 | 83 | 1e-004 | Adam | 1e-004 |

**SPRT-TANDEM: temporal integrator.**  Peephole-LSTM with a hidden layer of size 512s (total trainable parameters: 2.1M) is used. Table 17 shows the fixed parameter combination.

Table 17: Hyperparameter tuning result of the SPRT-TANDEM temporal integratorabase.

| Epoch | Batch size | Learning rate | Optimizer | Weight decay | Dropout |
|-------|-----------|---------------|-----------|--------------|---------|
| 50 | 83 | 1e-004 | Adam | 1e-004 | 0.3 |

**LSTM-m / LSTM-s.**  Peephole-LSTM with a hidden layer of size 512s (total trainable parameters: 2.1M) is used. Table 18 shows the fixed parameter combination.

Table 18: Hyperparameter tuning result of the LSTM-m / LSTM-s. on SiW database.

| Epoch | Batch size | Learning rate | Optimizer | Weight decay | Dropout | Lambda |
|-------|-----------|---------------|-----------|--------------|---------|--------|
| 50 | 83 | 1e-003 | Adam | 1e-003 | 0 | 100 |

**EARLIEST.**  LSTM with a hidden layer of size 512s (total trainable parameters: 2.1M) is used. Table 19 shows the fixed parameter combination.

Table 19: Hyperparameter combination of the EARLIEST on UCF database.

| Epoch | Learning rate | Optimizer | Weight decay | Dropout |
|-------|---------------|-----------|--------------|---------|
| 30 | 1e-004 | Adam | 1e-004 | 0.2 |

**3DResNet.**  3DResNet with 101 layers and 512 final output channels (total trainable parameters: 5.3M) is used. Table 20 shows the fixed parameter combination.

Table 20: Hyperparameter combination of the 3DResNet, on UCF database.

| Input frames | Epoch | Learning rate | Batch size | Optimizer | Weight decay |
|--------------|-------|---------------|-----------|-----------|--------------|
| 5 | 30 | 1e-004 | 5 | Adam | 1e-004 |
| 15 | 30 | 1e-004 | 3 | Adam | 1e-004 |
| 25 | 30 | 1e-004 | 3 | Adam | 1e-004 |

# I    STATISTICAL TEST DETAILS

The models we compared in the experiment have various numbers of trials due to the difference in training time; some models were prohibitively expensive for multiple runs (for example, 3DResNet takes 20 hrs/epoch on SiW database with NVIDIA RTX2080Ti.) In order to have an objective comparison of these models, we conducted statistical tests, Two-way ANOVA[6] followed by Tukey-Kramer multi-comparison test. In the tests, small numbers of trials lead to reduced test statistics, making it difficult to claim significance, because the test statistic of Tukey-Kramer method is proportional to $1/\sqrt{(1/n + 1/m)}$, where $n$ and $m$ are trial numbers of two models to be compared. Nevertheless, the SPRT-TANDEM is statistically significantly better than other baselines. One intuitive interpretation of this result is that "the SPRT-TANDEM achieved accuracy high enough so that only a few trials of baselines were needed to claim the significance." These statistical tests are standard practice in some research fields such as biological science, in which variable trial numbers are inevitable in experiments.

All the statistical tests are executed with a customized MATLAB (2017) script. Here, the two factors for ANOVA are (1) the model factor contains four members: the SPRT-TANDEM with the best performing order on the given database, LSTM-m, EARLIEST, and 3DResNet, and (2) the phase factor contains two or three members: early phase and late phase (NMNIST, UCF), or early, mid, and the late phase (SiW). The early, mid, and late phases are defined based on the number of frames used for classification. The actual number of frames is chosen so that the compared models can use as similar data samples as possible and thus depends on the database. The SPRT-TANDEM, LSTM-m, and 3DResNet can be compared with the same number of samples used. However, EARLIEST cannot flexibly change the average number of samples (i.e., mean hitting time); thus, we include the results of EARLIEST to groups with the closest number of data possible.

For NMNIST, five frames and ten frames are used to calculate the statistics of the early and late phases, respectively, except EARLIEST uses 4.37 and 19.66 frames on average in each phase. For UCF, 15 frames and 25 frames are used to calculate the statistics of the early and late phases, respectively, except EARLIEST uses 2.01 and 2.09 frames on average in each phase[7]. For SiW, 5, 15, and 25 frames are used to calculate the early, mid, and late phases, respectively, except EARLIEST uses 1.19, 8.21, and 32.06 frames. The p-values are summarized in the Tables 21-24. P-values with asterisks are statistically significant: one, two and three asterisks show $p < 0.05$, $p < 0.01$, and $p < 0.001$, respectively.

Table 21: p-values from the two-way ANOVA conducted on the three public databases.

|  | NMNIST | UCF | SiW |
|---|---|---|---|
| model | ***0.00 | ***0.00 | ***0.00 |
| phase | ***2.84E-261 | ***1.86E-65 | 0.63 |

Table 22: p-values from the Tukey-Kramer multi-comparison test conducted on NMNIST.

| | | 10th-order SPRT-TANDEM | | LSTM-m | | EARLIEST | | 3DResNet |
|---|---|---|---|---|---|---|---|---|
| | | early | late | early | late | early | late | early |
| 10th-order SPRT-TANDEM | late | ***5.99E-08 | | | | | | |
| LSTM-m | early | ***5.99E-08 | ***5.99E-08 | | | | | |
| | late | ***6.01E-08 | ***5.99E-08 | ***5.99E-08 | | | | |
| EARLIEST | early | ***5.99E-08 | ***5.99E-08 | 1.00 | ***5.99E-08 | | | |
| | late | ***1.26E-07 | ***5.99E-08 | ***5.99E-08 | 1.00 | ***5.99E-08 | | |
| 3DResNet | early | ***5.99E-08 | ***5.99E-08 | ***5.99E-08 | ***5.99E-08 | ***5.99E-08 | ***5.99E-08 | |
| | late | ***5.99E-08 | ***5.99E-08 | ***5.99E-08 | ***5.99E-08 | ***5.99E-08 | ***5.99E-08 | ***5.99E-08 |

---

[6]Note that we also conducted a three-way ANOVA with model, phase, and database factors to achieve qualitatively the same result verifying superiority of the SPRT-TANDEM over other algorithms.

[7]On UCF, EARLIEST does not use a large number of frames even when the hyperparameter lambda is set to a small value.

Table 23: p-values from the Tukey-Kramer multi-comparison test conducted on UCF.

| | | 10th-order SPRT-TANDEM | | LSTM-m | | EARLIEST | | 3DResNet |
|---|---|---|---|---|---|---|---|---|
| | | early | late | early | late | early | late | early |
| 10th-order SPRT-TANDEM | late | ***5.99E-08 | | | | | | |
| LSTM-m | early | ***1.24E-05 | ***5.99E-08 | | | | | |
| | late | ***5.99E-08 | ***1.24E-05 | ***5.99E-08 | | | | |
| EARLIEST | early | ***9.42E-07 | ***5.99E-08 | 0.24 | ***5.99E-08 | | | |
| | late | **3.49E-03 | ***9.42E-07 | ***6.37E-08 | 0.24 | ***5.99E-08 | | |
| 3DResNet | early | ***5.99E-08 | ***5.99E-08 | ***5.99E-08 | ***5.99E-08 | ***5.99E-08 | ***5.99E-08 | |
| | late | ***5.99E-08 | ***5.99E-08 | ***5.99E-08 | ***5.99E-08 | ***5.99E-08 | ***5.99E-08 | ***5.99E-08 |

Table 24: p-values from the Tukey-Kramer multi-comparison test conducted on SiW.

| | | 2nd-order SPRT-TANDEM | | LSTM-m | | EARLIEST | | 3DResNet |
|---|---|---|---|---|---|---|---|---|
| | | early | late | early | late | early | late | early |
| 2nd-order SPRT-TANDEM | late | 1.00 | | | | | | |
| LSTM-m | early | ***6.56E-08 | ***3.24E-04 | | | | | |
| | late | ***1.38E-05 | ***6.56E-08 | 1.00 | | | | |
| EARLIEST | early | ***5.99E-08 | ***5.99E-08 | ***5.99E-08 | ***5.99E-08 | | | |
| | late | ***5.99E-08 | ***5.99E-08 | ***6.01E-08 | ***5.99E-08 | 1.00 | | |
| 3DResNet | early | ***5.99E-08 | ***5.99E-08 | ***5.99E-08 | ***5.99E-08 | ***5.99E-08 | ***5.99E-08 | |
| | late | ***5.99E-08 | ***5.99E-08 | ***5.99E-08 | ***5.99E-08 | ***5.99E-08 | ***5.99E-08 | ***5.99E-08 |

## J  DETAILS OF THE EXPERIMENTS IN SECTION 5

Here we present the details of the experiments in Section 5. Figure 9 shows the SAT curves of all the models we use in the experiment. Figure 10, 11, and 12 show example LLR trajectories calculated using NMNIST, UCF, and SiW database, respectively. Tables 25 to 40 shows average balanced accuracy and standard error of the mean (SEM) at the corresponding number of frames that used for classification.

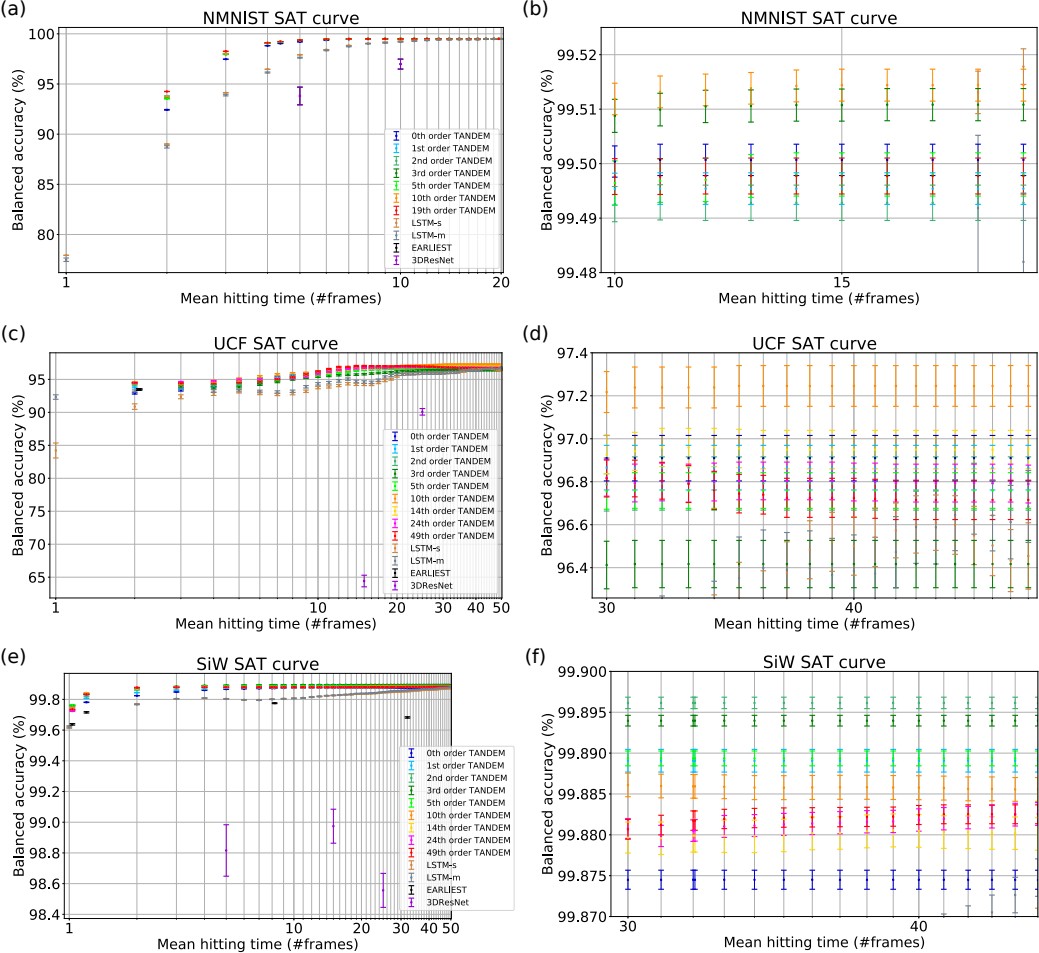

Figure 9: Speed-accuracy tradeoff (SAT) curves of all the models. The right three panels show magnified views of the left three panels. The magnified region is same as the region plotted in the insets in Figure 3a, 3b, and 3c. Error bars show the standard error of the mean (SEM). (a,b) NMNIST database. (c,d) UCF database. (e,f) SiW database.

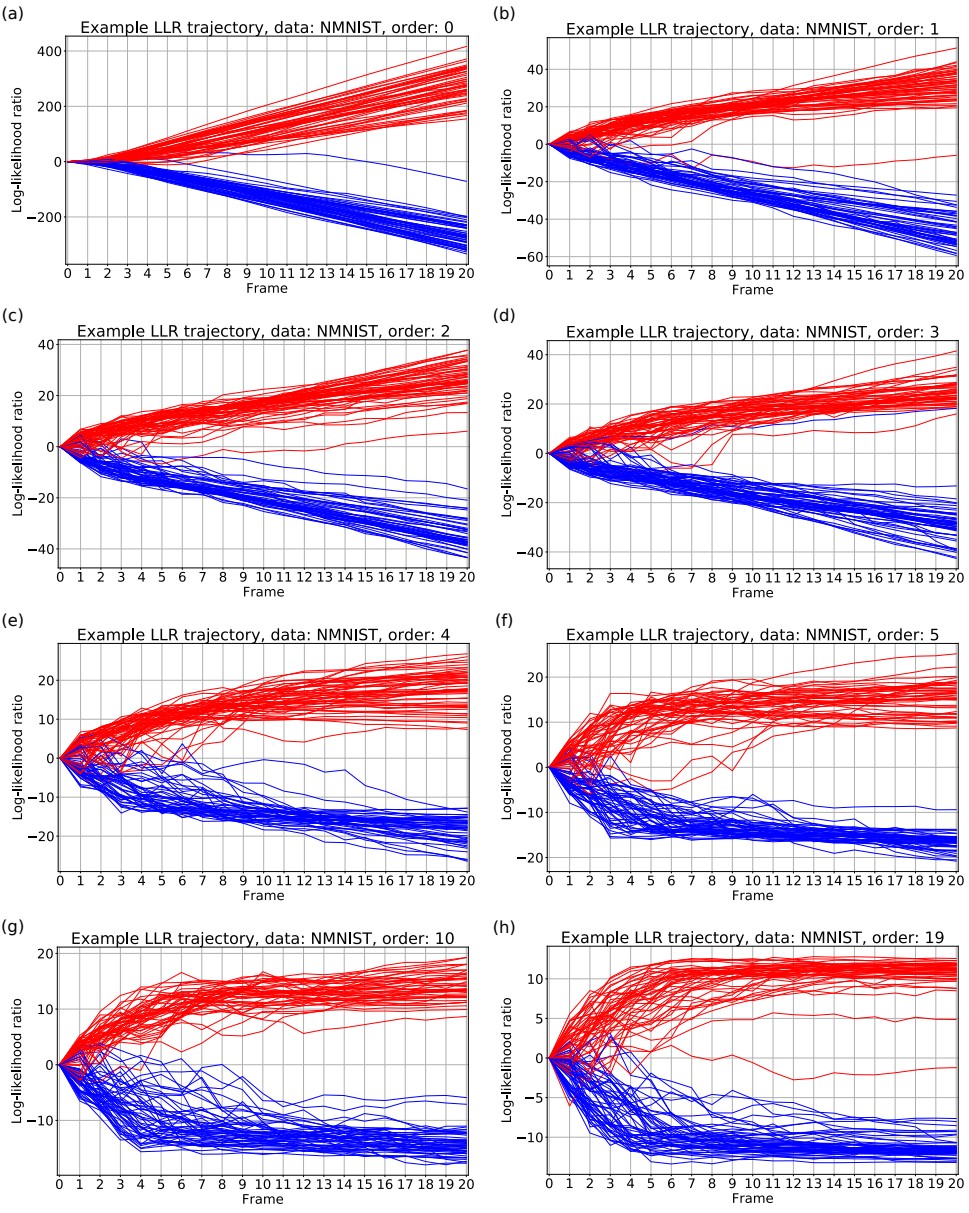

Figure 10: Log-likelihood ratio (LLR) trajectories calculated on NMNIST database. Red and blue trajectories represent odd and even class, respectively. Panels (a-i) shows results of 0th, 1st, 2nd, 3rd, 5th, 10th, 14th, 24th, and 49th-order SPRT-TANDEM, respectively.

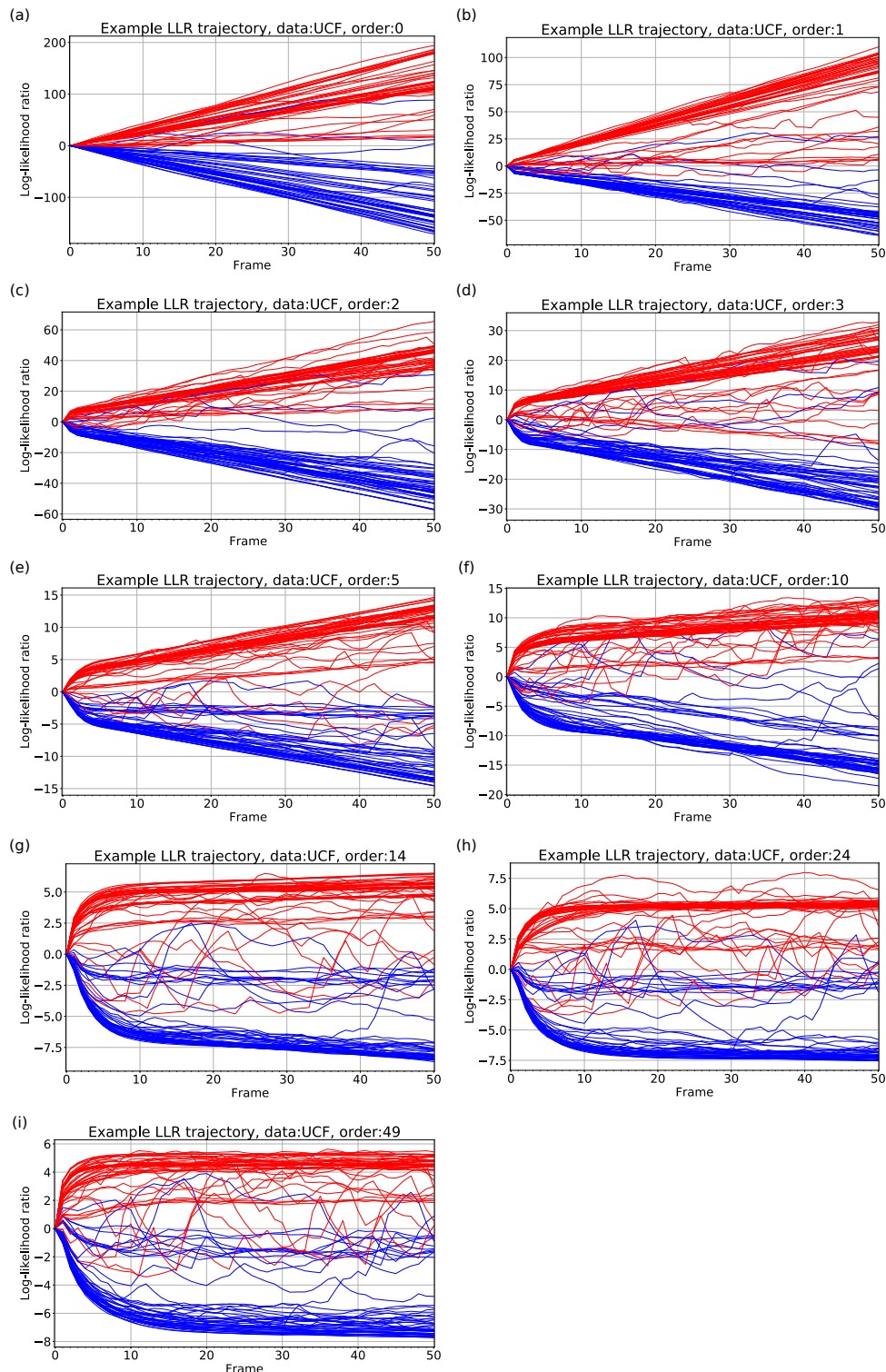

Figure 11: Log-likelihood ratio (LLR) trajectories calculated on UCF database. Red and blue trajectories represent handstand-pushups and handstand-walking class, respectively. Panels (a-i) shows results of 0th, 1st, 2nd, 3rd, 5th, 10th, 14th, 24th, and 49th-order SPRT-TANDEM, respectively.

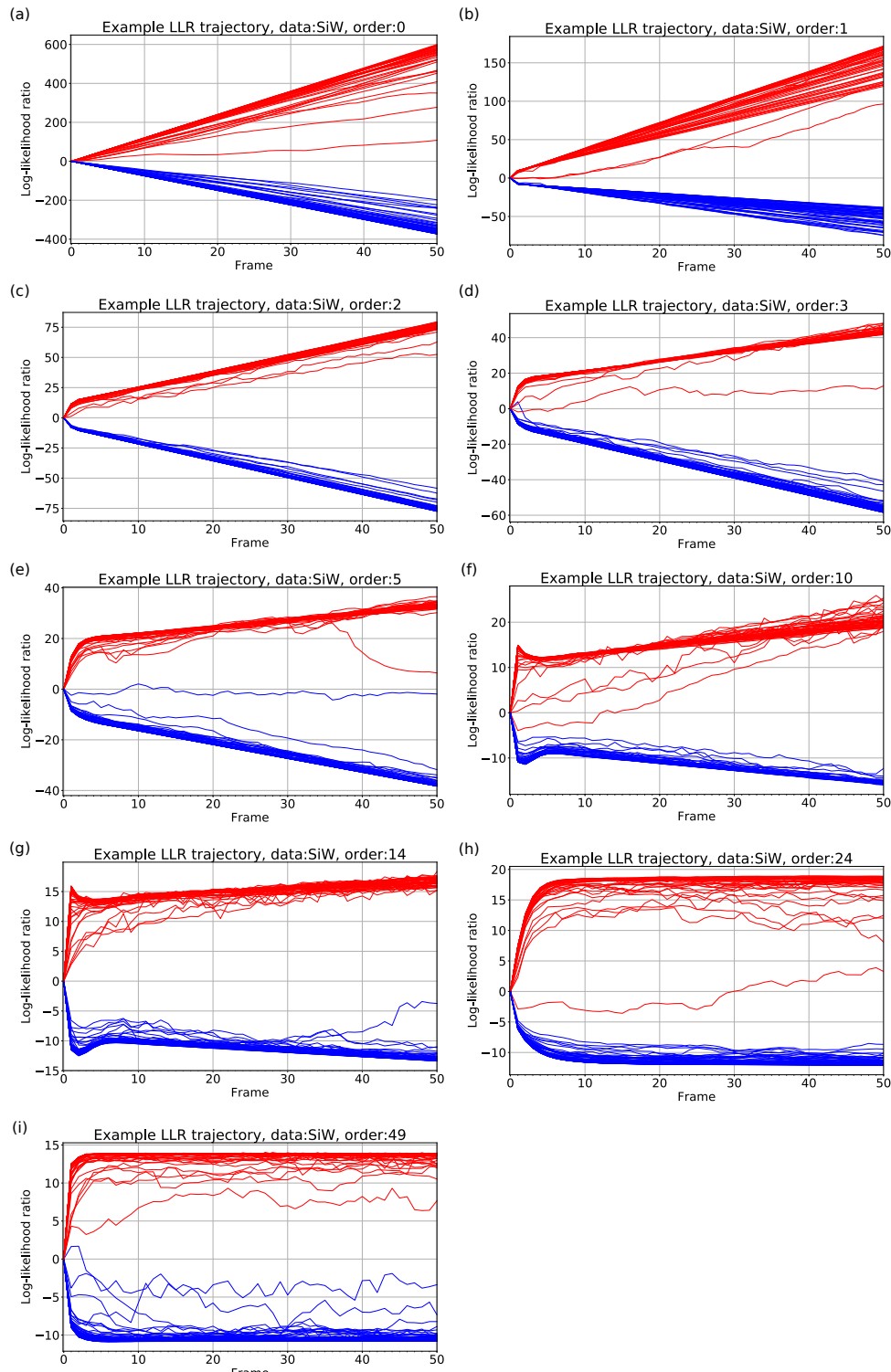

Figure 12: Log-likelihood ratio (LLR) trajectories calculated on SiW database. Red and blue trajectories represent live and spoof class, respectively. Panels (a-i) shows results of 0th, 1st, 2nd, 3rd, 5th, 10th, 14th, 24th, and 49th-order SPRT-TANDEM, respectively.

Table 25: Database: NMNIST, model: SPRT-TANDEM (1/2)

| Frame | 0th Mean | SEM | 1st Mean | SEM | 2nd Mean | SEM | 3rd Mean | SEM |
|---|---|---|---|---|---|---|---|---|
| 2 | 92.4260 | 3.5055e-004 | 93.8059 | 1.8919e-004 | 93.7317 | 1.8441e-004 | 93.5697 | 2.2729e-004 |
| 3 | 97.4691 | 1.0305e-004 | 98.0409 | 7.3110e-005 | 98.0061 | 8.2347e-005 | 97.9515 | 9.5436e-005 |
| 4 | 98.8174 | 5.1880e-005 | 99.0658 | 4.8780e-005 | 99.0728 | 4.9126e-005 | 99.0582 | 5.3061e-005 |
| 5 | 99.2033 | 3.4794e-005 | 99.3400 | 3.5587e-005 | 99.3565 | 4.2477e-005 | 99.3593 | 4.0801e-005 |
| 6 | 99.3700 | 3.4820e-005 | 99.4601 | 3.4799e-005 | 99.4456 | 3.8399e-005 | 99.4633 | 3.4886e-005 |
| 7 | 99.4480 | 3.2822e-005 | 99.4866 | 3.2035e-005 | 99.4783 | 3.5115e-005 | 99.4946 | 3.2541e-005 |
| 8 | 99.4837 | 2.9617e-005 | 99.4928 | 3.0137e-005 | 99.4897 | 3.3744e-005 | 99.5041 | 3.1241e-005 |
| 9 | 99.4957 | 2.9651e-005 | 99.4948 | 3.0002e-005 | 99.4917 | 3.2540e-005 | 99.5071 | 3.1055e-005 |
| 10 | 99.5004 | 2.8810e-005 | 99.4953 | 2.9388e-005 | 99.4926 | 3.2438e-005 | 99.5088 | 3.0463e-005 |
| 11 | 99.5007 | 2.8895e-005 | 99.4954 | 2.9156e-005 | 99.4929 | 3.2466e-005 | 99.5099 | 3.0071e-005 |
| 12 | 99.5007 | 2.8895e-005 | 99.4954 | 2.9156e-005 | 99.4928 | 3.2570e-005 | 99.5105 | 2.9752e-005 |
| 13 | 99.5007 | 2.8895e-005 | 99.4954 | 2.9156e-005 | 99.4928 | 3.2570e-005 | 99.5106 | 2.9649e-005 |
| 14 | 99.5007 | 2.8895e-005 | 99.4954 | 2.9156e-005 | 99.4928 | 3.2570e-005 | 99.5107 | 2.9599e-005 |
| 15 | 99.5007 | 2.8895e-005 | 99.4954 | 2.9156e-005 | 99.4928 | 3.2570e-005 | 99.5107 | 2.9599e-005 |
| 16 | 99.5007 | 2.8895e-005 | 99.4954 | 2.9156e-005 | 99.4928 | 3.2570e-005 | 99.5108 | 2.9628e-005 |
| 17 | 99.5007 | 2.8895e-005 | 99.4954 | 2.9156e-005 | 99.4928 | 3.2570e-005 | 99.5108 | 2.9628e-005 |
| 18 | 99.5007 | 2.8895e-005 | 99.4954 | 2.9156e-005 | 99.4928 | 3.2570e-005 | 99.5108 | 2.9628e-005 |
| 19 | 99.5007 | 2.8895e-005 | 99.4954 | 2.9156e-005 | 99.4928 | 3.2570e-005 | 99.5108 | 2.9628e-005 |
| stat. #trials | 100 | | 100 | | 120 | | 120 | |

Table 26: Database: NMNIST, model: SPRT-TANDEM (2/2)

| Frame | 4th Mean | SEM | 5th Mean | SEM | 10th Mean | SEM | 19th Mean | SEM |
|---|---|---|---|---|---|---|---|---|
| 2 | 93.5525 | 3.0287e-004 | 93.5443 | 3.5917e-004 | 93.7672 | 1.7120e-004 | 94.2514 | 1.0840e-004 |
| 3 | 97.9744 | 9.7434e-005 | 97.9483 | 1.5584e-004 | 98.0168 | 8.0357e-005 | 98.2590 | 5.3897e-005 |
| 4 | 99.0528 | 7.7568e-005 | 99.0647 | 6.0556e-005 | 99.0854 | 4.1829e-005 | 99.1162 | 4.0463e-005 |
| 5 | 99.3464 | 5.6959e-005 | 99.3553 | 5.1067e-005 | 99.3714 | 4.0986e-005 | 99.3675 | 3.7986e-005 |
| 6 | 99.4538 | 4.9952e-005 | 99.4513 | 4.3367e-005 | 99.4657 | 3.4016e-005 | 99.4645 | 3.5654e-005 |
| 7 | 99.4854 | 4.3020e-005 | 99.4803 | 4.3188e-005 | 99.4940 | 3.1882e-005 | 99.4943 | 3.3249e-005 |
| 8 | 99.4934 | 4.2301e-005 | 99.4894 | 4.2109e-005 | 99.5049 | 3.0668e-005 | 99.4975 | 3.3069e-005 |
| 9 | 99.5000 | 4.1649e-005 | 99.4940 | 4.1320e-005 | 99.5096 | 2.9477e-005 | 99.4976 | 3.3061e-005 |
| 10 | 99.5002 | 4.0642e-005 | 99.4964 | 4.1014e-005 | 99.5119 | 2.9025e-005 | 99.4976 | 3.3061e-005 |
| 11 | 99.5013 | 4.0759e-005 | 99.4969 | 4.0133e-005 | 99.5132 | 2.9260e-005 | 99.4976 | 3.3061e-005 |
| 12 | 99.5019 | 4.0572e-005 | 99.4970 | 4.0075e-005 | 99.5135 | 2.9101e-005 | 99.4977 | 3.3142e-005 |
| 13 | 99.5020 | 4.0595e-005 | 99.4977 | 3.9750e-005 | 99.5138 | 2.9129e-005 | 99.4977 | 3.3142e-005 |
| 14 | 99.5022 | 4.0562e-005 | 99.4980 | 4.0004e-005 | 99.5143 | 2.9373e-005 | 99.4977 | 3.3142e-005 |
| 15 | 99.5022 | 4.0562e-005 | 99.4980 | 4.0004e-005 | 99.5144 | 2.9410e-005 | 99.4977 | 3.3142e-005 |
| 16 | 99.5022 | 4.0562e-005 | 99.4980 | 4.0004e-005 | 99.5144 | 2.9410e-005 | 99.4977 | 3.3142e-005 |
| 17 | 99.5022 | 4.0562e-005 | 99.4980 | 4.0004e-005 | 99.5144 | 2.9410e-005 | 99.4977 | 3.3142e-005 |
| 18 | 99.5022 | 4.0562e-005 | 99.4980 | 4.0004e-005 | 99.5144 | 2.9410e-005 | 99.4977 | 3.3142e-005 |
| 19 | 99.5022 | 4.0562e-005 | 99.4980 | 4.0004e-005 | 99.5144 | 2.9410e-005 | 99.4977 | 3.3142e-005 |
| stat. #trials | 70 | | 70 | | 139 | | 100 | |

Table 27: Database: NMNIST, model: LSTM-m/s

| Frame | LSTM-m Mean | SEM | LSTM-s Mean | SEM |
|---|---|---|---|---|
| 1 | 77.4855 | 1.7242e-003 | 77.9245 | 9.4813e-005 |
| 2 | 88.7368 | 1.1776e-003 | 89.0127 | 9.0176e-005 |
| 3 | 93.8943 | 8.0486e-004 | 94.1324 | 7.0497e-005 |
| 4 | 96.1456 | 5.1075e-004 | 96.4685 | 6.6315e-005 |
| 5 | 97.6231 | 4.1456e-004 | 97.9092 | 4.6285e-005 |
| 6 | 98.3505 | 3.5449e-004 | 98.4303 | 4.0016e-005 |
| 7 | 98.7364 | 2.8124e-004 | 98.8419 | 3.6436e-005 |
| 8 | 99.0151 | 2.4863e-004 | 99.0461 | 4.2539e-005 |
| 9 | 99.0935 | 2.0928e-004 | 99.1751 | 4.2772e-005 |
| 10 | 99.1911 | 1.8862e-004 | 99.2758 | 4.7584e-005 |
| 11 | 99.2693 | 1.6739e-004 | 99.3305 | 4.4399e-005 |
| 12 | 99.3401 | 1.6169e-004 | 99.3907 | 3.8428e-005 |
| 13 | 99.3945 | 1.5403e-004 | 99.4398 | 4.6851e-005 |
| 14 | 99.3994 | 1.4535e-004 | 99.4494 | 4.4525e-005 |
| 15 | 99.4207 | 1.4639e-004 | 99.4548 | 4.2956e-005 |
| 16 | 99.4009 | 1.2789e-004 | 99.4469 | 3.4843e-005 |
| 17 | 99.4142 | 1.2698e-004 | 99.4518 | 3.8027e-005 |
| 18 | 99.4918 | 1.3414e-004 | 99.5131 | 3.9145e-005 |
| 19 | 99.4820 | 1.2662e-004 | 99.5178 | 3.2760e-005 |
| 20 | 99.4963 | 1.2963e-004 | 99.5217 | 3.2808e-005 |
| stat. #trials | 138 | | 120 | |

Table 28: Database: NMNIST, model: EARLIEST

| | Lambda: 1e-2 | | | Lambda: 1e-3 | | |
| | Mean hitting time | Mean | SEM | Mean hitting time | Mean | SEM |
|---|---|---|---|---|---|---|
| | 4.37 | 97.4830 | 1.7986e-004 | 19.66 | 99.3407 | 4.3111e-005 |
| stat. #trials | | 130 | | | 130 | |

Table 29: Database: NMNIST, model: 3DResNet

| | 5 frames | | 10 frames | |
| | Mean | SEM | Mean | SEM |
|---|---|---|---|---|
| | 93.8059 | 5.4340e-004 | 96.9833 | 6.9308e-004 |
| stat. #trials | | 100 | | 200 |

Table 30: Database: UCF, model: SPRT-TANDEM (1/2)

| | 0th | | 1st | | 2nd | | 3rd | | 5th | |
| Frame | Mean | SEM | Mean | SEM | Mean | SEM | Mean | SEM | Mean | SEM |
|---|---|---|---|---|---|---|---|---|---|---|
| 2 | 92.9177 | 1.4953e-003 | 93.7855 | 8.0825e-004 | 94.2036 | 9.6600e-004 | 94.4136 | 8.9367e-004 | 94.4764 | 6.0138e-004 |
| 3 | 93.3782 | 1.4701e-003 | 93.5709 | 1.0460e-003 | 93.9745 | 1.3348e-003 | 93.8827 | 1.3175e-003 | 94.3345 | 8.2804e-004 |
| 4 | 94.0586 | 1.5222e-003 | 93.9336 | 1.2928e-003 | 94.0073 | 1.3351e-003 | 93.7145 | 1.4442e-003 | 94.3264 | 8.7777e-004 |
| 5 | 94.6577 | 1.4199e-003 | 94.5586 | 1.3820e-003 | 94.0909 | 1.3482e-003 | 93.8582 | 1.4708e-003 | 94.5364 | 8.7558e-004 |
| 6 | 95.0777 | 1.3280e-003 | 95.1100 | 1.2698e-003 | 94.3550 | 1.3701e-003 | 94.2305 | 1.4317e-003 | 94.8036 | 8.8973e-004 |
| 7 | 95.3464 | 1.2162e-003 | 95.3909 | 1.1564e-003 | 94.7977 | 1.3590e-003 | 94.6282 | 1.3468e-003 | 94.8827 | 9.3600e-004 |
| 8 | 95.6127 | 1.1504e-003 | 95.6595 | 1.0512e-003 | 95.2432 | 1.3280e-003 | 95.0000 | 1.2496e-003 | 95.2145 | 9.8280e-004 |
| 9 | 95.8473 | 1.1150e-003 | 95.8295 | 1.0307e-003 | 95.6018 | 1.2870e-003 | 95.2432 | 1.2262e-003 | 95.4045 | 9.8075e-004 |
| 10 | 96.0418 | 1.1038e-003 | 95.9627 | 1.0613e-003 | 95.8377 | 1.2843e-003 | 95.4218 | 1.2005e-003 | 95.6064 | 1.0118e-003 |
| 11 | 96.1782 | 1.0819e-003 | 96.0909 | 1.0505e-003 | 95.9918 | 1.3183e-003 | 95.5318 | 1.1916e-003 | 95.8023 | 1.0206e-003 |
| 12 | 96.2664 | 1.0701e-003 | 96.2341 | 1.0541e-003 | 96.1245 | 1.2994e-003 | 95.6382 | 1.1507e-003 | 96.0500 | 9.5854e-004 |
| 13 | 96.4968 | 1.0795e-003 | 96.3382 | 1.0504e-003 | 96.2764 | 1.3292e-003 | 95.6905 | 1.1642e-003 | 96.1882 | 9.5132e-004 |
| 14 | 96.6577 | 1.0987e-003 | 96.4627 | 1.0824e-003 | 96.3818 | 1.3388e-003 | 95.7232 | 1.1691e-003 | 96.3395 | 9.3848e-004 |
| 15 | 96.8273 | 1.0797e-003 | 96.555 | 1.0815e-003 | 96.4618 | 1.3237e-003 | 95.7886 | 1.1597e-003 | 96.4382 | 9.3831e-004 |
| 16 | 96.8868 | 1.0663e-003 | 96.6600 | 1.0617e-003 | 96.5232 | 1.3099e-003 | 95.8759 | 1.1770e-003 | 96.5023 | 9.0577e-004 |
| 17 | 96.9009 | 1.0672e-003 | 96.7055 | 1.0741e-003 | 96.5714 | 1.3077e-003 | 95.9677 | 1.1931e-003 | 96.5264 | 8.9523e-004 |
| 18 | 96.9100 | 1.0621e-003 | 96.7700 | 1.0667e-003 | 96.5950 | 1.2969e-003 | 96.0177 | 1.1833e-003 | 96.5695 | 8.7976e-004 |
| 19 | 96.9050 | 1.0605e-003 | 96.8155 | 1.0478e-003 | 96.6500 | 1.2776e-003 | 96.0909 | 1.1694e-003 | 96.6064 | 8.8584e-004 |
| 20 | 96.9050 | 1.0605e-003 | 96.8609 | 1.0204e-003 | 96.6691 | 1.2741e-003 | 96.1927 | 1.1463e-003 | 96.6345 | 8.8151e-004 |
| 21 | 96.9095 | 1.0603e-003 | 96.8700 | 1.0182e-003 | 96.6873 | 1.2733e-003 | 96.2518 | 1.1451e-003 | 96.6668 | 8.6249e-004 |
| 22 | 96.9095 | 1.0603e-003 | 96.8655 | 1.0313e-003 | 96.7055 | 1.2740e-003 | 96.2845 | 1.1448e-003 | 96.6950 | 8.5438e-004 |
| 23 | 96.9095 | 1.0603e-003 | 96.8700 | 1.0312e-003 | 96.7327 | 1.2627e-003 | 96.3077 | 1.1310e-003 | 96.7182 | 8.3851e-004 |
| 24 | 96.9095 | 1.0603e-003 | 96.8655 | 1.0399e-003 | 96.7464 | 1.2558e-003 | 96.3486 | 1.1241e-003 | 96.7223 | 8.4269e-004 |
| 25 | 96.9095 | 1.0603e-003 | 96.8655 | 1.0399e-003 | 96.7555 | 1.2561e-003 | 96.3759 | 1.1030e-003 | 96.7268 | 8.4295e-004 |
| 26 | 96.9095 | 1.0603e-003 | 96.8655 | 1.0399e-003 | 96.7555 | 1.2561e-003 | 96.3941 | 1.1085e-003 | 96.7409 | 8.3699e-004 |
| 27 | 96.9095 | 1.0603e-003 | 96.8655 | 1.0399e-003 | 96.7645 | 1.2453e-003 | 96.4032 | 1.1102e-003 | 96.7455 | 8.2897e-004 |
| 28 | 96.9095 | 1.0603e-003 | 96.8655 | 1.0399e-003 | 96.7782 | 1.2406e-003 | 96.4077 | 1.1092e-003 | 96.7545 | 8.2935e-004 |
| 29 | 96.9095 | 1.0603e-003 | 96.8655 | 1.0399e-003 | 96.7827 | 1.2407e-003 | 96.4123 | 1.1100e-003 | 96.7545 | 8.2935e-004 |
| 30 | 96.9095 | 1.0603e-003 | 96.8655 | 1.0399e-003 | 96.7873 | 1.2407e-003 | 96.4123 | 1.1100e-003 | 96.7545 | 8.2935e-004 |
| 31 | 96.9095 | 1.0603e-003 | 96.8655 | 1.0399e-003 | 96.7918 | 1.2424e-003 | 96.4168 | 1.1033e-003 | 96.7591 | 8.3202e-004 |
| 32 | 96.9095 | 1.0603e-003 | 96.8655 | 1.0399e-003 | 96.7918 | 1.2424e-003 | 96.4168 | 1.1033e-003 | 96.7591 | 8.3202e-004 |
| 33 | 96.9095 | 1.0603e-003 | 96.8655 | 1.0399e-003 | 96.7918 | 1.2424e-003 | 96.4168 | 1.1033e-003 | 96.7591 | 8.3202e-004 |
| 34 | 96.9095 | 1.0603e-003 | 96.8655 | 1.0399e-003 | 96.7918 | 1.2424e-003 | 96.4168 | 1.1033e-003 | 96.7591 | 8.3202e-004 |
| 35 | 96.9095 | 1.0603e-003 | 96.8655 | 1.0399e-003 | 96.7918 | 1.2424e-003 | 96.4168 | 1.1033e-003 | 96.7591 | 8.3202e-004 |
| 36 | 96.9095 | 1.0603e-003 | 96.8655 | 1.0399e-003 | 96.7918 | 1.2424e-003 | 96.4168 | 1.1033e-003 | 96.7591 | 8.3202e-004 |
| 37 | 96.9095 | 1.0603e-003 | 96.8655 | 1.0399e-003 | 96.7918 | 1.2424e-003 | 96.4168 | 1.1033e-003 | 96.7591 | 8.3202e-004 |
| 38 | 96.9095 | 1.0603e-003 | 96.8655 | 1.0399e-003 | 96.7918 | 1.2424e-003 | 96.4168 | 1.1033e-003 | 96.7591 | 8.3202e-004 |
| 39 | 96.9095 | 1.0603e-003 | 96.8655 | 1.0399e-003 | 96.7918 | 1.2424e-003 | 96.4168 | 1.1033e-003 | 96.7591 | 8.3202e-004 |
| 40 | 96.9095 | 1.0603e-003 | 96.8655 | 1.0399e-003 | 96.7918 | 1.2424e-003 | 96.4168 | 1.1033e-003 | 96.7591 | 8.3202e-004 |
| 41 | 96.9095 | 1.0603e-003 | 96.8655 | 1.0399e-003 | 96.7918 | 1.2424e-003 | 96.4168 | 1.1033e-003 | 96.7591 | 8.3202e-004 |
| 42 | 96.9095 | 1.0603e-003 | 96.8655 | 1.0399e-003 | 96.7918 | 1.2424e-003 | 96.4168 | 1.1033e-003 | 96.7591 | 8.3202e-004 |
| 43 | 96.9095 | 1.0603e-003 | 96.8655 | 1.0399e-003 | 96.7918 | 1.2424e-003 | 96.4168 | 1.1033e-003 | 96.7591 | 8.3202e-004 |
| 44 | 96.9095 | 1.0603e-003 | 96.8655 | 1.0399e-003 | 96.7918 | 1.2424e-003 | 96.4168 | 1.1033e-003 | 96.7591 | 8.3202e-004 |
| 45 | 96.9095 | 1.0603e-003 | 96.8655 | 1.0399e-003 | 96.7918 | 1.2424e-003 | 96.4168 | 1.1033e-003 | 96.7591 | 8.3202e-004 |
| 46 | 96.9095 | 1.0603e-003 | 96.8655 | 1.0399e-003 | 96.7918 | 1.2424e-003 | 96.4168 | 1.1033e-003 | 96.7591 | 8.3202e-004 |
| 47 | 96.9095 | 1.0603e-003 | 96.8655 | 1.0399e-003 | 96.7918 | 1.2424e-003 | 96.4168 | 1.1033e-003 | 96.7591 | 8.3202e-004 |
| 48 | 96.9095 | 1.0603e-003 | 96.8655 | 1.0399e-003 | 96.7918 | 1.2424e-003 | 96.4168 | 1.1033e-003 | 96.7591 | 8.3202e-004 |
| 49 | 96.9095 | 1.0603e-003 | 96.8655 | 1.0399e-003 | 96.7918 | 1.2424e-003 | 96.4168 | 1.1033e-003 | 96.7591 | 8.3202e-004 |
| stat. #trials | 200 | | 200 | | 200 | | 200 | | 200 | |

Table 31: Database: UCF, model: SPRT-TANDEM (2/2)

| Frame | 10th Mean | 10th SEM | 14th Mean | 14th SEM | 24th Mean | 24th SEM | 49th Mean | 49th SEM |
|---|---|---|---|---|---|---|---|---|
| 2 | 94.3693 | 7.0583e-004 | 94.5973 | 6.3592e-004 | 94.4741 | 6.5887e-004 | 94.5164 | 5.4415e-004 |
| 3 | 94.2933 | 9.2496e-004 | 94.4836 | 7.6021e-004 | 94.6227 | 7.1819e-004 | 94.4027 | 5.6738e-004 |
| 4 | 94.7670 | 1.2622e-003 | 94.6000 | 8.4566e-004 | 94.7445 | 8.5464e-004 | 94.3614 | 6.3765e-004 |
| 5 | 95.1009 | 1.2382e-003 | 94.7959 | 9.7762e-004 | 94.9823 | 1.0140e-003 | 94.5136 | 7.4998e-004 |
| 6 | 95.4595 | 1.2265e-003 | 95.0105 | 1.0293e-003 | 95.0127 | 1.0180e-003 | 94.6668 | 8.6725e-004 |
| 7 | 95.7621 | 1.1532e-003 | 95.1614 | 1.0559e-003 | 95.1064 | 1.0513e-003 | 95.0105 | 9.9242e-004 |
| 8 | 95.9233 | 1.2003e-003 | 95.3300 | 1.0940e-003 | 95.1927 | 1.0602e-003 | 95.3632 | 1.0896e-003 |
| 9 | 95.9844 | 1.1858e-003 | 95.5682 | 1.1080e-003 | 95.4273 | 1.0599e-003 | 95.7618 | 1.1041e-003 |
| 10 | 96.1847 | 1.1972e-003 | 95.8186 | 1.1270e-003 | 95.8045 | 9.8347e-004 | 96.2027 | 1.0252e-003 |
| 11 | 96.3864 | 1.1868e-003 | 96.0300 | 1.0826e-003 | 96.1218 | 9.5190e-004 | 96.5682 | 9.4495e-004 |
| 12 | 96.5376 | 1.1523e-003 | 96.3145 | 1.0642e-003 | 96.4105 | 9.7122e-004 | 96.8759 | 9.6024e-004 |
| 13 | 96.6683 | 1.1441e-003 | 96.4641 | 9.9651e-004 | 96.5586 | 9.6887e-004 | 96.8882 | 9.2910e-004 |
| 14 | 96.7571 | 1.1529e-003 | 96.5886 | 9.7034e-004 | 96.6436 | 9.3090e-004 | 97.0291 | 8.5486e-004 |
| 15 | 96.8494 | 1.1567e-003 | 96.6045 | 9.7399e-004 | 96.7273 | 8.7228e-004 | 97.0350 | 8.5588e-004 |
| 16 | 96.8558 | 1.1123e-003 | 96.6395 | 9.5199e-004 | 96.7314 | 8.5885e-004 | 97.0091 | 8.5215e-004 |
| 17 | 96.8565 | 1.0894e-003 | 96.6732 | 9.3998e-004 | 96.7618 | 8.5201e-004 | 96.9873 | 8.6177e-004 |
| 18 | 96.8636 | 1.0654e-003 | 96.7214 | 9.3324e-004 | 96.7486 | 8.3516e-004 | 96.9891 | 8.4492e-004 |
| 19 | 96.9304 | 1.0291e-003 | 96.7586 | 9.4147e-004 | 96.7814 | 8.1684e-004 | 96.9605 | 8.3036e-004 |
| 20 | 96.9901 | 1.0344e-003 | 96.7764 | 9.3966e-004 | 96.8059 | 7.9474e-004 | 97.0177 | 8.1004e-004 |
| 21 | 97.0405 | 1.0043e-003 | 96.7945 | 9.3204e-004 | 96.8295 | 7.8640e-004 | 97.0082 | 8.1590e-004 |
| 22 | 97.0767 | 1.0043e-003 | 96.8314 | 9.1138e-004 | 96.8245 | 7.9476e-004 | 97.0186 | 7.8815e-004 |
| 23 | 97.0746 | 4.0004e-005 | 96.8455 | 9.0550e-004 | 96.8477 | 7.9762e-004 | 96.9991 | 7.9405e-004 |
| 24 | 97.1179 | 9.5514e-004 | 96.8350 | 9.1151e-004 | 96.8373 | 8.0698e-004 | 96.9886 | 7.9391e-004 |
| 25 | 97.1243 | 9.6886e-004 | 96.8677 | 8.9903e-004 | 96.8600 | 8.1298e-004 | 96.9591 | 7.8125e-004 |
| 26 | 97.1385 | 9.6524e-004 | 96.8864 | 9.0036e-004 | 96.8605 | 8.0449e-004 | 96.9377 | 8.0226e-004 |
| 27 | 97.1740 | 9.6070e-004 | 96.8945 | 9.0864e-004 | 96.8595 | 8.3259e-004 | 96.9423 | 8.1041e-004 |
| 28 | 97.1882 | 9.6176e-004 | 96.9182 | 9.0371e-004 | 96.8600 | 8.3312e-004 | 96.9036 | 8.2951e-004 |
| 29 | 97.2024 | 9.5737e-004 | 96.9132 | 9.0583e-004 | 96.8441 | 8.4849e-004 | 96.8595 | 8.3695e-004 |
| 30 | 97.2173 | 9.5695e-004 | 96.9268 | 8.9741e-004 | 96.8191 | 8.6376e-004 | 96.8150 | 8.6132e-004 |
| 31 | 97.2386 | 9.5507e-004 | 96.9414 | 8.8623e-004 | 96.7941 | 8.8176e-004 | 96.8150 | 8.5573e-004 |
| 32 | 97.2386 | 9.5507e-004 | 96.9595 | 8.9188e-004 | 96.7986 | 8.7948e-004 | 96.8045 | 8.5903e-004 |
| 33 | 97.2386 | 9.5507e-004 | 96.9595 | 8.9188e-004 | 96.7936 | 8.8576e-004 | 96.7895 | 8.6692e-004 |
| 34 | 97.2386 | 9.5507e-004 | 96.9595 | 8.9188e-004 | 96.7986 | 8.8208e-004 | 96.7595 | 8.9009e-004 |
| 35 | 97.2457 | 9.5259e-004 | 96.9545 | 8.9598e-004 | 96.8036 | 8.7836e-004 | 96.7445 | 8.9987e-004 |
| 36 | 97.2457 | 9.5259e-004 | 96.9495 | 8.9724e-004 | 96.8036 | 8.7836e-004 | 96.7395 | 9.0308e-004 |
| 37 | 97.2457 | 9.5259e-004 | 96.9495 | 8.9724e-004 | 96.8036 | 8.7836e-004 | 96.7250 | 9.1193e-004 |
| 38 | 97.2457 | 9.5259e-004 | 96.9495 | 8.9724e-004 | 96.8036 | 8.7836e-004 | 96.7250 | 9.1193e-004 |
| 39 | 97.2457 | 9.5259e-004 | 96.9495 | 8.9724e-004 | 96.8036 | 8.7836e-004 | 96.7250 | 9.1193e-004 |
| 40 | 97.2457 | 9.5259e-004 | 96.9495 | 8.9724e-004 | 96.7986 | 8.8208e-004 | 96.7250 | 9.1193e-004 |
| 41 | 97.2457 | 9.5259e-004 | 96.9495 | 8.9724e-004 | 96.7986 | 8.8208e-004 | 96.7205 | 9.0941e-004 |
| 42 | 97.2457 | 9.5259e-004 | 96.9495 | 8.9724e-004 | 96.7936 | 8.8292e-004 | 96.7155 | 9.1751e-004 |
| 43 | 97.2457 | 9.5259e-004 | 96.9495 | 8.9724e-004 | 96.7936 | 8.8292e-004 | 96.7155 | 9.1751e-004 |
| 44 | 97.2457 | 9.5259e-004 | 96.9495 | 8.9724e-004 | 96.7936 | 8.8292e-004 | 96.7155 | 9.1751e-004 |
| 45 | 97.2457 | 9.5259e-004 | 96.9495 | 8.9724e-004 | 96.7936 | 8.8292e-004 | 96.7155 | 9.1751e-004 |
| 46 | 97.2457 | 9.5259e-004 | 96.9495 | 8.9724e-004 | 96.7936 | 8.8292e-004 | 96.7155 | 9.1751e-004 |
| 47 | 97.2457 | 9.5259e-004 | 96.9495 | 8.9724e-004 | 96.7936 | 8.8292e-004 | 96.7155 | 9.1751e-004 |
| 48 | 97.2457 | 9.5259e-004 | 96.9495 | 8.9724e-004 | 96.7936 | 8.8292e-004 | 96.7155 | 9.1751e-004 |
| 49 | 97.2457 | 9.5259e-004 | 96.9495 | 8.9724e-004 | 96.7891 | 8.8285e-004 | 96.7155 | 9.1751e-004 |
| stat. #trials | 228 | | 200 | | 200 | | 200 | |

Table 32: Database: UCF, model: LSTM-m/s

| | LSTM-m | | LSTM-s | |
|---|---|---|---|---|
| Frame | Mean | SEM | Mean | SEM |
| 1 | 92.3218 | 3.4054e-003 | 84.2160 | 1.1426e-002 |
| 2 | 93.1418 | 2.5895e-003 | 90.8731 | 4.1958e-003 |
| 3 | 93.5945 | 2.1883e-003 | 92.3609 | 2.9801e-003 |
| 4 | 93.2318 | 2.0714e-003 | 92.8218 | 2.5565e-003 |
| 5 | 93.3100 | 1.9444e-003 | 93.1665 | 2.2873e-003 |
| 6 | 93.1673 | 1.9889e-003 | 92.8992 | 2.1601e-003 |
| 7 | 93.0718 | 1.9675e-003 | 92.7300 | 1.9121e-003 |
| 8 | 93.1936 | 1.8432e-003 | 92.8425 | 1.8468e-003 |
| 9 | 93.8773 | 1.8754e-003 | 93.1998 | 1.8342e-003 |
| 10 | 94.3245 | 1.8930e-003 | 93.7480 | 1.7591e-003 |
| 11 | 94.4927 | 1.9408e-003 | 93.8524 | 1.6778e-003 |
| 12 | 94.7718 | 1.9919e-003 | 94.0666 | 1.6810e-003 |
| 13 | 95.0364 | 1.8573e-003 | 94.3366 | 1.8066e-003 |
| 14 | 94.6818 | 1.9646e-003 | 94.2475 | 1.7542e-003 |
| 15 | 94.5936 | 1.9803e-003 | 94.2277 | 1.6357e-003 |
| 16 | 94.5455 | 1.8724e-003 | 94.2187 | 1.6134e-003 |
| 17 | 94.8809 | 1.8213e-003 | 94.5104 | 1.6086e-003 |
| 18 | 95.3664 | 1.7234e-003 | 94.8128 | 1.6269e-003 |
| 19 | 95.5745 | 1.6877e-003 | 95.0261 | 1.4468e-003 |
| 20 | 95.7255 | 1.6622e-003 | 95.4770 | 1.4714e-003 |
| 21 | 95.7536 | 1.6173e-003 | 95.7462 | 1.4674e-003 |
| 22 | 95.8109 | 1.6107e-003 | 95.8596 | 1.4104e-003 |
| 23 | 95.8236 | 1.6813e-003 | 95.8812 | 1.4703e-003 |
| 24 | 95.900 | 1.6703e-003 | 95.8686 | 1.4755e-003 |
| 25 | 95.9273 | 1.6787e-003 | 95.9253 | 1.4515e-003 |
| 26 | 95.9500 | 1.7337e-003 | 95.9460 | 1.4797e-003 |
| 27 | 95.9364 | 1.7446e-003 | 95.9406 | 1.4437e-003 |
| 28 | 95.9536 | 1.7631e-003 | 95.9586 | 1.4707e-003 |
| 29 | 95.9391 | 1.7810e-003 | 95.9550 | 1.4902e-003 |
| 30 | 96.0027 | 1.7433e-003 | 95.9730 | 1.4463e-003 |
| 31 | 96.0209 | 1.7723e-003 | 96.0414 | 1.5190e-003 |
| 32 | 96.0964 | 1.7272e-003 | 96.0567 | 1.5807e-003 |
| 33 | 96.0727 | 1.8089e-003 | 96.0468 | 1.5618e-003 |
| 34 | 96.1527 | 1.8490e-003 | 96.1134 | 1.6079e-003 |
| 35 | 96.3518 | 1.9190e-003 | 96.2592 | 1.6641e-003 |
| 36 | 96.3991 | 1.7732e-003 | 96.2313 | 1.5489e-003 |
| 37 | 96.3882 | 1.7635e-003 | 96.3762 | 1.4489e-003 |
| 38 | 96.3973 | 1.7492e-003 | 96.4833 | 1.3558e-003 |
| 39 | 96.3655 | 1.7172e-003 | 96.5203 | 1.3800e-003 |
| 40 | 96.3836 | 1.6966e-003 | 96.5005 | 1.3618e-003 |
| 41 | 96.3727 | 1.6963e-003 | 96.5410 | 1.3637e-003 |
| 42 | 96.4718 | 1.6655e-003 | 96.5734 | 1.2837e-003 |
| 43 | 96.5882 | 1.6877e-003 | 96.6013 | 1.3400e-003 |
| 44 | 96.5873 | 1.7095e-003 | 96.6094 | 1.2952e-003 |
| 45 | 96.6445 | 1.6614e-003 | 96.5986 | 1.3716e-003 |
| 46 | 96.7209 | 1.6584e-003 | 96.5500 | 1.3648e-003 |
| 47 | 96.6427 | 1.6560e-003 | 96.5032 | 1.4112e-003 |
| 48 | 96.6118 | 1.7179e-003 | 96.4329 | 1.4514e-003 |
| 49 | 96.6845 | 1.6819e-003 | 96.4545 | 1.5521e-003 |
| 50 | 96.6891 | 1.7006e-003 | 96.5149 | 1.5340e-003 |
| stat. #trials | 100 | | 101 | |

Table 33: Database: UCF, model: EARLIEST

| | Lambda: 1e-3 | | | Lambda: 1e-5 | |
|---|---|---|---|---|---|
| Mean hitting time | Mean | SEM | Mean hitting time | Mean | SEM |
| 2.0710 | 93.4600 | 1.2397e-003 | 2.0924 | 93.4800 | 8.5977e-004 |
| stat. #trials | 130 | | | 130 | |

Table 34: Database: UCF, model: 3DResNet

| | 15 frames | | 25 frames | |
|---|---|---|---|---|
| | Mean | SEM | Mean | SEM |
| | 64.4188 | 8.7905e-003 | 90.0827 | 4.9585e-003 |
| stat. #trials | 49 | | 100 | |

Table 35: Database: SiW, model: SPRT-TANDEM (1/2)

| Frame | 0th Mean | 0th SEM | 1st Mean | 1st SEM | 2nd Mean | 2nd SEM | 3rd Mean | 3rd SEM | 5th Mean | 5th SEM |
|---|---|---|---|---|---|---|---|---|---|---|
| 2 | 99.8243 | 6.3459e-006 | 99.8441 | 1.2108e-005 | 99.862 | 9.1680e-006 | 99.8677 | 7.4066e-006 | 99.8647 | 1.1755e-005 |
| 3 | 99.8482 | 8.7040e-006 | 99.8604 | 1.3893e-005 | 99.8781 | 8.4124e-006 | 99.8846 | 9.1821e-006 | 99.8782 | 1.0775e-005 |
| 4 | 99.8596 | 1.2285e-005 | 99.8671 | 1.3990e-005 | 99.8884 | 8.0382e-006 | 99.8904 | 7.5388e-006 | 99.8861 | 1.0707e-005 |
| 5 | 99.8665 | 1.0850e-005 | 99.8741 | 1.5640e-005 | 99.8918 | 7.2057e-006 | 99.8917 | 6.7430e-006 | 99.8880 | 1.0342e-005 |
| 6 | 99.8695 | 1.0821e-005 | 99.8797 | 1.6082e-005 | 99.8933 | 7.0796e-006 | 99.8920 | 6.5525e-006 | 99.8885 | 9.6929e-006 |
| 7 | 99.8710 | 1.1622e-005 | 99.8834 | 1.5900e-005 | 99.8939 | 7.3545e-006 | 99.8925 | 6.5869e-006 | 99.8888 | 9.2890e-006 |
| 8 | 99.8722 | 1.1225e-005 | 99.8841 | 1.5857e-005 | 99.8940 | 7.2435e-006 | 99.8930 | 6.5214e-006 | 99.8888 | 9.2371e-006 |
| 9 | 99.8734 | 1.0900e-005 | 99.8847 | 1.5688e-005 | 99.8944 | 7.0277e-006 | 99.8935 | 6.5158e-006 | 99.8890 | 9.1903e-006 |
| 10 | 99.8743 | 1.1475e-005 | 99.8859 | 1.5007e-005 | 99.8947 | 7.4926e-006 | 99.8938 | 6.5011e-006 | 99.8892 | 8.9494e-006 |
| 11 | 99.8745 | 1.1716e-005 | 99.8868 | 1.4293e-005 | 99.8954 | 7.4938e-006 | 99.8939 | 6.4358e-006 | 99.8893 | 9.1568e-006 |
| 12 | 99.8745 | 1.1710e-005 | 99.8880 | 1.3965e-005 | 99.8960 | 7.0799e-006 | 99.8940 | 6.4512e-006 | 99.8893 | 9.1771e-006 |
| 13 | 99.8745 | 1.1710e-005 | 99.8886 | 1.3493e-005 | 99.8961 | 7.0500e-006 | 99.8940 | 6.4760e-006 | 99.8894 | 9.1487e-006 |
| 14 | 99.8745 | 1.1710e-005 | 99.8888 | 1.3068e-005 | 99.8961 | 7.0445e-006 | 99.8940 | 6.4613e-006 | 99.8894 | 9.1073e-006 |
| 15 | 99.8745 | 1.1710e-005 | 99.8890 | 1.2824e-005 | 99.8961 | 6.9214e-006 | 99.8940 | 6.4501e-006 | 99.8893 | 9.1593e-006 |
| 16 | 99.8745 | 1.1710e-005 | 99.8892 | 1.2619e-005 | 99.8962 | 6.9156e-006 | 99.8940 | 6.4501e-006 | 99.8894 | 9.1099e-006 |
| 17 | 99.8745 | 1.1710e-005 | 99.8893 | 1.2774e-005 | 99.8962 | 6.9156e-006 | 99.8940 | 6.4501e-006 | 99.8894 | 9.0983e-006 |
| 18 | 99.8745 | 1.1710e-005 | 99.8893 | 1.2955e-005 | 99.8962 | 6.9156e-006 | 99.8940 | 6.4501e-006 | 99.8894 | 9.1055e-006 |
| 19 | 99.8745 | 1.1710e-005 | 99.8892 | 1.3248e-005 | 99.8962 | 6.9156e-006 | 99.8940 | 6.4501e-006 | 99.8894 | 9.1055e-006 |
| 20 | 99.8745 | 1.1710e-005 | 99.8892 | 1.3414e-005 | 99.8962 | 6.9156e-006 | 99.8940 | 6.5535e-006 | 99.8894 | 9.1055e-006 |
| 21 | 99.8745 | 1.1710e-005 | 99.8892 | 1.3440e-005 | 99.8962 | 6.9156e-006 | 99.8940 | 6.5535e-006 | 99.8894 | 9.1055e-006 |
| 22 | 99.8745 | 1.1710e-005 | 99.8891 | 1.3708e-005 | 99.8961 | 7.0169e-006 | 99.8940 | 6.5535e-006 | 99.8894 | 9.1055e-006 |
| 23 | 99.8745 | 1.1710e-005 | 99.8891 | 1.3804e-005 | 99.8961 | 7.0169e-006 | 99.8940 | 6.5535e-006 | 99.8894 | 9.1055e-006 |
| 24 | 99.8745 | 1.1710e-005 | 99.8890 | 1.3890e-005 | 99.8961 | 7.0169e-006 | 99.8940 | 6.5535e-006 | 99.8894 | 9.1055e-006 |
| 25 | 99.8745 | 1.1710e-005 | 99.8891 | 1.3882e-005 | 99.8961 | 7.0169e-006 | 99.8940 | 6.5535e-006 | 99.8893 | 9.0979e-006 |
| 26 | 99.8745 | 1.1710e-005 | 99.8891 | 1.3882e-005 | 99.8961 | 7.0169e-006 | 99.8940 | 6.5535e-006 | 99.8893 | 9.0979e-006 |
| 27 | 99.8745 | 1.1710e-005 | 99.8891 | 1.3877e-005 | 99.8961 | 7.0169e-006 | 99.8940 | 6.5535e-006 | 99.8893 | 9.0979e-006 |
| 28 | 99.8745 | 1.1710e-005 | 99.8891 | 1.3922e-005 | 99.8961 | 7.0169e-006 | 99.8940 | 6.5535e-006 | 99.8893 | 9.0979e-006 |
| 29 | 99.8745 | 1.1710e-005 | 99.8891 | 1.3922e-005 | 99.8961 | 7.0169e-006 | 99.8940 | 6.5535e-006 | 99.8893 | 9.0979e-006 |
| 30 | 99.8745 | 1.1710e-005 | 99.8891 | 1.3922e-005 | 99.8961 | 7.0169e-006 | 99.8940 | 6.5535e-006 | 99.8893 | 9.0979e-006 |
| 31 | 99.8745 | 1.1710e-005 | 99.8891 | 1.3922e-005 | 99.8961 | 7.0169e-006 | 99.8940 | 6.5535e-006 | 99.8893 | 9.0979e-006 |
| 32 | 99.8745 | 1.1710e-005 | 99.8891 | 1.3922e-005 | 99.8961 | 7.0169e-006 | 99.8940 | 6.5535e-006 | 99.8893 | 9.0979e-006 |
| 33 | 99.8745 | 1.1710e-005 | 99.8891 | 1.3922e-005 | 99.8961 | 7.0169e-006 | 99.8940 | 6.5535e-006 | 99.8893 | 9.0979e-006 |
| 34 | 99.8745 | 1.1710e-005 | 99.8891 | 1.3922e-005 | 99.8961 | 7.0169e-006 | 99.8940 | 6.5535e-006 | 99.8893 | 9.0979e-006 |
| 35 | 99.8745 | 1.1710e-005 | 99.8891 | 1.3922e-005 | 99.8961 | 7.0169e-006 | 99.8940 | 6.5535e-006 | 99.8893 | 9.0979e-006 |
| 36 | 99.8745 | 1.1710e-005 | 99.8891 | 1.3922e-005 | 99.8961 | 7.0169e-006 | 99.8940 | 6.5535e-006 | 99.8893 | 9.0979e-006 |
| 37 | 99.8745 | 1.1710e-005 | 99.8891 | 1.3922e-005 | 99.8961 | 7.0169e-006 | 99.8940 | 6.5535e-006 | 99.8893 | 9.0979e-006 |
| 38 | 99.8745 | 1.1710e-005 | 99.8891 | 1.3922e-005 | 99.8961 | 7.0169e-006 | 99.8940 | 6.5535e-006 | 99.8893 | 9.0979e-006 |
| 39 | 99.8745 | 1.1710e-005 | 99.8891 | 1.3922e-005 | 99.8961 | 7.0169e-006 | 99.8940 | 6.5535e-006 | 99.8893 | 9.0979e-006 |
| 40 | 99.8745 | 1.1710e-005 | 99.8891 | 1.3922e-005 | 99.8961 | 7.0169e-006 | 99.8940 | 6.5535e-006 | 99.8893 | 9.0979e-006 |
| 41 | 99.8745 | 1.1710e-005 | 99.8891 | 1.3922e-005 | 99.8961 | 7.0169e-006 | 99.8940 | 6.5535e-006 | 99.8893 | 9.0979e-006 |
| 42 | 99.8745 | 1.1710e-005 | 99.8891 | 1.3922e-005 | 99.8961 | 7.0169e-006 | 99.8940 | 6.5535e-006 | 99.8893 | 9.0979e-006 |
| 43 | 99.8745 | 1.1710e-005 | 99.8891 | 1.3922e-005 | 99.8961 | 7.0169e-006 | 99.8940 | 6.5535e-006 | 99.8893 | 9.0979e-006 |
| 44 | 99.8745 | 1.1710e-005 | 99.8891 | 1.3922e-005 | 99.8961 | 7.0169e-006 | 99.8940 | 6.5535e-006 | 99.8893 | 9.0979e-006 |
| 45 | 99.8745 | 1.1710e-005 | 99.8891 | 1.3922e-005 | 99.8961 | 7.0169e-006 | 99.8940 | 6.5535e-006 | 99.8893 | 9.0979e-006 |
| 46 | 99.8745 | 1.1710e-005 | 99.8891 | 1.3922e-005 | 99.8961 | 7.0169e-006 | 99.8940 | 6.5535e-006 | 99.8893 | 9.0979e-006 |
| 47 | 99.8745 | 1.1710e-005 | 99.75 | 1.3222e-005 | 99.8961 | 7.0169e-006 | 99.8940 | 6.5535e-006 | 99.8893 | 9.0979e-006 |
| 48 | N.A. | N.A. | N.A. | N.A. | 99.8961 | 7.0169e-006 | 99.8940 | 6.5535e-006 | 99.8893 | 9.0979e-006 |
| 49 | N.A. | N.A. | N.A. | N.A. | 99.8961 | 7.0169e-006 | 99.8940 | 6.5535e-006 | 99.8893 | 9.0979e-006 |
| stat. #trials | 116 | | 112 | | 110 | | 109 | | 109 | |

Table 36: Database: SiW, model: SPRT-TANDEM (2/2)

| Frame | 10th Mean | 10th SEM | 14th Mean | 14th SEM | 24th Mean | 24th SEM | 49th Mean | 49th SEM |
|---|---|---|---|---|---|---|---|---|
| 2 | 99.8726 | 1.3614e-005 | 99.8706 | 1.7393e-005 | 99.8749 | 8.6377e-006 | 99.8774 | 1.2239e-005 |
| 3 | 99.8818 | 1.2983e-005 | 99.8762 | 1.7440e-005 | 99.8781 | 8.4636e-006 | 99.8780 | 1.2553e-005 |
| 4 | 99.8840 | 1.3748e-005 | 99.8772 | 1.7972e-005 | 99.8787 | 8.8320e-006 | 99.8784 | 1.1964e-005 |
| 5 | 99.8829 | 1.2982e-005 | 99.8770 | 1.7717e-005 | 99.8789 | 9.4704e-006 | 99.8788 | 1.1849e-005 |
| 6 | 99.8827 | 1.3123e-005 | 99.8766 | 1.7587e-005 | 99.8789 | 9.9605e-006 | 99.8790 | 1.1873e-005 |
| 7 | 99.8826 | 1.3214e-005 | 99.8769 | 1.7567e-005 | 99.8791 | 1.0263e-005 | 99.8791 | 1.1747e-005 |
| 8 | 99.8826 | 1.3378e-005 | 99.8770 | 1.7595e-005 | 99.8792 | 1.0219e-005 | 99.8793 | 1.1696e-005 |
| 9 | 99.8826 | 1.3489e-005 | 99.8770 | 1.7693e-005 | 99.8793 | 1.0189e-005 | 99.8794 | 1.1529e-005 |
| 10 | 99.8826 | 1.3590e-005 | 99.8768 | 1.7846e-005 | 99.8793 | 1.0293e-005 | 99.8795 | 1.1611e-005 |
| 11 | 99.8827 | 1.3508e-005 | 99.8767 | 1.7933e-005 | 99.8793 | 1.0553e-005 | 99.8796 | 1.1370e-005 |
| 12 | 99.8828 | 1.3459e-005 | 99.8768 | 1.8085e-005 | 99.8794 | 1.0665e-005 | 99.8796 | 1.1568e-005 |
| 13 | 99.8828 | 1.3483e-005 | 99.8768 | 1.8005e-005 | 99.8792 | 1.0980e-005 | 99.8797 | 1.1657e-005 |
| 14 | 99.8829 | 1.3567e-005 | 99.8768 | 1.8001e-005 | 99.8794 | 1.1115e-005 | 99.8797 | 1.1590e-005 |
| 15 | 99.8830 | 1.3633e-005 | 99.8769 | 1.7941e-005 | 99.8795 | 1.1157e-005 | 99.8799 | 1.1677e-005 |
| 16 | 99.8831 | 1.3554e-005 | 99.8770 | 1.7929e-005 | 99.8795 | 1.1151e-005 | 99.8800 | 1.1644e-005 |
| 17 | 99.8833 | 1.3680e-005 | 99.8771 | 1.7919e-005 | 99.8796 | 1.1150e-005 | 99.8801 | 1.1814e-005 |
| 18 | 99.8837 | 1.3756e-005 | 99.8771 | 1.7914e-005 | 99.8796 | 1.1183e-005 | 99.8804 | 1.1772e-005 |
| 19 | 99.8840 | 1.3827e-005 | 99.8771 | 1.7910e-005 | 99.8797 | 1.1165e-005 | 99.8805 | 1.1870e-005 |
| 20 | 99.8842 | 1.4034e-005 | 99.8771 | 1.8036e-005 | 99.8797 | 1.1265e-005 | 99.8806 | 1.1850e-005 |
| 21 | 99.8846 | 1.4100e-005 | 99.8771 | 1.8068e-005 | 99.8797 | 1.1437e-005 | 99.8805 | 1.1870e-005 |
| 22 | 99.8850 | 1.4170e-005 | 99.8772 | 1.8052e-005 | 99.8796 | 1.1560e-005 | 99.8806 | 1.2015e-005 |
| 23 | 99.8853 | 1.4267e-005 | 99.8772 | 1.8090e-005 | 99.8796 | 1.1647e-005 | 99.8805 | 1.2147e-005 |
| 24 | 99.8854 | 1.4274e-005 | 99.8775 | 1.8268e-005 | 99.8797 | 1.1655e-005 | 99.8806 | 1.2093e-005 |
| 25 | 99.8856 | 1.4233e-005 | 99.8777 | 1.8324e-005 | 99.8798 | 1.1665e-005 | 99.8806 | 1.2077e-005 |
| 26 | 99.8858 | 1.4249e-005 | 99.8780 | 1.8502e-005 | 99.8798 | 1.1682e-005 | 99.8807 | 1.2167e-005 |
| 27 | 99.8859 | 1.4328e-005 | 99.8783 | 1.8562e-005 | 99.8799 | 1.1696e-005 | 99.8807 | 1.2167e-005 |
| 28 | 99.8860 | 1.4389e-005 | 99.8785 | 1.8597e-005 | 99.8799 | 1.1685e-005 | 99.8807 | 1.2174e-005 |
| 29 | 99.8860 | 1.4422e-005 | 99.8790 | 1.8702e-005 | 99.8801 | 1.1653e-005 | 99.8814 | 1.2269e-005 |
| 30 | 99.8861 | 1.4459e-005 | 99.8797 | 1.9037e-005 | 99.8808 | 1.2218e-005 | 99.8805 | 1.2158e-005 |
| 31 | 99.8860 | 1.4525e-005 | 99.8796 | 1.9993e-005 | 99.8799 | 1.3022e-005 | 99.8812 | 1.2066e-005 |
| 32 | 99.8859 | 1.4595e-005 | 99.8799 | 2.0144e-005 | 99.8805 | 1.3221e-005 | 99.8817 | 1.1924e-005 |
| 33 | 99.8859 | 1.4610e-005 | 99.8798 | 1.9976e-005 | 99.8810 | 1.3483e-005 | 99.8819 | 1.1909e-005 |
| 34 | 99.8858 | 1.4752e-005 | 99.8798 | 1.9963e-005 | 99.8811 | 1.3512e-005 | 99.8821 | 1.1898e-005 |
| 35 | 99.8858 | 1.4743e-005 | 99.8799 | 1.9998e-005 | 99.8813 | 1.3567e-005 | 99.8821 | 1.1915e-005 |
| 36 | 99.8858 | 1.4757e-005 | 99.8802 | 2.0120e-005 | 99.8814 | 1.3630e-005 | 99.8821 | 1.1899e-005 |
| 37 | 99.8858 | 1.4749e-005 | 99.8802 | 2.0154e-005 | 99.8815 | 1.3667e-005 | 99.8822 | 1.1883e-005 |
| 38 | 99.8858 | 1.4749e-005 | 99.8804 | 2.0233e-005 | 99.8816 | 1.3677e-005 | 99.8822 | 1.1828e-005 |
| 39 | 99.8858 | 1.4732e-005 | 99.8805 | 2.0255e-005 | 99.8816 | 1.3695e-005 | 99.8822 | 1.1830e-005 |
| 40 | 99.8858 | 1.4713e-005 | 99.8805 | 2.0155e-005 | 99.8818 | 1.3735e-005 | 99.8824 | 1.1719e-005 |
| 41 | 99.8857 | 1.4698e-005 | 99.8803 | 2.0054e-005 | 99.8820 | 1.3742e-005 | 99.8825 | 1.1632e-005 |
| 42 | 99.8856 | 1.4673e-005 | 99.8803 | 2.0004e-005 | 99.8821 | 1.3765e-005 | 99.8825 | 1.1632e-005 |
| 43 | 99.8856 | 1.4673e-005 | 99.8802 | 1.9986e-005 | 99.8822 | 1.3852e-005 | 99.8825 | 1.1632e-005 |
| 44 | 99.8856 | 1.4625e-005 | 99.8802 | 1.9992e-005 | 99.8825 | 1.3955e-005 | 99.8825 | 1.1632e-005 |
| 45 | 99.8856 | 1.4625e-005 | 99.8801 | 1.9928e-005 | 99.8825 | 1.3976e-005 | 99.8825 | 1.1632e-005 |
| 46 | 99.8856 | 1.4625e-005 | 99.8801 | 1.9904e-005 | 99.8826 | 1.3984e-005 | 99.8825 | 1.1632e-005 |
| 47 | 99.8856 | 1.4625e-005 | 99.8801 | 1.9904e-005 | 99.8826 | 1.4013e-005 | 99.8825 | 1.1632e-005 |
| 48 | 99.8856 | 1.4625e-005 | 99.8801 | 1.9892e-005 | 99.8826 | 1.4013e-005 | 99.8825 | 1.1632e-005 |
| 49 | 99.8856 | 1.4625e-005 | 99.8801 | 1.9892e-005 | 99.8826 | 1.4013e-005 | 99.8825 | 1.1632e-005 |
| stat. #trials | 107 | | 108 | | 107 | | 73 | |

Table 37: Database: SiW, model: LSTM-m/s

| | LSTM-m | | LSTM-s | |
|---|---|---|---|---|
| Frame | Mean | SEM | Mean | SEM |
| 1 | 99.6207 | 7.6320e-005 | 99.6207 | 7.6320e-005 |
| 2 | 99.7677 | 2.2591e-005 | 99.7677 | 2.2591e-005 |
| 3 | 99.8044 | 1.1279e-005 | 99.8044 | 1.1279e-005 |
| 4 | 99.8089 | 6.0939e-006 | 99.8089 | 6.0939e-006 |
| 5 | 99.8029 | 8.5723e-006 | 99.8029 | 8.5723e-006 |
| 6 | 99.7969 | 1.2098e-005 | 99.7969 | 1.2098e-005 |
| 7 | 99.7951 | 1.3207e-005 | 99.7951 | 1.3207e-005 |
| 8 | 99.8003 | 1.1980e-005 | 99.8003 | 1.1980e-005 |
| 9 | 99.8031 | 1.3397e-005 | 99.8031 | 1.3397e-005 |
| 10 | 99.8057 | 1.4686e-005 | 99.8057 | 1.4686e-005 |
| 11 | 99.8092 | 1.6821e-005 | 99.8092 | 1.6821e-005 |
| 12 | 99.8159 | 1.8133e-005 | 99.8159 | 1.8133e-005 |
| 13 | 99.8196 | 1.8401e-005 | 99.8196 | 1.8401e-005 |
| 14 | 99.8241 | 1.7274e-005 | 99.8241 | 1.7274e-005 |
| 15 | 99.8264 | 1.5607e-005 | 99.8264 | 1.5607e-005 |
| 16 | 99.8290 | 1.4499e-005 | 99.8290 | 1.4499e-005 |
| 17 | 99.8316 | 1.6679e-005 | 99.8316 | 1.6679e-005 |
| 18 | 99.8335 | 1.6096e-005 | 99.8335 | 1.6096e-005 |
| 19 | 99.8374 | 1.8177e-005 | 99.8374 | 1.8177e-005 |
| 20 | 99.8383 | 1.8938e-005 | 99.8383 | 1.8938e-005 |
| 21 | 99.8391 | 1.7974e-005 | 99.8391 | 1.7974e-005 |
| 22 | 99.8405 | 1.7900e-005 | 99.8405 | 1.7900e-005 |
| 23 | 99.8422 | 1.7473e-005 | 99.8422 | 1.7473e-005 |
| 24 | 99.8449 | 1.8326e-005 | 99.8449 | 1.8326e-005 |
| 25 | 99.8474 | 1.8391e-005 | 99.8474 | 1.8391e-005 |
| 26 | 99.8497 | 1.9941e-005 | 99.8497 | 1.9941e-005 |
| 27 | 99.8525 | 2.0347e-005 | 99.8525 | 2.0347e-005 |
| 28 | 99.8540 | 1.9410e-005 | 99.8540 | 1.9410e-005 |
| 29 | 99.8550 | 1.9969e-005 | 99.8550 | 1.9969e-005 |
| 30 | 99.8562 | 1.9671e-005 | 99.8562 | 1.9671e-005 |
| 31 | 99.8571 | 1.9252e-005 | 99.8571 | 1.9252e-005 |
| 32 | 99.8578 | 1.9140e-005 | 99.8578 | 1.9140e-005 |
| 33 | 99.8600 | 1.8532e-005 | 99.8600 | 1.8532e-005 |
| 34 | 99.8627 | 2.0120e-005 | 99.8627 | 2.0120e-005 |
| 35 | 99.8634 | 2.1863e-005 | 99.8634 | 2.1863e-005 |
| 36 | 99.8642 | 2.2477e-005 | 99.8642 | 2.2477e-005 |
| 37 | 99.8663 | 2.2142e-005 | 99.8663 | 2.2142e-005 |
| 38 | 99.8675 | 2.1921e-005 | 99.8675 | 2.1921e-005 |
| 39 | 99.8675 | 2.1091e-005 | 99.8675 | 2.1091e-005 |
| 40 | 99.8679 | 2.0355e-005 | 99.8679 | 2.0355e-005 |
| 41 | 99.8683 | 2.0592e-005 | 99.8683 | 2.0592e-005 |
| 42 | 99.8692 | 2.1503e-005 | 99.8692 | 2.1503e-005 |
| 43 | 99.8705 | 2.1348e-005 | 99.8705 | 2.1348e-005 |
| 44 | 99.8726 | 2.1915e-005 | 99.8726 | 2.1915e-005 |
| 45 | 99.8748 | 2.3104e-005 | 99.8748 | 2.3104e-005 |
| 46 | 99.8754 | 2.1851e-005 | 99.8754 | 2.1851e-005 |
| 47 | 99.8759 | 2.1820e-005 | 99.8759 | 2.1820e-005 |
| 48 | 99.8764 | 2.1974e-005 | 99.8764 | 2.1974e-005 |
| 49 | 99.8780 | 2.2132e-005 | 99.8780 | 2.2132e-005 |
| 50 | 99.8782 | 2.0800e-005 | 99.8782 | 2.0800e-005 |
| stat. #trials | 63 | | 58 | |

Table 38: Database: SiW, model: EARLIEST

| | Lambda: 1e-3 | | | Lambda: 1e-5 | | | Lambda: 1e-10 | | |
|---|---|---|---|---|---|---|---|---|---|
| | Mean hitting time | Mean | SEM | Mean hitting time | Mean | SEM | Mean hitting time | Mean | SEM |
| | 1.1924 | 99.7156 | 4.4309e-005 | 8.2087 | 99.7753 | 2.9947e-005 | 32.0550 | 99.6823 | 4.5947e-005 |
| stat. #trials | 30 | | | 17 | | | 2 | | |

Table 39: Database: SiW, model: 3DResNet

| Frame | Mean | SEM |
|---|---|---|
| 5 | 98.8160 | 1.6762e-003 |
| 15 | 98.9740 | 1.1066e-003 |
| 25 | 98.5560 | 1.1066e-003 |

Table 40: Database: SiW, model: 3DResNet

| | 5 frames | | 15 frames | | 25 frames | |
|---|---|---|---|---|---|---|
| | Mean | SEM | Mean | SEM | Mean | SEM |
| | 98.8160 | 1.6762e-003 | 98.9740 | 1.1066e-003 | 98.5560 | 1.1066e-003 |
| stat. #trials | 5 | | 5 | | 5 | |

## K    COMPUTING INFRASTRUCTURE

All the experiments are conducted with custom python scripts running on NVIDIA GeForce RTX 2080 Ti, GTX 1080 Ti, or GTX 1080 graphics card. Numpy (Harris et al. (2020)) and Scipy (Virtanen et al. (2020)) are used for mathematical computations. We use Tensorflow 2.0.0 (Abadi et al. (2015)) as a machine learning framework except when running baseline algorithms that are implemented with PyTorch (Paszke et al. (2019)).

## L   AN EXAMPLE VIDEO OF THE NOSAIC MNIST DATABASE.

As we described in Section 5, the Nosaic (Noise + mOSAIC) MNIST, NMNIST for short, contains videos with 20 frames of MNIST handwritten digits, buried with noise at the first frame, gradually denoised toward the last frame. The first frame has all 255-valued pixels (white) except only 40 masks of pixels that are randomly selected to reveal the original image. Another forty pixels are randomly selected at each of the next timestamps, finally revealing the original image at the last, 20th frame. An example video is shown in Figure 13.



Figure 13: Nosaic MNIST (NMNIST) database consists of videos of 20 frames, each of which has $28 \times 28 \times 1$ pixels. The frames are buried with noise at the first frame, gradually denoised toward the last frame. NMNIST provides a typical task in early classification of time series.

# M  SUPPLEMENTARY EXPERIMENT ON MOVING MNIST DATABASE

Prior to the experiment on Nosaic MNIST, we conducted a preliminary experiment on the Moving MNIST (MMNIST) database. 1st, 2nd, 3rd, and 5th-order SPRT-TANDEM were compared to the LSTM-m. Hyperparameters of each model were independently optimized with Optuna. The result plotted in Figure 14 showed that the balanced accuracy of the SPRT-TANDEM peaked and reached the plateau phase only after two or three frames. This indicated that each of the frames in MMNIST contained too much information so that a well-trained classifier could classify a video easily. Thus, although our SPRT-TANDEM outperformed LSTM-m with a large margin, we decided to design the original database, Nosaic MNNIST (NMNIST) for the early-classification task. NMNIST contains videos with noise-buried handwritten digits, gradually denoised towards the end of the videos, increasing mean hitting time compared to the MMNIST.

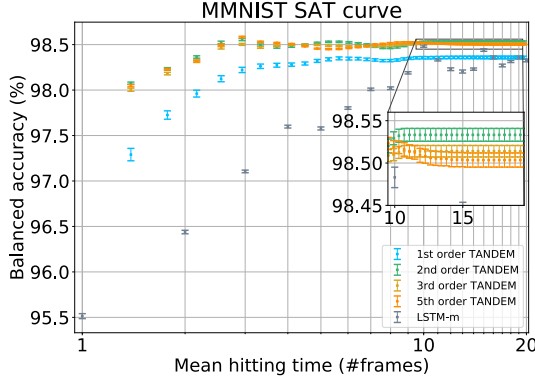

Figure 14: Speed-accuracy tradeoff (SAT) curves of the Moving MNIST database. Error bars show the standard error of the mean (SEM).

# N  SUPPLEMENTARY ABLATION EXPERIMENT

In addition to the ablation experiment presented in Figure 3e, which is calculated with 1st-order SPRT-TANDEM, we also conduct an experiment with 19th-order SPRT-TANDEM. The result shown in Figure 15 is qualitatively in line with Figure 3e: the $L_{\mathrm{multiplet}}$ has an advantage at the early phase with a few data samples, while the $L_{\mathrm{LLR}}$ leads to the higher final balanced accuracy at the late phase, and using both loss functions the best SAT curves can be obtained.

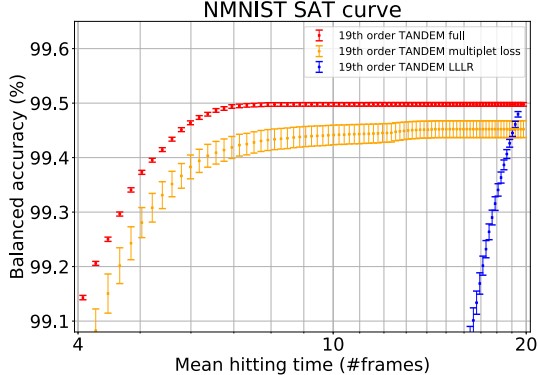

Figure 15: Speed-accuracy tradeoff (SAT) curves of the ablation experiment with the 19th-order SPRT-TANDEM on NMNIST database. Error bars show the standard error of the mean (SEM).

## SUPPLEMENTARY REFERENCES

M. Abadi, A. Agarwal, P. Barham, E. Brevdo, Z. Chen, C. Citro, G. S. Corrado, A. Davis, J. Dean, M. Devin, S. Ghemawat, I. Goodfellow, A. Harp, G. Irving, M. Isard, Y. Jia, R. Jozefowicz, L. Kaiser, M. Kudlur, J. Levenberg, D. Mané, R. Monga, S. Moore, D. Murray, C. Olah, M. Schuster, J. Shlens, B. Steiner, I. Sutskever, K. Talwar, P. Tucker, V. Vanhoucke, V. Vasudevan, F. Viégas, O. Vinyals, P. Warden, M. Wattenberg, M. Wicke, Y. Yu, and X. Zheng. TensorFlow: Large-scale machine learning on heterogeneous systems, 2015. Software available from tensorflow.org.

P. Armitage. Sequential analysis with more than two alternative hypotheses, and its relation to discriminant function analysis. *Journal of the Royal Statistical Society. Series B (Methodological)*, 12(1):137–144, 1950.

F. Ashby. A biased random walk model for two choice reaction times. *Journal of Mathematical Psychology*, 27:277–297, 1983.

A. Bagnall, J. Lines, J. Hills, and A. Bostrom. Time-series classification with cote: The collective of transformation-based ensembles. *IEEE Transactions on Knowledge and Data Engineering*, 27(9): 2522–2535, 2015.

A. Bagnall. Time series classification with ensembles of elastic distance measures. *Data Mining and Knowledge Discovery*, 29, 06 2014.

C. W. Baum and V. V. Veeravalli. A sequential procedure for multihypothesis testing. *IEEE Transactions on Information Theory*, 40(6):1994–2007, Nov 1994.

N. H. Bingham, G. Peskir, N. H. Bingham, and G. Peskir. Optimal stopping and dynamic programming, 2006.

A. Borovkov. *Mathematical Statistics*. Gordon and Breach Science Publishers, 1998.

V. Campos, B. Jou, X. G. i Nieto, J. Torres, and S.-F. Chang. Skip rnn: Learning to skip state updates in recurrent neural networks. In *ICLR*, 2018.

J. Carreira and A. Zisserman. Quo vadis, action recognition? a new model and the kinetics dataset. *2017 IEEE Conference on Computer Vision and Pattern Recognition (CVPR)*, pp. 4724–4733, 2017.

A. Czarnecki. Positronium properties. *arXiv preprint hep-ph/9911455*, 1999.

H. A. Dau, E. Keogh, K. Kamgar, C.-C. M. Yeh, Y. Zhu, S. Gharghabi, C. A. Ratanamahatana, Yanping, B. Hu, N. Begum, A. Bagnall, A. Mueen, and G. Batista. The ucr time series classification archive, October 2018.

K. Doya. Modulators of decision making. *Nat. Neurosci.*, 11(4):410–416, Apr 2008.

V. P. Dragalin, A. G. Tartakovsky, and V. V. Veeravalli. Multihypothesis sequential probability ratio tests .i. asymptotic optimality. *IEEE Transactions on Information Theory*, 45(7):2448–2461, Nov 1999.

V. P. Dragalin, A. G. Tartakovsky, and V. V. Veeravalli. Multihypothesis sequential probability ratio tests. ii. accurate asymptotic expansions for the expected sample size. *IEEE Transactions on Information Theory*, 46(4):1366–1383, July 2000.

J. Duchi, E. Hazan, and Y. Singer. Adaptive subgradient methods for online learning and stochastic optimization. *Journal of Machine Learning Research*, 12(Jul):2121–2159, 2011.

W. Edwards. Optimal strategies for seeking information: Models for statistics, choice reaction times, and human information processing. *Journal of Mathematical Psychology*, 2(2):312 – 329, 1965.

J. P. Gallivan, C. S. Chapman, D. M. Wolpert, and J. R. Flanagan. Decision-making in sensorimotor control. *Nat. Rev. Neurosci.*, 19(9):519–534, 09 2018.

J. I. Gold and M. N. Shadlen. The neural basis of decision making. *Annu. Rev. Neurosci.*, 30:535–574, 2007.

A. Graves. Generating sequences with recurrent neural networks. *arXiv preprint arXiv:1308.0850*, 2013.

K. Hara, H. Kataoka, and Y. Satoh. Learning spatio-temporal features with 3d residual networks for action recognition. *2017 IEEE International Conference on Computer Vision Workshops (ICCVW)*, pp. 3154–3160, 2017.

C. R. Harris, K. J. Millman, S. J. van der Walt, R. Gommers, P. Virtanen, D. Cournapeau, E. Wieser, J. Taylor, S. Berg, N. J. Smith, R. Kern, M. Picus, S. Hoyer, M. H. van Kerkwijk, M. Brett, A. Haldane, J. F. Del Río, M. Wiebe, P. Peterson, P. Gérard-Marchant, K. Sheppard, T. Reddy, W. Weckesser, H. Abbasi, C. Gohlke, and T. E. Oliphant. Array programming with NumPy. *Nature*, 585(7825):357–362, 09 2020.

M. M. Hu, H. Sun, and N. J. Kasdin. Sequential generalized likelihood ratio test for planet detection with photon-counting mode. In S. B. Shaklan (ed.), *Techniques and Instrumentation for Detection of Exoplanets IX*, volume 11117, pp. 492 – 498. International Society for Optics and Photonics, SPIE, 2019.

A. Irle and N. Schmitz. On the optimality of the sprt for processes with continuous time parameter. *Statistics: A Journal of Theoretical and Applied Statistics*, 15(1):91–104, 1984.

Y.-S. Jeong, M. K. Jeong, and O. A. Omitaomu. Weighted dynamic time warping for time series classification. *Pattern Recognition*, 44(9):2231 – 2240, 2011. Computer Analysis of Images and Patterns.

R. Johari, P. Koomen, L. Pekelis, and D. Walsh. Peeking at a/b tests: Why it matters, and what to do about it. In *Proceedings of the 23rd ACM SIGKDD International Conference on Knowledge Discovery and Data Mining*, KDD '17, pp. 1517–1525, New York, NY, USA, 2017. Association for Computing Machinery.

N. Ju, D. Hu, A. Henderson, and L. Hong. A sequential test for selecting the better variant: Online a/b testing, adaptive allocation, and continuous monitoring. In *Proceedings of the Twelfth ACM International Conference on Web Search and Data Mining*, WSDM '19, pp. 492–500, New York, NY, USA, 2019. Association for Computing Machinery.

T. Kanamori, S. Hido, and M. Sugiyama. A least-squares approach to direct importance estimation. *Journal of Machine Learning Research*, 10(Jul):1391–1445, 2009.

F. Karim, S. Majumdar, H. Darabi, and S. Chen. LSTM fully convolutional networks for time series classification. *IEEE Access*, 6:1662–1669, 2018. ISSN 2169-3536. doi: 10.1109/ACCESS.2017. 2779939.

R. Kate. Using dynamic time warping distances as features for improved time series classification. *Data Mining and Knowledge Discovery*, 30, 05 2015.

H. Khan, L. Marcuse, and B. Yener. Deep density ratio estimation for change point detection. *arXiv preprint arXiv:1905.09876*, 2019.

D. P. Kingma and J. Ba. Adam: A method for stochastic optimization. *arXiv preprint arXiv:1412.6980*, 2014.

S. Kira, T. Yang, and M. N. Shadlen. A neural implementation of wald's sequential probability rato test. *Neuron*, 85(4):861–873, February 2015.

M. Kulldorff, R. L. Davis, M. Kolczak†, E. Lewis, T. Lieu, and R. Platt. A maximized sequential probability ratio test for drug and vaccine safety surveillance. *Sequential Analysis*, 30(1):58–78, 2011.

A. Kumar, A. Gupta, and S. Levine. Discor: Corrective feedback in reinforcement learning via distribution correction. In *Proceedings of the 33rd International Conference on Neural Information Processing Systems*, pp. 13.

T. L. Lai. Asymptotic optimality of invariant sequential probability ratio tests. *Ann. Statist.*, 9(2): 318–333, 03 1981.

K. W. Latimer, J. L. Yates, M. L. Meister, A. C. Huk, and J. W. Pillow. Single-trial spike trains in parietal cortex reveal discrete steps during decision-making. *Science*, 349(6244):184–187, Jul 2015.

E. L. Lehmann and J. P. Romano. *Testing statistical hypotheses*. Springer Science & Business Media, 2006.

J. Lines, S. Taylor, and A. Bagnall. Hive-cote: The hierarchical vote collective of transformation-based ensembles for time series classification. In *2016 IEEE 16th International Conference on Data Mining (ICDM)*, pp. 1041–1046, 2016.

V. Lotov. Asymptotic expansions in a sequential likelihood ratio test. *Theory of Probability & Its Applications*, 32(1):57–67, 1988.

MATLAB. *version 9.3.0 (R2017b)*. The MathWorks Inc., Natick, Massachusetts, 2017.

S. M. McClure, D. I. Laibson, G. Loewenstein, and J. D. Cohen. Separate neural systems value immediate and delayed monetary rewards. *Science*, 306(5695):503–507, Oct 2004.

H. Nam and M. Sugiyama. Direct density ratio estimation with convolutional neural networks with application in outlier detection. *IEICE TRANSACTIONS on Information and Systems*, 98(5): 1073–1079, 2015.

E. Nikishin, P. Izmailov, B. Athiwaratkun, D. Podoprikhin, T. Garipov, P. Shvechikov, D. Vetrov, and A. G. Wilson. Improving stability in deep reinforcement learning with weight averaging. In *Uncertainty in artificial intelligence workshop on uncertainty in Deep learning*, 2018.

G. Okazawa, C. E. Hatch, A. Mancoo, C. K. Machens, and R. Kiani. The geometry of the representation of decision variable and stimulus difficulty in the parietal cortex. *bioRxiv*, 2021. doi: 10.1101/2021.01.04.425244. URL https://www.biorxiv.org/content/early/2021/01/04/2021.01.04.425244. Publisher: Cold Spring Harbor Laboratory _eprint: https://www.biorxiv.org/content/early/2021/01/04/2021.01.04.425244.full.pdf.

A. Paszke, S. Gross, F. Massa, A. Lerer, J. Bradbury, G. Chanan, T. Killeen, Z. Lin, N. Gimelshein, L. Antiga, A. Desmaison, A. Kopf, E. Yang, Z. DeVito, M. Raison, A. Tejani, S. Chilamkurthy, B. Steiner, L. Fang, J. Bai, and S. Chintala. Pytorch: An imperative style, high-performance deep learning library. In *Advances in Neural Information Processing Systems 32*, pp. 8024–8035. Curran Associates, Inc., 2019.

J. D. Roitman and M. N. Shadlen. Response of neurons in the lateral intraparietal area during a combined visual discrimination reaction time task. *J. Neurosci.*, 22(21):9475–9489, Nov 2002.

P. H. Rudebeck, M. E. Walton, A. N. Smyth, D. M. Bannerman, and M. F. Rushworth. Separate neural pathways process different decision costs. *Nat. Neurosci.*, 9(9):1161–1168, Sep 2006.

D. E. Rumelhart, G. E. Hinton, and R. J. Williams. Learning representations by back-propagating errors. *Nature*, 323:533–536, 1986.

P. Schäfer and U. Leser. Multivariate time series classification with weasel+ muse. *arXiv preprint arXiv:1711.11343*, 2017.

M. N. Shadlen, R. Kiani, W. T. Newsome, J. I. Gold, D. M. Wolpert, A. Zylberberg, J. Ditterich, V. de Lafuente, T. Yang, and J. Roitman. Comment on "Single-trial spike trains in parietal cortex reveal discrete steps during decision-making". *Science*, 351(6280):1406, Mar 2016.

J. Sochman and J. Matas. Waldboost - learning for time constrained sequential detection. In *2005 IEEE Computer Society Conference on Computer Vision and Pattern Recognition (CVPR'05)*, volume 2, pp. 150–156 vol. 2, June 2005.

M. Stone. Models for choice-reaction time. *Psychometrika*, 25(3):251–260, September 1960.

M. Sugiyama, T. Suzuki, S. Nakajima, H. Kashima, P. von Bünau, and M. Kawanabe. Direct importance estimation for covariate shift adaptation. *Annals of the Institute of Statistical Mathematics*, 60(4):699–746, 2008.

M. Sugiyama, T. Suzuki, and T. Kanamori. Density ratio estimation: A comprehensive review (statistical experiment and its related topics). 2010.

M. Sugiyama, T. Suzuki, and T. Kanamori. *Density ratio estimation in machine learning*. Cambridge University Press, 2012.

S. C. Tanaka, K. Doya, G. Okada, K. Ueda, Y. Okamoto, and S. Yamawaki. Prediction of immediate and future rewards differentially recruits cortico-basal ganglia loops. *Nat. Neurosci.*, 7(8):887–893, Aug 2004.

A. Tartakovsky. *Sequential methods in the theory of information systems (in Russian)*. Radio i Svyaz', Moscow, 1991.

A. Tartakovsky. Asymptotically optimal sequential tests for nonhomogeneous processes. *Sequential Analysis*, 17, 04 1999.

A. Tartakovsky, I. Nikiforov, and M. Basseville. *Sequential Analysis: Hypothesis Testing and Changepoint Detection*. Chapman & Hall/CRC, 1st edition, 2014.

V. V. Veeravalli and C. W. Baum. Asymptotic efficiency of a sequential multihypothesis test. *IEEE Transactions on Information Theory*, 41(6):1994–1997, Nov 1995.

P. Viola and M. Jones. Rapid object detection using a boosted cascade of simple features. In *Proceedings of the 2001 IEEE Computer Society Conference on Computer Vision and Pattern Recognition. CVPR 2001*, volume 1, pp. I–I, Dec 2001.

P. Virtanen, R. Gommers, T. E. Oliphant, M. Haberland, T. Reddy, D. Cournapeau, E. Burovski, P. Peterson, W. Weckesser, J. Bright, S. J. van der Walt, M. Brett, J. Wilson, K. Jarrod Millman, N. Mayorov, A. R. J. Nelson, E. Jones, R. Kern, E. Larson, C. Carey, İ. Polat, Y. Feng, E. W. Moore, J. Vand erPlas, D. Laxalde, J. Perktold, R. Cimrman, I. Henriksen, E. A. Quintero, C. R. Harris, A. M. Archibald, A. H. Ribeiro, F. Pedregosa, P. van Mulbregt, and S. . . Contributors. SciPy 1.0: Fundamental Algorithms for Scientific Computing in Python. *Nature Methods*, 17: 261–272, 2020.

Z. Wang, W. Yan, and T. Oates. Time series classification from scratch with deep neural networks: A strong baseline. In *2017 International Joint Conference on Neural Networks (IJCNN)*, pp. 1578–1585, 2017.

L. Wei and E. J. Keogh. Semi-supervised time series classification. In *KDD '06*, 2006.

X. Xiong and F. De la Torre. Supervised descent method and its applications to face alignment. In *2013 IEEE Conference on Computer Vision and Pattern Recognition*, pp. 532–539, June 2013.

K. Yang and C. Shahabi. An efficient k nearest neighbor search for multivariate time series. *Information and Computation*, 205(1):65 – 98, 2007.

