# OpenReview forum: "Sequential Density Ratio Estimation for Simultaneous Optimization of Speed and Accuracy"
_ICLR.cc/2021/Conference — ICLR 2021 Spotlight_

### Official Review · AnonReviewer2 · 2020-10-28
**Elegant combination of SPRT and density ratio estimation, though optimality claims are overstated**

**Rating:** 8
**Confidence:** 4

**Review:**

[EDITED AFTER DISCUSSION: My concerns are largely addressed and the paper is now stronger. I very much hope to see it at the conference, and have updated my review and rating accordingly.]

The paper proposes a new algorithm for early classification of sequential data, exploiting approaches to density ratio estimation to enable applying a sequential probability ratio test-type algorithm on perceptually rich data with no explicit likelihood. The algorithm is trained using a novel density ratio estimation loss alongside more standard cross-entropy, and shows strong performance on a number of provided benchmarks, including on a new variant of sequential MNIST. The paper claims ability to control speed-accuracy tradeoff in early classification, and applying the Wald SPRT on arbitrary sequential data.

I enjoyed reading this paper: it combines two ideas (sequential likelihood ratio testing and density ratio estimation) in a way that appears obvious after the fact, but as often with these sort of post-hoc obvious ideas, is novel to my knowledge, and elegant. Trying to get something like the SPRT working beyond pairs of simple hypotheses has been under investigation for over half a century, and this paper is a worthwhile attempt. The empirical results look pretty good as well.

At the same time, I think the paper does overstate the benefit of the theoretical connection to the classical SPRT and previous non-i.i.d. extensions, and doesn't engage sufficiently with the challenges of stopping rules. I discuss these issues in more detail next, and conclude with some minor points about presentation, background work, and analysis.

# Connections to the classical SPRT #
The paper claims to extend Wald's SPRT to non-i.i.d. data, and in particular its optimality properties. This includes claims in section 1 (on approaching Bayes-optimality and extending the Wald SPRT to arbitrary sequential data), at the beginning of section 4 (on how the LLLR enables performing the SPRT, which is provably optimal), and in section 5 (on approaching if not reaching Bayes-optimality). As far as I can tell, the paper does not enable performing the Wald SPRT on arbitrary data, and does not provide a provably optimal sequential test. The claim that it approaches optimality in any formal sense is likewise not supported as far as I can tell. Broadly, a sequential test consists of an update rule and a decision rule -- for the SPRT the update rule and decision rule are both optimal. For most extensions (non-i.i.d., multi-hypothesis, deadlines, etc), the update still follows Bayes' rule, and the optimal decision rule can only be found numerically (if at all), so some heuristic is given. This heuristic is often a fixed threshold, with asymptotic optimality guarantees). SPRT-TANDEM seems to be in the family of such extensions: it still applies Bayes' rule sequentially, and the stopping is given based on a fixed threshold. Thus, its optimality is asymptotic at best, and stronger claims are not supported. Furthermore, the paper doesn't make it clear whether the standard conditions for asymptotic optimality apply to the SPRT-TANDEM either: in my rough understanding, the standard asymptotic result is as risk goes to 0 (or equivalently, the LLR goes to infinity, and the threshold goes to infinity). I'm not sure that we know the SPRT-TANDEM LLR to grow in this way, and empirically, it seems like the LLR saturates to some fixed value, especially with high-order N, which means high threshold values are not achievable and risk cannot go to 0. We should also expect the SPRT-TANDEM to depart further from optimality as the model approaches the end of the video (since the optimal thing to do there is to gradually collapse the decision boundary, as the appendix reminds us). I recommend moderating these claims regarding optimality, and / or strengthening the results if possible.

# Stopping rules #
The paper does not provide guidance on stopping rules, which limits practical use, and does not report on the thresholds used to generate the speed-accuracy tradeoff figures. Presumably, the simplest thing is to set the threshold to the desired accuracy (which I think will do the right thing in the no-overshoot case?). Does this work for SPRT-TANDEM to achieve a given accuracy? If not, is there another heuristic that applies? I recommend addressing this question in more detail. Relatedly, the paper criticizes Mori et al. 2018 and Hartvigsen et al. 2020 for using a separate objective for determining stopping and accuracy, but in fact SPRT-TANDEM would likewise need some dynamic programming or RL solver to have an optimal stopping policy, similarly to that prior work. I recommend providing explicit guidance about stopping rules, and moderating the claims relative to prior work. Solving for an optimal stopping policy would also strengthen the paper.

# Presentation issues #
The paper tries to cram a lot into the short ICLR format, supported by an extensive appendix. I appreciate the inclusion of classical SPRT results in the appendix, which may be unfamiliar to the ICLR audience. At the same time, the main text does not provide much intuition about the novel LLLR loss, which is given very little explanation considering it is presented as one of the paper's major contributions. The relationships and improvements relative to KLIEP are presented too tersely, and a reader not familiar with that precise method will not know what to make of them. The paper would do better to provide more exposition there, perhaps in favor of moving the results tables to the appendix (since they show the same information as figure 3 as far as I can tell).
In addition, the SAT curves are too busy, small, and hard to read. For the main document, I would recommend increasing font and symbol sizes, and presenting fewer orders of SPRT-TANDEM models (e.g. just order-1 and best-order), and fewer hitting times per model (e.g. there is no need to present the accuracy at every one of the last 10 frames if the accuracies are all the same there). Finally, the LLR trajectory figures can use partial transparency to make the individual traces easier to see.

# Additional/minor points #
- **Background work**. I appreciated the fairly detailed review of past work related to the SPRT. A few notable missing pieces related to neuroscience are work predating Kira et al. 2015 in applying the SPRT to neural data (e.g. Gold & Shadlen 2002) and to human decision making more broadly (e.g. Stone, 1960; Edwards, 1965, Ashby 1983, and others). Missing work related to practical applications of the SPRT includes Johari et al. 2017, Ju et al. 2019, and others from the domain of internet experimentation. None of these are critical omissions, I only bring them up considering the already-broad review.
- **Statistical analysis**.  Given that the paper has a clear hypothesis (that SPRT-TANDEM outperforms competitors), it seems more sensible to perform repeated measures regression with planned contrasts rather than post-hoc testing for significance. This is a minor issue.

---

> ### Author Response · Authors · 2020-11-17
> **Reply to Reviewer 2 (4/4)**
>
>
> # Presentation issues
>
> >## The relationships and improvements relative to KLIEP are presented too tersely, and a reader not familiar with that precise method will not know what to make of them. The paper would do better to provide more exposition there, perhaps in favor of moving the results tables to the appendix.
>
> We agree with your suggestion. The tables are moved to Appendix J, and we added a more detailed explanation about KLIEP in the main text. Please see Section 4.1.
>
>
> # Additional/Minor points
>
> >## Background work. I appreciated the fairly detailed review of past work related to the SPRT. A few notable missing pieces related to neuroscience are work predating Kira et al. 2015 in applying the SPRT to neural data (e.g. Gold & Shadlen 2002) Missing work related to practical applications of the SPRT includes Johari et al. 2017, Ju et al. 2019, and others from the domain of internet experimentation. None of these are critical omissions, I only bring them up considering the already-broad review.
>
> Thank you for bringing up the additional related works, from classical psychology to modern machine learning. As for neural correlate of decision making, we already wrote an extensive review in Appendix B of the first submission, citing articles including Gold & Shadlen 2002. Please find it if you have not done so, we hope you would enjoy it.
>
> We included all the other suggested papers in the updated manuscript.
>
> ## Last comment from the authors
>
> Again, thank you for your insightful comments. We added the above discussion to the updated manuscript (Appendix D, subsection "How optimal is the SPRT-TANDEM?"). The discussion with you really helped to clarify the advantages and limitations of the SPRT-TANDEM.

---

> ### Author Response · Authors · 2020-11-17
> **Reply to Reviewer 2 (3/4)**
>
>
> # Stopping rules
>
> >## The paper does not provide guidance on stopping rules, which limits practical use, and does not report on the thresholds used to generate the speed-accuracy tradeoff figures. Presumably, the simplest thing is to set the threshold to the desired accuracy (which I think will do the right thing in the no-overshoot case?). Does this work for SPRT-TANDEM to achieve a given accuracy? If not, is there another heuristic that applies?
>
> Because of the experimental limitations, it is difficult to achieve the theoretically-predicted target accuracies from the user-defined thresholds. Therefore, practical usage of the SPRT-TANDEM we assume is as follows: after the training, (1) plot the speed-accuracy tradeoff curve on the test dataset and (2) choose a desired accuracy and the corresponding threshold values. Computing the speed-accuracy-tradeoff curve is not expensive (from several seconds to at most a few minutes on CPU in our experiments), and importantly is computable without re-training. Note that this flexible property is missing in most of the other deep neural networks: controlling speed usually means changing the network structures and training it all over again.
>
> To plot the speed-accuracy tradeoff curves in our paper, we determine the thresholds following the steps below. (1) Compute all the LLR trajectories of the test dataset; (2) Compute the maximum and minimum value of $|\mathrm{LLR}|$, where $|...|$ is the absolute value symbol; (3) Generate the upper, positive thresholds restricted between the max $|\mathrm{LLR}|$ and the min $|\mathrm{LLR}|$, where the thresholds are linearly uniformly separated (the lower thresholds are the upper thresholds multiplied by -1); (4) Run the SPRT and obtain 2D points (mean hitting time, balanced accuracy) corresponding to the pairs of thresholds; (5) Plot them on the speed-accuracy 2D plane, and linearly interpolate the points. The last interpolation provides a continuous curve.
>
>
> >## Relatedly, the paper criticizes Mori et al. 2018 and Hartvigsen et al. 2020 for using a separate objective for determining stopping and accuracy, but in fact SPRT-TANDEM would likewise need some dynamic programming or RL solver to have an optimal stopping policy, similarly to that prior work.
>
> The statement about separate objectives is meant for the following. Firstly, during the training, the SPRT-TANDEM does not need two objectives for speed and accuracy. After the training, running the SPRT-TANDEM with multiple thresholds is not computationally expensive, as we stated above. Once the threshold is fixed, the SPRT-TANDEM does not need a stopping mechanism, unlike reinforcement learning-based models.
>
> Minor note: we did not intend to "criticize" the two previous works - rather, we wanted to highlight the advantage of the SPRT-TANDEM. The two are both insightful papers.

---

> ### Author Response · Authors · 2020-11-17
> **Reply to Reviewer 2 (2/4)**
>
> >## the paper doesn't make it clear whether the standard conditions for asymptotic optimality apply to the SPRT-TANDEM either: in my rough understanding, the standard asymptotic result is as risk goes to 0 (or equivalently, the LLR goes to infinity, and the threshold goes to infinity).
>
> Our experiment truncates the SPRT at the maximum timestamp, which is just an experimental setting and is not an essential assumption of the SPRT-TANDEM. Under the truncated SPRT condition, gradually collapsing thresholds gives the optimal stopping. In the i.i.d. case, Theorem 3.2.3 on p.154 in [Tartakovsky+2014](https://apps.dtic.mil/dtic/tr/fulltext/u2/a625103.pdf) shows that the optimal truncated sequential test is the truncated SPRT with collapsing thresholds, which are obtained via backward induction (more general setting is given in [Bingham+2006](http://citeseerx.ist.psu.edu/viewdoc/download?doi=10.1.1.570.9068&rep=rep1&type=pdf)). However, the backward induction is possible only after observing the full sequence, which critically limits practical applicability.
>
> If we ignore the truncation, the asymptotic, non-i.i.d. optimality is given in Theorem A.6, which we rely on for a theoretical backbone of the SPRT-TANDEM. The asymptotic assumption requires that the LLRs asymptotically increase or decrease and that the user-defined risk goes to 0, or equivalently, that the thresholds go to infinity.
>
> However, because we cannot know the true LLR of real-world datasets, it is difficult to discuss whether the assumption of the increasing LLR is valid on the three databases (NMNIST, UCF, and SiW) we tested. Numerical simulation may be possible, but it is out of our scope because our primary interest is to implement a practically usable SPRT under real-world scenarios. That being said, it certainly is an interesting future work.
>
> We can at least make a discussion on _estimated_ LLRs. Unlike true LLRs, they are not directly related to the asymptotic assumption. Still, we may expect that if the estimated LLRs satisfy the asymptotic assumption, the true LLRs also do, given that the estimation goes well: the LLLR provably provides the true LLRs up to normalization at its minimum, given a sufficiently large sample size ([Sugiyama+2008](https://www.ism.ac.jp/editsec/aism/pdf/060_4_0699.pdf),  and Appendix E).
>
> >## I'm not sure that we know the SPRT-TANDEM LLR to grow in this way, and empirically, it seems like the LLR saturates to some fixed value, especially with high-order N, which means high threshold values are not achievable and risk cannot go to 0.
>
> However, as you pointed out, the resulting estimated LLRs tend to be asymptotically flat, especially when $N$ is large. One potential reason is the TANDEM formula: the first and second term of the formula has a different sign. Thus, the resulting log-likelihood ratio will be updated only when the difference between the two terms are non-zero. Because the first and second term depends on $N+1$ and $N$ inputs, respectively, it is expected that the contribution of one input becomes relatively small as $N$ is enlarged. We are aware of this issue and already started working on it as future work. But this flatness at least does not spoil the practical efficiency of the SPRT-TANDEM, as our experiment shows.

---

> > ### Comment · AnonReviewer2 · 2020-11-19
> > **I'm still confused here**
> >
> > I appreciate the correction re: estimated vs true LLR w.r.t. the asymptotic optimality claim, and I agree that it does not spoil the empirical validation. That said, I don't think you can have it both ways here: if the estimated LLR is a good approximation for the true LLR, then both flatten. If the estimated LLR is a bad approximation of the true LLR, then you have a mismatch between true and estimated LLR. Either of those possibilities is problematic for the claim on p7 that SPRT-TANDEM is "getting close to asymptotic Bayes optimally" (also, note missing closing parentheses there and should probably be optimally->optimality).
> >
> > As an aside, you note that Bingham and Tartakovsky give the optimal collapsing threshold by backward induction (as do Frazier & Yu, NeurIPS 2008), but if I understand things correctly, this problem is basically a 2-state 3-action POMDP for which fancier solvers should be possible (see e.g. Ross et al. JAIR 2008) that better approximate the optimal policy.

---

> > > ### Author Response · Authors · 2020-11-20
> > > **Thank you for the quick reply. Please find our additonal comments below.**
> > >
> > > > I appreciate the correction re: estimated vs true LLR w.r.t. the asymptotic optimality claim, and I agree that it does not spoil the empirical validation.
> > > That said, I don't think you can have it both ways here: if the estimated LLR is a good approximation for the true LLR, then both flatten.
> > > If the estimated LLR is a bad approximation of the true LLR, then you have a mismatch between true and estimated LLR.
> > > Either of those possibilities is problematic for the claim on p7 that SPRT-TANDEM is "getting close to asymptotic Bayes optimally"
> > >
> > > Yes, we agree with you. Actually, we did not intend to persist on strict asymptotic Bayes optimality even under the flat LLRs: Because Theorem A.6 restricts the distribution of LLR, the non-i.i.d. SPRT can depart from optimal under certain conditions.
> > >
> > > To moderate our optimality argument, we swapped the statement, "getting close to asymptotic Bayes optimality" with "Is the proposed algorithm's superiority because the SPRT-TANDEM successfully estimates the true LLR to approach asymptotic Bayes optimality? We discuss potential interpretations of the experimental results in Appendix D", inviting readers to a more detailed discussion in Appendix (subsection "How optimal is the SPRT-TANDEM?", which summarizes our discussion). Please see the updated manuscript.
> > >
> > >
> > > > missing closing parentheses there and should probably be optimally->optimality
> > >
> > > Thank you for pointing out, we also fixed them in the updated manuscript.

---

> ### Author Response · Authors · 2020-11-17
> **Reply to Reviewer 2 (1/4)**
>
> First of all, we would like to thank you for providing critical and insightful comments based on a deep understanding of sequential hypothesis testing. We heartily enjoyed reading your review.
>
> Note that the updated appendix is now following the main text, within the same pdf.
>
> >## I enjoyed reading this paper: it combines two ideas (sequential likelihood ratio testing and density ratio estimation) ... Trying to get something like the SPRT working beyond pairs of simple hypotheses has been under investigation for over half a century, and this paper is a worthwhile attempt.
>
> We are pleased to hear that you enjoyed our paper. Combining the SPRT and density ratio estimation, we could import rich multidisciplinary knowledge (including theoretical and experimental one) across machine learning and psychology/neuroscience communities.
>
> # Connections to the classical SPRT
>
> >## At the same time, I think the paper does overstate the benefit of the theoretical connection to the classical SPRT and previous non-i.i.d. extensions
>
> Indeed, to fully enjoy the theoretical benefit of the classical and non-i.i.d. SPRT (Theorem A.5 and A.6), the experiments should (at least approximately) satisfy the necessary conditions (assumptions) of the theorems. On the other hand, realizing/devising such an algorithm on real-world datasets is challenging, mainly because we cannot know the true distribution or check the validity of the assumptions in a strict sense neither. Addressing this problem is one of our central and exciting future works. In the following, we share our discussions and current understanding, many of which turned out to be closely related to your insightful comments.
>
> >## As far as I can tell, the paper does not enable performing the Wald SPRT on arbitrary data, and does not provide a provably optimal sequential test. The claim that it approaches optimality in any formal sense is likewise not supported as far as I can tell.
> >## Broadly, ... Thus, its optimality is asymptotic at best, and stronger claims are not supported
>
> Yes, we agree with you - but actually, we intended to mention very similar to what you suggested - thus, it is a miscommunication, at least partially. As you noticed, we often used the phrase "approaching" Bayes optimality. We used the phrase to indicate that our algorithm is "getting close to the asymptotically optimal solution (in the sense that the deep neural network can be trained to estimate the true likelihood ratio)," unlike "almost reaching the exact optimal solution." In the updated manuscript, we explicitly stated that our algorithm is not strictly optimal because of the experimental limitations.
>
> > in section 1 (on approaching Bayes-optimality and extending the Wald SPRT to arbitrary sequential data),
>
> Strictly speaking, the statement "arbitrary sequential data" ignores a technical assumption; i.e., the SPRT terminates for all the LLR trajectories under consideration with probability one (Equation (61)) (In the non-i.i.d. case, the corresponding assumption is Equation (66), which states that the LLRs asymptotically increase or decrease, ensuring the non-i.i.d. SPRT's termination). Given this assumption, *the more precisely we estimate the LLRs, the more we approach the genuine SPRT implementation and thus its asymptotic Bayes optimality.* (Though we aware of the "flat LLR" issue: please see our comments below).

---

> > ### Comment · AnonReviewer2 · 2020-11-19
> > **Writeup is improved, but I think still could use more nuance**
> >
> > I appreciate that some of the claims in the paper are moderated. I still think that the proposed test is not strictly Wald's SPRT, even though the paper implies it to be so (in the same way that the other SPRT variants aren't the Wald SPRT, they're alternate SPRTs that address various challenges with the original formulation). But this is a nitpick.

---

### Official Review · AnonReviewer3 · 2020-10-29
**The paper involves a certain interesting property connecting a flexible neural network function-approximator to conventional SPRT setting.**

**Rating:** 6
**Confidence:** 4

**Review:**

The authors propose how to use the neural networks to estimate the posteriors for each label y for the log likelihood ratio (LLR) estimation. The LLR estimation is performed using the multivariate inputs from the windows of time series, and it is used for the SPRT criterion without conventional iid assumption.

The paper contains an interesting idea of using the neural networks for the prediction of likelihoods and accumulating the information for the conventional sequantial probability ratio test (SPRT), which is well known for explaining the speed-accuracy tradeoff for decision making. The experimental results using various datasets show the relevance of the algorithm in terms of improving the speed-accuaracy tradeoff.

The authors presented a reasonable combination of two different objective functions: LLLR and L_multiplet. Though the authors did not mention explicitly, one is the objective for LLR which is ill-posed because pairs of high-biased neural network outputs can result in a small LLR by preserving only the ratio of the outputs correct. The other is the posterior objective (L_multiplet) which will alleviate the ill-posedness of the first objective by making the outputs of neural networks as close as possible to the correct one though they do not use those posteriors once it can estimate the LLR correctly.

My question about this paper is the learning procedure. For LLLR training, there is no need for the sync of x^(t) for different ys. I don’t see the explanation about choosing the index t in LLLR.

In the paragraph below Eq. (6), I don't understand what it means by the KL divergence between two ratios.

---

> ### Author Response · Authors · 2020-11-17
> **Reply to Reviewer 3**
>
> >## My question about this paper is the learning procedure. For LLLR training, there is no need for the sync of x^(t) for different ys. I don’t see the explanation about choosing the index t in LLLR.
>
> Could you please provide details on what did you mean by "there is no need for the sync of x^(t) for different ys"?
>
> While we are waiting for your reply, we recap the LLLR training procedure described in Section 4.
>
> Given a maximum timestamp $T\in\mathbb{N}$ and dataset size $M\in\mathbb{N}$, let $S := ${$ (X_i^{(1,T)}, y_i) $}$_{i=1}^{M}$ be a sequential training dataset. The LLLR,
>
> $$L_\mathrm{LLR} = \frac{1}{MT} \sum_{i=1}^{M}
>             \sum_{t=1}^{T} \left| y_i - \sigma\left(
>             \log\left(\frac{\hat{p}(x_i^{(1)},x_i^{(2)}, ..., x_i^{(t)} | y=1)}{\hat{p}(x_i^{(1)},x_i^{(2)}, ..., x_i^{(t)} | y=0)}\right)
>             				           \right) \right|$$
>
> is computed at every timestamp at every data and averaged. Thus, the index $t$ starts from the beginning of the sequential data, increases towards the end of the data.
>
> >## In the paragraph below Eq. (6), I don't understand what it means by the KL divergence between two ratios.
>
> Thank you for pointing out the inappropriate expression. We provided a more detailed and precise description of the LLLR and KLIEP in Section 4 of the updated manuscript.
>
> Note that the updated appendix is now following the main text, within the same pdf.
>
> # A question from the authors to Reviewer 3
>
> Could you please provide a more detailed reason for your rating? Your rating, 6 (marginally above acceptance), is relatively low compared to other reviewers but we had a hard time figuring out why you thought so. We are open to discussion - please feel free to extend any further questions.

---

### Official Review · AnonReviewer1 · 2020-10-30
**An exceptionally documented and motivated piece of work!**

**Rating:** 7
**Confidence:** 3

**Review:**

SUMMARY:
This work describes a new algorithm, SPRT-TANDEM, for classifying sequential data as early as possible. It builds upon several previous works (SPRT, KLIEP, and deep neural networks) to propose a well-engineered and well-motivated solution for the considered problem.

STRENGTHS:
- This work is exceptionally documented, on all accounts: related work is multidisciplinary, broad and thorough; all losses and algorithms are derived from first principles; experiments are varied and include every last details regarding their setups, evaluation metrics or outcomes.
- Albeit only briefly mentioned in the main, the method has strong theoretical foundations, as documented in Appendix A.
- Experiments show stronger benchmark results than the considered baselines (LSTM-m/s, EARLIEST and 3DResNet), on a quite diverse set of experiments (images, videos).
- Although the classification of sequential data is not among the most popular topics, I believe this contribution to be significant for the field.

WEAKNESSES:
- I believe the 8-page limit of ICLR does make this work justice, given the extensive documentation that comes along in the supplementary materials. The short format of the main paper makes it difficult at places to fully follow or appreciate the contributions presented in this work. (Should this paper be rejected, I would recommend it to be submitted to JMLR, which format is certainly a better fit.)

---

> ### Author Response · Authors · 2020-11-17
> **Reply to Reviewer 1**
>
> Thank you for the comment, "An exceptionally documented and motivated piece of work!" We are also pleased to see that you found that our work has strong theoretical foundations.
>
> >## I believe the 8-page limit of ICLR does make this work justice, given the extensive documentation that comes along in the supplementary materials. The short format of the main paper makes it difficult at places to fully follow or appreciate the contributions presented in this work.
>
> Indeed, we had a hard time squeezing both theoretical and experimental results into the 8-page limit. Thus, we hope that readers refer to Appendix (note that now the updated appendix is following the main text, within the same pdf) depending on their intellectual curiosity.

---

### Official Review · AnonReviewer4 · 2020-11-02
**Very well written paper proposing an algorithm minimizing the divergence between estimated and true Log-Likelihood Ratios of SPRT and making it thereby Bayes optimal for various real-world applications.**

**Rating:** 9
**Confidence:** 4

**Review:**

The paper proposes a novel SPRT-TANDEM algorithm minimizing the divergence between estimated and true Log-Likelihood Ratios of SPRT and making it thereby Bayes optimal for various real-world applications. The paper is very well written, clear and scientifically sound and provides  extensive contributions, e.g. a database in addition to the algorithm. Performance of the algorithm is demonstrated via three experiments.

Previous research is given sufficient credit. The only thing I would still like to see more is the discussion at the conclusions. Why does this seemingly simple modification to the existing SPRT method provide so superior performance.

The appendices are referred a lot in the text but they are missing from the paper?

A very minor comment: The following sentence is a bit vaguely written:
Long short-term memory (LSTM)-s/LSTM-m impose monotonicity on classification ...
I guess it should be : Long short-term memory (LSTM) variants LSTM-S and LSTM-M impose monotonicity on classification ...

---

> ### Author Response · Authors · 2020-11-17
> **Reply to Reviewer 4**
>
> Thank you for the high rating! We are also encouraged to hear that the paper is very well-written.
>
> >## The only thing I would still like to see more is the discussion at the conclusions. Why does this seemingly simple modification to the existing SPRT method provide so superior performance.
>
> Could you please provide details on what did you mean by "simple modification?" Is it about the TANDEM formula, the deep neural network, the proposed loss function LLLR, or something else?
>
> While we are waiting for your reply, we provide a general explanation of why the SPRT-TANDEM shows superior performance.
>
> The original SPRT is proven to be Bayes-optimal, but the SPRT works under strict assumptions: sequential data must be (conditionally) independent, and a user must calculate the log-likelihood ratio from the data. The SPRT-TANDEM overcomes these limitations by estimating the true log-likelihood ratio using a deep neural network aided with the density ratio estimation algorithm.
>
> Please also see our related answer to Reviewer 5's first question.
>
> >## The following sentence is a bit vaguely written: Long short-term memory (LSTM)-s/LSTM-m impose monotonicity on classification ... I guess it should be : Long short-term memory (LSTM) variants LSTM-S and LSTM-M impose monotonicity on classification ...
>
> Thank you for the suggestion; we updated the paper according to your recommendation.
>
> >## The appendices
>
> To avoid confusion, we attached the appendix, which was originally included in the supplementary material, at the end of the main text.

---

### Official Review · AnonReviewer5 · 2020-11-04
**SPRT-TANDEM additional review**

**Rating:** 7
**Confidence:** 3

**Review:**

# Summary
This work introduces SPRT-TANDEM an algorithm to train a sequential probability ratio test (SPRT) as a neural network. This network is then used to discriminate between two hypotheses as fast as possible (seeing the smallest number of observations in a sequence) while maintaining a certain level of accuracy. The main contribution of this work is to enable Wald's SPRT without actual knowledge of the ratio, learning a neural network to model it.

# Major comments
## Pros:
The paper does a good job of introducing the problem statement, that is the "fast" classification of sequential data. The algorithm introduced is well motivated and bridges the gap between "classic statistical" methods and machine learning approaches for sample-efficient time series classification. The experimental, results though not outstanding, show that SPRT-TANDEM outperforms other deep learning methods.  These experiments are insightful by the fact that they compare the performance of the different methods for different mean hitting time. Overall the paper is pleasant to read and introduce a new method that could be helpful for some practitioners.

## Cons:
1) The related work is quite superficial (even taking into account app B). In particular, I would have liked a deeper comparison with LSTM-s/m and EARLIEST, discussing the drawback/advantages of these methods with respect to SPRT-TANDEM.
2) In the proof of 4, just before eq 70 you say: "Let us assume that the process {x(s)}ts=1 is i.i.d., namely -> eq 70". This seems wrong to me. The assumption you're making there is that the process has independent component conditionally to the class y (e.g for t = 2, the Bayesian network: x_1 <- y -> x_2). This is still a reasonable hypothesis however this is not equivalent to simply assuming the process is iid (which then would mean for t = 2, the Bayesian network: x_1 -> y <- x_2 and would not make eq 70 correct).
3) The three tasks on which you test the models seem to be quite well solved after a few samples on average for all models. I think it would be worth testing the models on tasks that require more samples for reaching good performance and maybe where the temporality required (hyperparameter N) is larger.
4) It is not very clear to me what are the respective roles of LLLR and MCEL. I do not understand why both are useful, I would have thought that CE in itself would be sufficient. The ablation study you did is interesting but I would have liked to get further insights about what is happening there.
5) I believe that testing your method on a toy problem for which the correct ratio is known would be insightful about the "optimality" of your method.

# Minor comments
- Page 2 "As an .. orDEr"
- Loss written with capital everywhere.
- How to choose N: You say that training the features extractor is faster than the integrator's however it is not clear to me how you can train these two parts independently from each other.
- For comparison on NMNIST it could be interesting to see the performance of a simple classifier on the 19th image alone.
- You're talking about Optuna for hyperparameter optimization, this is unknown to me. A word about how it is working could be nice.
- Why are the number of trials different?
- The purpose of SPRT-TANDEM is to be as fast as possible, it could be interesting to clearly state somewhere how comparable are the different methods in term of computing times even if they are very close to each other.

---

> ### Author Response · Authors · 2020-11-17
> **Reply to Reviewer 5 (4/4)**
>
> >## Why are the number of trials different?
>
> It is because some models are prohibitively expensive to run multiple times. However, we conducted statistical tests in order to have a fair comparison among results with different numbers of trials. Let us provide the details below.
>
> Our experiment has two types of "trial numbers": for tuning and statistics. In the tuning step, the numbers of trials between models differ only at most by a factor of a few. After fixing the hyperparameters, the statistics step follows, where a naive comparison of models with largely different trial numbers can be unfair. Thus, we conducted statistical tests (two-way ANOVA followed by the Tukey-Kramer multi-comparison test), in which small numbers of trials lead to reduced test statistics, making it challenging to claim significance. For example, the Tukey-Kramer method's test statistic is proportional to $1/\sqrt{(1/n + 1/m)}$, where $n$ and $m$ are trial numbers of two models to be compared. Nevertheless, the results show that the SPRT-TANDEM is statistically significantly better than the other baselines, as shown in Appendix H. These statistical tests are standard practice in some research fields, such as biological science, in which variable trial numbers are inevitable in experiments.
>
> We ran each model multiple times with different random initial values to conduct statistical tests. As described in Section 5, all the models use the best hyperparameters, objectively found with Optuna; therefore, no models have disadvantages. Again, the statistical tests are crucial for not reporting the "champion data" but showing the results' reproducibility, often lacking in modern machine learning research. Thus, we respectfully disagree with your comment in the Pros section:
>
> >The experimental, results though not outstanding
>
>
> >## How to choose N: You say that training the features extractor is faster than the integrator's however it is not clear to me how you can train these two parts independently from each other.
>
> As we described in Section 3 - subsection _Neural network that calculates the SPRT-TANDEM formula_, the training of the feature extractor is followed by the training of the temporal integrator.
>
> Firstly, the feature extractor, a ResNet with a global average pooling layer, is trained for a binary classification problem. All the video frames are separated, and the feature extractor is trained using each image. During the training, the global average pooling layer is followed by a fully-connected layer that outputs two-dimensional logits.
>
> After the training of the feature extractor, each frame's feature vector is extracted from the global average pooling layer and arranged as a sequential dataset of maximum timestamp $T \in \mathbb{N}$ and dataset size $M \in \mathbb{N}$. The temporal integrator is trained on the sequential dataset. The output vectors from the temporal integrator are transformed with a fully-connected layer into two-dimensional logits, which are then inputted to the softmax layer to obtain posterior probabilities.
>
> We trained the feature extractor and temporal integrator separately (i.e., non-end-to-end) because we found that the separated training achieves better performance on many databases.
>
>
> >## You're talking about Optuna for hyperparameter optimization, this is unknown to me. A word about how it is working could be nice.
>
> We agree with your suggestion: a brief description will be reader-friendly. The default algorithm of Optuna is [Tree-structured Parzen Estimator (TPE)](https://optuna.readthedocs.io/en/stable/reference/samplers.html). The original article is [Bergstra+2011](https://www.lri.fr/~kegl/research/PDFs/BeBaBeKe11.pdf). We documented the above specs in the original location in the main text (Section 3, subsection _How to choose the hyperparameter N?_).
>
> >## The purpose of SPRT-TANDEM is to be as fast as possible, it could be interesting to clearly state somewhere how comparable are the different methods in term of computing times even if they are very close to each other.
>
> Thank you for the suggestion. The difference in computing times is negligible: please find the details below.
>
> As we stated in Section 5, all the early-classification models, namely the SPRT-TANDEM, LSTM-m, LSTM-s, and EARLIEST, share the same feature extractor (ResNet). Also, the above four models share the same temporal integrator (LSTM). Because these two processes are computation bottlenecks, the computing times are very close to each other, as you mentioned.
>
> The SPRT-TANDEM has a few additional steps compared to the other baselines:
>
> - Compute the TANDEM formula (Equation 4): two divisions, one subtraction, and one addition per timestamp.
> - Conduct the SPRT (Equation 1-3): one if-elseif-else statement per timestamp.
>
> Thus, computing the above steps is negligible compared to that of the feature extractor / temporal integrator.

---

> > ### Author Response · Authors · 2020-11-25
> > **A question from the authors to Reviewer 5**
> >
> > Dear Reviewer 5,
> >
> > Thank you again for your valuable feedbacks. We hope that we could provide clear answers to all the major questions. Now we would like to hear back from you - were the answers satisfactory? If you have any additional question, please feel free to address here (although the clock is ticking). We are open to discussion until the deadline, the end of Nov.24 (Anywhere on Earth).
> >
> > Sincerely,
> > Paper #95 anonymous authors

---

> ### Author Response · Authors · 2020-11-17
> **Reply to Reviewer 5 (3/4)**
>
> >## I believe that testing your method on a toy problem for which the correct ratio is known would be insightful about the "optimality" of your method.
>
> Please see the updated Appendix F. We created and ran a toy model trained with the LLLR, taking inputs sampled from multivariate Gaussian distributions. Experimental results show that a multi-layer perceptron (MLP) trained with the proposed LLLR achieves a smaller estimation error than an MLP with cross-entropy (CE)-loss.
>
> ### Experimental Settings
> Following [Sugiyama+2008](https://www.ism.ac.jp/editsec/aism/pdf/060_4_0699.pdf), let $p_0(x)$ be the $d$-dimensional Gaussian density with mean $(2, 0, 0, ..., 0)$ and covariance identity, and $p_1(x)$ be the $d$-dimensional Gaussian density with mean $(0, 2, 0, ..., 0)$ and covariance identity. In the experiment, the dimension $d$ is set to $100$.
>
> The task for the neural network is to estimate the density ratio:
>
> $$
> \hat{r_i} := \hat{r}(x_i) := \frac{\hat{p}_1(x_i)}{\hat{p}_0(x_i)}.
> $$
>
> Here, $x$ is sampled from one of the two Gaussian distributions, $p_0$ or $p_1$, and is associated with class label $y=0$ or $y=1$, respectively. We compared the two loss functions, CE-loss and LLLR:
>
> $$\mathrm{LLLR}:= \frac{1}{N}\sum_{i=1}^{N}
> \left|
>     y - \sigma\left(\log\hat{r_i}\right)
> \right|
> $$
>
> where $\sigma$ is the sigmoid function.
>
> A simple Neural network consists of 3-layer fully-connected network with nonlinear activation (ReLU) and BatchNorm layers is used for estimating $\hat{r}(x)$. The numbers of nodes in the hidden layers are $100$, $100$, and $2$. Evaluation metric is normalized mean squared error (NMSE, [Sugiyama+2008](https://www.ism.ac.jp/editsec/aism/pdf/060_4_0699.pdf)):
>
> $$
> \mathrm{NMSE}:= \frac{1}{N}\sum_{i=1}^{N}
> \left(
>     \frac{\hat{r_j}}{\sum_{j=1}^{N}\hat{r_j}} -
>     \frac{r_i}{\sum_{j=1}^{N}r_j}
> \right)
> $$
>
> The MLP is trained either with the LLLR or CE-loss, repeated 56 times with different random initializations to calculate statistics. The optimizer is [Adam](https://arxiv.org/abs/1412.6980). The result (see the figure below or Appendix F) shows that the LLLR can reduce NMSE more than the error-bar range. Thus, the proposed LLLR not only facilitates the binary hypothesis testing but also facilitates the density ratio estimation.
>
> Figure URL (Please also see Appendix F):
>
> https://github.com/authors-anonymous/ICLR2021/blob/main/LLLRvsCE_NMSE.png?raw=true

---

> > ### Author Response · Authors · 2020-11-25
> > **We disclosed the code of the toy model to ensure reproducibility**
> >
> > The code of the toy model is now included in Supplementary Material. Please find the folder named "LLLR_toymodel" in the .zip file.

---

> ### Author Response · Authors · 2020-11-17
> **Reply to Reviewer 5 (2/4)**
>
> >## The three tasks on which you test the models seem to be quite well solved after a few samples on average for all models. I think it would be worth testing the models on tasks that require more samples for reaching good performance and maybe where the temporality required (hyperparameter N) is larger.
>
> We agree that it is worth testing the models on tasks that require more samples, while our fair experiments and statistical tests indicate that the SPRT-TANDEM's superiority is fairly reproducible. Here, we ran an experiment on so challenging a dataset that the models require more timestamps to reach good performances. Please see below.
>
> ### NMNIST-HARD
>
> To test the SPRT-TANDEM on a harder database, we started to run experiments on _Nosaic MNIST-HARD_, where the MNIST handwritten digits are buried with heavier noise than the original NMNIST (only 10 pixels/frame are revealed, while it is 40 pixels/frame for the original NMNIST). The resulting speed-accuracy tradeoff curves below show that, while it takes more timestamps than the original NMNIST to attain the accuracy saturation, the SPRT-TANDEM outperforms LSTM-s/m more than the error-bar range. The curves themselves look similar to what we found on the three databases (NMNIST, UCF, and SiW).
>
> We updated our paper to include this result in Appendix D. Besides, we are planning to employ longer sequential datasets as a future work. However, again, our fair experiments and statistical tests in our paper indicate that the SPRT-TANDEM's superiority is reproducible, often lacking in modern machine learning research. For the statistical test details, please see Appendix H and the answer to your related question, "Why are the number of trials different?" below.
>
> Figure URL (Please also see Appendix D):
> https://raw.githubusercontent.com/authors-anonymous/ICLR2021/8b3e44eae3fa873a222ca9301f904acb8df21b71/binNMNIST-H%20(1).svg
>
> - Details:
>     - The speed-accuracy tradeoff curve. Please compare this with Figure 3 in the main text.
>     - "10th TANDEM" means the 10-th order SPRT-TANDEM.
>     - "19th TANDEM" means the 19-th order SPRT-TANDEM.
>     - Number of trials for hyperparameter tuning: 200 for all models.
>     - The error bars are standard error of mean (SEM).
>     - Number of trials for statistics: 440, 240, 200, and 200 for 10th TANDEM, 19th TANDEM, LSTM-s, and LSTM-m, respectively.
>
>
> >## It is not very clear to me what are the respective roles of LLLR and MCEL. I do not understand why both are useful,
>
> We employ both, because, in a nutshell, it shows the best performance (Figure 3 (e)).
>
> > I would have thought that CE in itself would be sufficient.
>
> Indeed, the multiplet cross-entropy loss (MCEL) can estimate the LLR through posterior estimation. Besides, the cross-entropy loss is a standard and stable loss for classification problems, and because of its stability, is widely used to train deep neural networks (DNNs). However, the MCEL alone may enhance the estimation error of density ratio, because it requires a division of two estimates. This problem has been a long-standing problem in density ratio estimation, which is recently named _density-chasm problem_ in an excellent work by Rhodes, Xu, and Gutmann ([Rhodes+2020](https://arxiv.org/abs/2006.12204)). Besides, from the viewpoint of DNNs, the MCEL tends to return too high confident values ([Corbière1+2019](https://papers.nips.cc/paper/2019/file/757f843a169cc678064d9530d12a1881-Paper.pdf)), causing extreme values of LLRs with high estimation errors.
>
> Therefore, the MCEL alone is likely to show poor performance for density ratio estimation, an essential task of the SPRT-TANDEM. The LLLR, on the other hand, directly estimates the density ratio and does not return too large/small values, as is discussed in Appendix E in detail; hence, the LLLR alleviates the problems above. This is an implicit but important motivation to use the LLLR. In fact, our additional experiment below supports our statements here.

---

> ### Author Response · Authors · 2020-11-17
> **Reply to Reviewer 5 (1/4)**
>
> >## The related work is quite superficial (even taking into account app B). In particular, I would have liked a deeper comparison with LSTM-s/m and EARLIEST, discussing the drawback/advantages of these methods with respect to SPRT-TANDEM.
>
> Thank you for the suggestion. Indeed, the first version did not explicitly state a detailed discussion of LSTM-s/m and EARLIEST results, though partially indicated in Section 3. Why is the SPRT-TANDEM superior to them? Please find the discussion below.
>
> The potential drawbacks common to the LSTM-s/m and EARLIEST is that they incorporate long temporal correlation: it may lead to (1) the class signature length problem and (2) vanishing gradient problem, as we described in Section 3. (1) If a class signature is significantly shorter than the correlation length in consideration, uninformative data samples are included in calculating the log-likelihood ratio, resulting in a late or wrong decision. (2) long correlations require calculating a long-range of backpropagation, prone to the vanishing gradient problem.
>
>
>
> An LSTM-s/m-specific drawback is similar to that of Neyman-Pearson test, in the sense that it fixes the number of samples before performance evaluations. On the other hand, the SPRT, and the SPRT-TANDEM, classify various lengths of samples: thus, the SPRT-TANDEM can achieve a smaller sampling number with high accuracy on average. Another potential drawback of LSTM-s/m is that their loss function explicitly imposes monotonicity to the scores. While the monotonicity is advantageous for quick decisions, it may sacrifice flexibility: the LSTM-s/m can hardly change its mind during a classification.
>
> EARLIEST, the reinforcement-learning based classifier, decides on the various length of samples. A potential EARLIEST-specific drawback is that deep reinforcement learning is known to be unstable ([Nikishin+2018](http://www.gatsby.ucl.ac.uk/~balaji/udl-camera-ready/UDL-24.pdf), [Kumar+2020](https://arxiv.org/pdf/2003.07305.pdf)).
>
> We incorporated the above discussion into the updated version of our paper: please see Appendix D (note that now the appendix is following the main text, within the same pdf). The supplementary related works (Appendix B) were also beefed up with more psychology/neuroscience papers and more recent machine learning papers, thanks to R2.
>
> >## In the proof of 4, just before eq 70 you say: "Let us assume that the process {x(s)}ts=1 is i.i.d., namely -> eq 70". This seems wrong to me. The assumption you're making there is that the process has independent component conditionally to the class y
> Strictly speaking, yes, it is "conditionally independently and identically distributed." We explicitly stated in Appendix C that it is conditionally independently and identically distributed.
>
> Note that in many SPRT works they just call it "i.i.d." For example, [Tartakovsky+2014](https://apps.dtic.mil/dtic/tr/fulltext/u2/a625103.pdf) says, "the observations are i.i.d. under both hypotheses," using the term "i.i.d." indicating conditional independence. Note that [Tartakovsky+2014](https://apps.dtic.mil/dtic/tr/fulltext/u2/a625103.pdf) sometimes just uses "i.i.d." as well. Another example papers using the term "i.i.d." are [Lai 1981](https://projecteuclid.org/download/pdf_1/euclid.aos/1176345398), [Finkelman 2008](https://www.tandfonline.com/doi/abs/10.1080/07474940802241033), [Liu+ 2011](https://ieeexplore.ieee.org/document/5977560), to name a few. Some of them indicated that its conditionally independent (e.g., Liu+2011: "If $s_k$ are i.i.d. (conditioned on parameter $\theta$")), others not.

---

### Author Response · Authors · 2020-11-17
**General message to ACs / Reviewers**

### Dear Area Chairs,
We would like to extend our gratitude for the hard work to supervise the reviewing process. We are looking forward to discussing our paper with you at Discussion stage 2.



### Dear reviewers,
We thank all of you for careful reading to appreciate the strengths of the paper:
- Multidisciplinary work of classical sequential hypothesis testing and modern density ratio estimation [R2, R5]
- Strong theoretical foundations [R1]
- Well-designed loss functions [R1, R3]
- Detailed [R1], compelling [R1, R2], and insightful [R5] experimental results
- Thoroughly covered related works [R1, R4]
- And the new database, Nosaic MNIST [R2, R4]

We are also encouraged to know that the paper is well-written [R2, R4, R5]. Moreover, we are excited to see insightful, critical, and creative comments.

Please find our official comments in each of the review threads. We are looking forward to discussing our paper with all of you.

---

### Comment · ~Akinori_F_Ebihara1 · 2021-04-09
**Re: Final Decision (2/2)**

>## baseline methods are too straightforward.
If I understand correctly, this is a new comment that none of the five reviewers mentioned. Thank you for raising the concern. Could you please provide details on what did you mean by "too straightforward"?

In the paper, we extensively compared our SPRT-TANDEM with baseline models, including the early-classifier LSTM-m/s, the reinforcement learning-based algorithm EARLIEST, and fixed-length approaches, Neyman Pearson Test and 3DResNet. If you have any other particular baseline methods in mind, we would love to hear your suggestion.


By "straightforward", you might have meant the SPRT-TANDEM and some of the baseline networks have common architectures, such as ResNet or LSTM. If that is the case, please note that this paper's goal was not to study neural network _architecture_, but to study an effective _usage_ of a neural network for fast and accurate sequential decision making. In other words, it was important to have a common network architecture as much as possible to have a fair comparison across the baseline. In our paper, we kept the network backbone as similar as possible while comparing the sequential decision making strategy across baselines - namely sequential density ratio estimation (SPRT-TANDEM) vs. monotonically increasing logits (LSTM-m/s) vs. reinforcement learning (EARLIEST). Adding variants of the network (e.g., SE-ResNet (Hu+2017), ResNeXt (Xie+2016), LResNet50E (Deng+2018), etc.) is always possible, but it disables a fair comparison across baselines or drastically increases experiments to make the paper out of focus.

>## Also, it is not clear how the proposed algorithm can handle the data with sparse observations (data with idle times in the middle).
If I understand correctly, this is a new comment that none of the five reviewers mentioned. Thank you for raising a new issue.

Handling sparse observations was not under the scope of this paper but will be solved by changing the temporal integrator's backbone. For example, SkipRNN (Campos+2018) is a modified recurrent neural network architecture to skip unwanted datapoints. SkipRNN is compatible with our TANDEM formula to estimate the probability density ratio sequentially.

Thus, handling sparse observations is an interesting extension of the SPRT-TANDEM. Although it would be a great future work to propose, we believe it is out of scope, and the possibility of the extension will not deteriorate our contribution in this paper.


>## Moreover, it does not provide rigorous stopping criteria although the authors proposed a simple method to determine the threshold, which is contradictory to the main objective of the proposed algorithm---making early predictions on sequential data---because the method requires "plotting the speed-accuracy tradeoff curve on the test dataset." This response implies that it at least requires a withheld dataset. Although this issue can be regarded as a separate problem, the paper could have provided an ablation study with respect to the criteria.

Firstly, we apologize for causing confusion by saying that "plotting the SAT curve on the \*_test_\* dataset," which can potentially sound contradictory: it can indicate that while we want to classify the test dataset as soon as possible, we need to sample all the sequence to determine the threshold.

What we really meant is to determine the threshold with some subset of data used during training, for example training or validation data. This procedure is entirely a common practice in supervised learning, and the fixed threshold will work just fine under unseen test data given that domain discrepancy is not large.

An important point here is the threshold's flexibility is an advantage of the SPRT-TANDEM, not a disadvantage. In practice, we sometimes keep the threshold unfixed until deployment so that the model can adapt to environmental changes (e.g., lighting conditions, camera parameters). The SPRT-TANDEM can flexibly adapt its speed-accuracy tradeoff without retraining, which is a missing feature in most of the other deep neural networks, as we wrote in response to R2.


# Summary
On the basis of the above discussion, we respectfully disagree with the final evaluation comment,
>### high scores but relatively low supports and confidences, and practical limitations.

The SPRT-TANDEM is actually already implemented on one of our commercial products and actively utilized under real-world scenarios.

However, we are open to discussion anytime - and thanks to OpenReview's great system, we can now continue the discussion without concerning our interests (i.e., freed from publication pressure). Further comments are always welcome - it will definitely fuel our future work.

Sincerely,
Akinori

---

### Comment · ~Akinori_F_Ebihara1 · 2021-04-09
**Re: Final Decision (1/2)**

Dear Conference Program Chairs,

Thank you again for organizing the review process. It was our pleasure to discuss our paper in detail with experts in the research field. We are looking forward to presenting our work at the upcoming Spotlight session.

On the other hand, we noticed that you itemized several concerns of our work to conclude that our work had "high scores but relatively low supports and confidences." We would like to resolve the raised concerns below and would very much like to hear back from you, if possible.

>## the evaluation of this work is relatively weak because synthetic or simple datasets were employed for the experiment
In addition to the Nosaic-MNIST database, we used two real-world databases, UCF and SiW, widely used action recognition and face spoofing detection databases. We wonder what you exactly mean by these databases are "simple."

If you meant to indicate that the SPRT-TANDEM's balanced accuracy is too high on this database, as R5 mentioned, we clearly responded during the rebuttal period. As a result, R5 raised the rating from 6 to 7. Although R5 did not leave additional comments, we believe that the raised rating indicated that our explanation was convincing. We summarize our points below:

### Statistical tests
To show that the SPRT-TANDEM does not just "barely win" but achieve statistically significantly better performance than other baselines, we conducted statistical tests. We ran our SPRT-TANDEM and baseline methods multiple times with different random initial values to conduct statistical tests, namely two-way ANOVA followed by Tukey-Kramer multi-comparison test. To ensure a fair comparison, all the models use the best hyperparameters objectively found with Optuna so that no models have disadvantages. The results of the statistical tests indicated that the SPRT-TANDEM achieved statistically signicantly better performance on all the tested databases. In other words, the SPRT-TANDEM reproducibly outperforms existing models.

As additional information, we provide a direct comparison between the SPRT-TANDEM and the second-best algorithm on the UCF and SiW databases. All numbers are calculated based on average performance, not a single run.

UCF database:
- The SPRT-TANDEM is 3.3 times faster than the second-best algorithm to achieve 96.0% balanced accuracy.
- The SPRT-TANDEM reduces error (i.e., 100 - balanced accuracy in %) by 45.1% compared with the second-best algorithm at the 15th frame.

SiW database:
- The SPRT-TANDEM is 24.5 times faster than the second-best algorithm to achieve 99.88% balanced accuracy.
- The SPRT-TANDEM reduces error by 41.2% compared with the second-best algorithm at the 15th frame.

### Nosaic-MNIST-HARD
We tested the SPRT-TANDEM on a challenging dataset on which the algorithms require more timestamps to reach good performances. Coined as Nosaic-MNIST-HARD, the database contains MNIST handwritten digits that are buried with heavier noise than the original Nosaic-MNIST  (only 10 pixels/frame are revealed, while it is 40 pixels/frame for the original NMNIST).

The resulting speed-accuracy tradeoff curves below show that, while it takes more timestamps than the original NMNIST to attain the accuracy saturation, the SPRT-TANDEM outperforms LSTM-s/m more than the error-bar range. The curves themselves look similar to what we found on the three databases (NMNIST, UCF, and SiW). See the figure below for the speed-accuracy-tradeoff (SAT) curve of the Nosaic-MNIST-HARD database.

Figure URL (Please also see Appendix D): https://raw.githubusercontent.com/authors-anonymous/ICLR2021/8b3e44eae3fa873a222ca9301f904acb8df21b71/binNMNIST-H%20(1).svg

### Interim summary
Through our fair experiments and statistical tests, we showed that the SPRT-TANDEM consistently, reproducibly, and statistically significantly outperformed on five publicly available databases, namely Nosaic-MNIST, Moving-MNIST (see Appendix), UCF, SiW, and Nosaic-MNIST-HARD. It is true that some of them are synthetic in the sense that they are artificially noisified: however, it is hard to believe that the artificially added noise provides an unfair advantage to our proposed method.

If you still expect that the SPRT-TANDEM can fail on another particular database, we would love to know its concrete reason with the database name.  - it will facilitate our future work greatly.

---

### Decision · Program_Chairs · 2021-01-07
**Final Decision**

**Decision:**

Accept (Spotlight)

**Comment:**

This paper presents a density ratio estimation approach to make the early decision for sequential data. The main contribution of this paper is the mathematical soundness of the proposed algorithm and all reviewers are unanimously positive about this paper with pretty good scores (7, 8, 6, 9, 7). However, despite the good scores, the verbal comments by the reviewers are not very strong except for one reviewer (R2); the reviewer with the highest score (9) did not provide detailed information about his/her rating. Also, the evaluation of this work is relatively weak because synthetic or simple datasets were employed for the experiment and the baseline methods are too straightforward. Also, it is not clear how the proposed algorithm can handle the data with sparse observations (data with idle times in the middle). Moreover, it does not provide rigorous stopping criteria although the authors proposed a simple method to determine the threshold, which is contradictory to the main objective of the proposed algorithm---making early predictions on sequential data---because the method requires "plotting the speed-accuracy tradeoff curve on the test dataset." This response implies that it at least requires a withheld dataset. Although this issue can be regarded as a separate problem, the paper could have provided an ablation study with respect to the criteria.

Considering all these facts--high scores but relatively low supports and confidences, and practical limitations, I would recommend accepting this paper as a spotlight presentation.